# Extracellular vesicles adhere to cells primarily by interactions of integrins and GM1 with laminin

Tatsuki Isogai[1], Koichiro M. Hirosawa[2], Miki Kanno[3], Ayano Sho[4], Rinshi S. Kasai[5], Naoko Komura[2], Hiromune Ando[1,2,3,4,8], Keiko Furukawa[6], Yuhsuke Ohmi[6], Koichi Furukawa[6], Yasunari Yokota[7], and Kenichi G.N. Suzuki[1,2,3,4,5,8]

**Tumor-derived extracellular vesicles (EVs) have attracted significant attention, yet the molecular mechanisms that govern their specific binding to recipient cells remain elusive. Our in vitro study utilizing single-particle tracking demonstrated that integrin heterodimers comprising α6β4 and α6β1 and ganglioside, GM1, are responsible for the binding of small EV (sEV) subtypes to laminin. EVs derived from four distinct tumor cell lines, regardless of size, exhibited high binding affinities for laminin but not for fibronectin, although fibronectin receptors are abundant in EVs and have functional roles in EV-secreting cells. Our findings revealed that integrins in EVs bind to laminin via the conventional molecular interface, facilitated by CD151 rather than by inside-out signaling of talin-1 and kindlin-2. Super-resolution movie observation revealed that sEV integrins bind only to laminin on living recipient cells. Furthermore, sEVs bound to HUVEC and induced cell branching morphogenesis in a laminin-dependent manner. Thus, we demonstrated that EVs predominantly bind to laminin on recipient cells, which is indispensable for cell responses.**

## Introduction

Extracellular vesicles (EVs) of a variety of sizes (∼30–1,000 nm) contain a diverse array of cargo, including microRNAs, metabolites, and proteins, and play a critical role in intercellular communication (Valadi et al., 2007; Raposo and Stoorvogel, 2013; van Niel et al., 2018). EVs have attracted extensive attention in a wide range of research fields (Théry et al., 2018; Witwer et al., 2021). In particular, several reports have suggested that tumor-derived EVs can induce phenotypic changes in recipient cells, creating a premetastatic niche that promotes cancer metastasis (Peinado et al., 2012; Ono et al., 2014). Small EV (sEV) studies might elucidate the underlying mechanism of the "seed and soil" hypothesis (Paget, 1989), which posits that cancer cells tend to metastasize and proliferate in organs suitable for these activities (Hart and Fidler, 1980; Peinado et al., 2017). A previous proteomic study of sEVs revealed that integrins α6β4 and α6β1 in sEVs from 4175-LuT human breast cancer cells were associated with lung metastasis, while αVβ5 in sEVs from BxPC3 human pancreatic adenocarcinoma cells was linked to liver metastasis (Hoshino et al., 2015). In addition, several studies have shown that blocking integrin in sEVs through anti-integrin antibodies or integrin ligands inhibits the internalization of integrin by recipient cells (Nazarenko et al., 2010; Altei et al.,

2020). However, no direct evidence that integrin heterodimers in sEVs bind to the ECM components on recipient cell plasma membranes (PMs) has been reported. In addition, the binding affinities between EVs and diverse ECM components have yet to be determined via comparative analysis. Furthermore, the detailed molecular-level mechanisms that govern the selective binding of sEVs to recipient cell PMs remain elusive. Elucidating these mechanisms is critical for developing effective strategies to inhibit cancer metastasis (Möller and Lobb, 2020; Marar et al., 2021). Moreover, it is essential to validate the biological functions of sEVs and their roles in physiological and pathological processes (Gurung et al., 2021).

In this study, we aimed to investigate whether integrin heterodimers play a critical role in the binding of sEVs to ECM components on recipient cell PMs. Furthermore, we sought to elucidate the mechanisms of selective binding using our cutting-edge imaging techniques. Since we demonstrated that sEVs can be classified into several subtypes with different tetraspanin marker proteins (CD63, CD81, and CD9) and that each subtype has unique properties (Hirosawa et al., 2025), it was crucial to examine the binding of each sEV subtype to ECM components. To this end, we first established an in vitro assay system that

---

[1]The United Graduate School of Agricultural Science, Gifu University, Gifu, Japan; [2]Institute for Glyco-core Research (iGCORE), Gifu University, Gifu, Japan; [3]Graduate School of Natural Science and Technology, Gifu University, Gifu, Japan; [4]Faculty of Applied Biological Sciences, Gifu University, Gifu, Japan; [5]Division of Advanced Bioimaging, National Cancer Center Research Institute (NCCRI), Tokyo, Japan; [6]Department of Biomedical Sciences, Chubu University, Kasugai, Japan; [7]Department of Information Science, Faculty of Engineering, Gifu University, Gifu, Japan; [8]Innovation Research Center for Quantum Medicine, Graduate School of Medicine, Gifu University, Gifu, Japan.

Correspondence to Kenichi G. N. Suzuki: suzuki.kenichi.b7@f.gifu-u.ac.jp.

enabled us to quantitatively evaluate the binding of three sizes of EVs (sEVs; medium-sized EVs, mEVs; and microvesicles, MVs) to ECM components on glass by single-particle tracking with single-molecule detection sensitivity. Second, we developed a state-of-the-art imaging system that enables the acquisition of super-resolution (dSTORM) movies, as opposed to static images. These advances enabled us to examine the binding affinities of various EV subtypes to ECM components on living cell PMs.

## Results

### Characterization of sEVs by single-particle tracking

To assess whether integrin subunits in tumor-derived sEVs are essential for selective ECM binding, we characterized sEVs isolated from human prostate cancer (PCa) (PC3) cells using ultracentrifugation (Fig. S1 A). Analysis by transmission electron microscopy (TEM) showed circular sEVs with a diameter of 83 ± 19 nm (mean ± SD) (Fig. S1 B), while measurement by a tunable resistive pulse sensing instrument (qNano) yielded 69 ± 17 nm (Fig. S1 C).

Our recent study by single-particle imaging revealed distinct PC3-derived sEV subtypes with different tetraspanin markers (Hirosawa et al., 2025). About 40% of the sEVs contained CD63, CD81, and CD9, while the remaining 60% had only CD63. Notably, CD63 only sEVs colocalized with caveolae in the recipient cell PMs, unlike triple-positive sEVs, indicating the subtype-specific interaction with recipient cell PM structures (Hirosawa et al., 2025). Therefore, in this study, to explore integrin heterodimer roles in sEV subtypes, we isolated PC3-derived sEVs stably expressing CD63-Halo7, CD81-Halo7, or CD9-Halo7. These were labeled with SaraFlour650T (SF650T) ligand, referred to as "sEV-tetraspanin-Halo7-SF650T," and successfully imaged using total internal reflection fluorescence microscopy (TIRFM) with no contamination from free dye molecules or autofluorescence from non-labeled sEVs (Fig. S1 D). Since the estimated average numbers of CD63⁻, CD81⁻, and CD9-Halo7 molecules per sEV are between 4.2 and 4.8 (Hirosawa et al., 2025), according to the Poisson distribution, >98% of sEVs should contain at least one Halo7-tagged tetraspanins. To standardize sEV concentrations, we used single-particle imaging with single-molecule detection sensitivity. We obtained TIRFM images (Fig. S1, E, G, and I) and a calibration curve (Fig. S1, F, H, and J). sEV solutions were adjusted to 0.8–4 × 10¹⁰ particles/ml to compare ECM binding on glass across different cell-derived sEVs.

### Knockout of integrin subunits

We employed CRISPR-Cas9 gene editing to knock out integrin β1, β4, α2, or α6 subunits in PC3 cells to examine their roles in sEV binding to ECMs. Western blotting of sEVs from intact cells showed the presence of the integrin β1, β4, α2, α6, α3, α5, and αV subunits, along with higher CD63, CD81, and CD9 levels than parental cells (Fig. 1 A). The integrin β1, β4, α2, and α6 subunits were indeed eliminated in sEVs derived from integrin-KO cells. Additionally, integrin α2 and α3 were undetectable in sEVs from integrin β1 KO cells (Fig. 1 A). Moreover, integrin β4 was not

detected in sEVs from integrin α6 KO cells, and integrin α6 expression was decreased in sEVs from β4 KO cells (Fig. 1 A). Thus, integrin KO in sEVs often decreases expression levels of the heterodimer counterparts (Fig. 1 B) (Hynes, 2002), while other integrins remain unaffected.

### Integrin β1 in tumor-derived sEVs plays a critical role in binding to laminin and collagen type I on glass

Integrins mediate bidirectional signaling, namely, inside-out and outside-in signaling. For instance, intracellular signaling molecules like talin activate integrins (Hynes, 2002; Moser et al., 2009; Lu et al., 2022). Although sEVs contain integrin heterodimers for ECM receptors, this does not necessarily imply that the integrin heterodimers in sEVs can bind to the ECM. To determine whether integrin subunits in tumor-derived sEVs mediate ECM binding, we performed in vitro binding assays. dSTORM images confirmed uniform coating of fibronectin, laminin, and collagen type I on glass (Fig. 1 C). Nearly all sEV-secreting cells attached to and spread on these ECM components at similar rates (Fig. 1, D and E), indicating that integrin heterodimers functioning as receptors for all three ECM components are active on the cell PMs.

Fig. 2 A shows TIRFM images of individual sEV–CD63-Halo7-SF650T particles derived from WT PC3 cells (PC3-WT) and integrin β1 KO cells (PC3-β1KO), attached to fibronectin, laminin, or collagen type I–coated glass. Quantitative analysis revealed very small numbers of all the sEV subtypes bound to fibronectin per image area (82 × 82 μm), with no reduction in integrin β1 KO sEVs (left in Fig. 2 B; Fig. S2, A and B; and Table S1). In contrast, binding of all the sEV subtypes to collagen type I was higher but significantly reduced by integrin β1 KO (middle in Fig. 2 B; Fig. S2, A and B; and Table S1). Moreover, binding of all the sEV subtypes to laminin was highest and drastically decreased in integrin β1 KO sEVs (right in Fig. 2 B; Fig. S2, A and B; and Table S1), indicating that integrin β1 in PC3-derived sEVs mediates the sEV binding to laminin and collagen type I.

A previous study reported that myeloma-derived sEVs isolated by ultracentrifugation are covered with fibronectin, a heparan sulfate–binding ligand (Purushothaman et al., 2016). Consequently, the enrichment of fibronectin in sEVs marginally (by a factor of ~1.2) amplifies the interactions between sEVs and recipient cells. Therefore, to examine the presence of fibronectin and laminin on PC3-derived sEVs, we performed immunoprecipitation (Fig. S3 A). However, neither fibronectin nor laminin was detected in the fraction isolated by the immunoprecipitation of sEVs with anti-CD63 antibody (Fig. S3 B). Furthermore, tetraspanin markers (CD63, CD81, or CD9) were not detected in the fraction obtained by the immunoprecipitation of sEVs with anti-fibronectin antibody (Fig. S3 C), indicating that neither laminin nor fibronectin was present on PC3-derived sEV surfaces.

### Integrin α2 in tumor-derived sEVs is responsible for binding to collagen type I on glass

The in vitro assay revealed that integrin β1 is essential for all sEV subtypes to bind to collagen type I and laminin on glass. Thus,

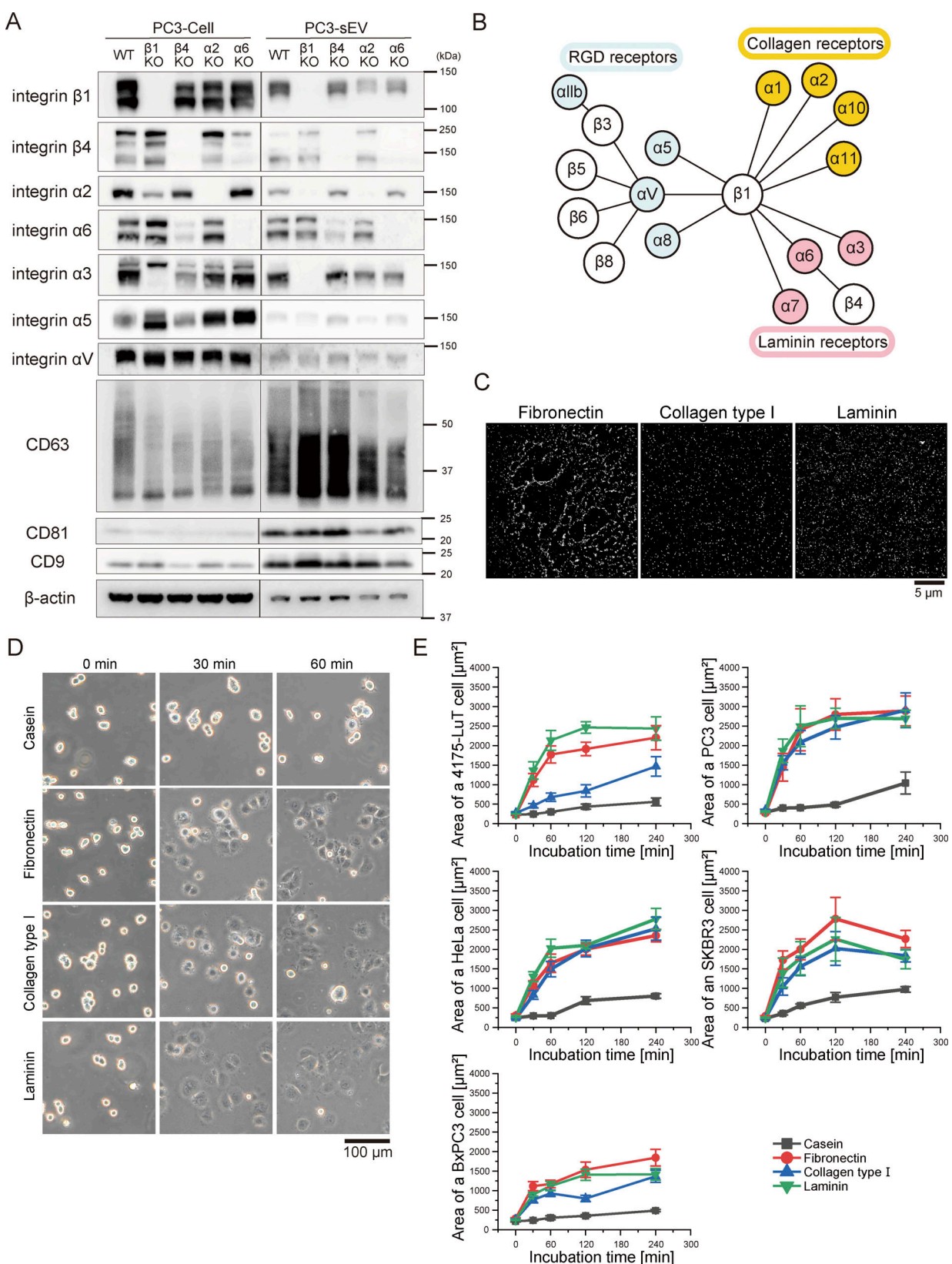

Figure 1. **Western blotting of integrin subunits in PC3 cells and sEVs and spreading assay of the cells that secreted sEVs in this study on all three ECMs (fibronectin, laminin, and collagen type I) coated on glass. (A)** sEVs were isolated from the cell culture supernatant of intact PC3 cells or from cells in which an integrin subunit (β1, β4, α2, or α6) was knocked out via the CRISPR-Cas9 method. **(B)** The correlation map of integrin heterodimers and the ECM. **(C)** dSTORM images of the ECM (left-top: fibronectin, left-bottom: collagen type I, and right: laminin) coated on glass. **(D)** Images of HeLa cells attached to glass coated with ECM components (fibronectin, collagen type I, and laminin) or casein after 0, 30, or 60 min of incubation. **(E)** The time course of the area of five tumor cell lines on glass coated with ECM components (fibronectin, collagen type I, and laminin) or casein (n = 8 cells). Data are presented as the mean ± SE. Source data are available for this figure: SourceData F1.

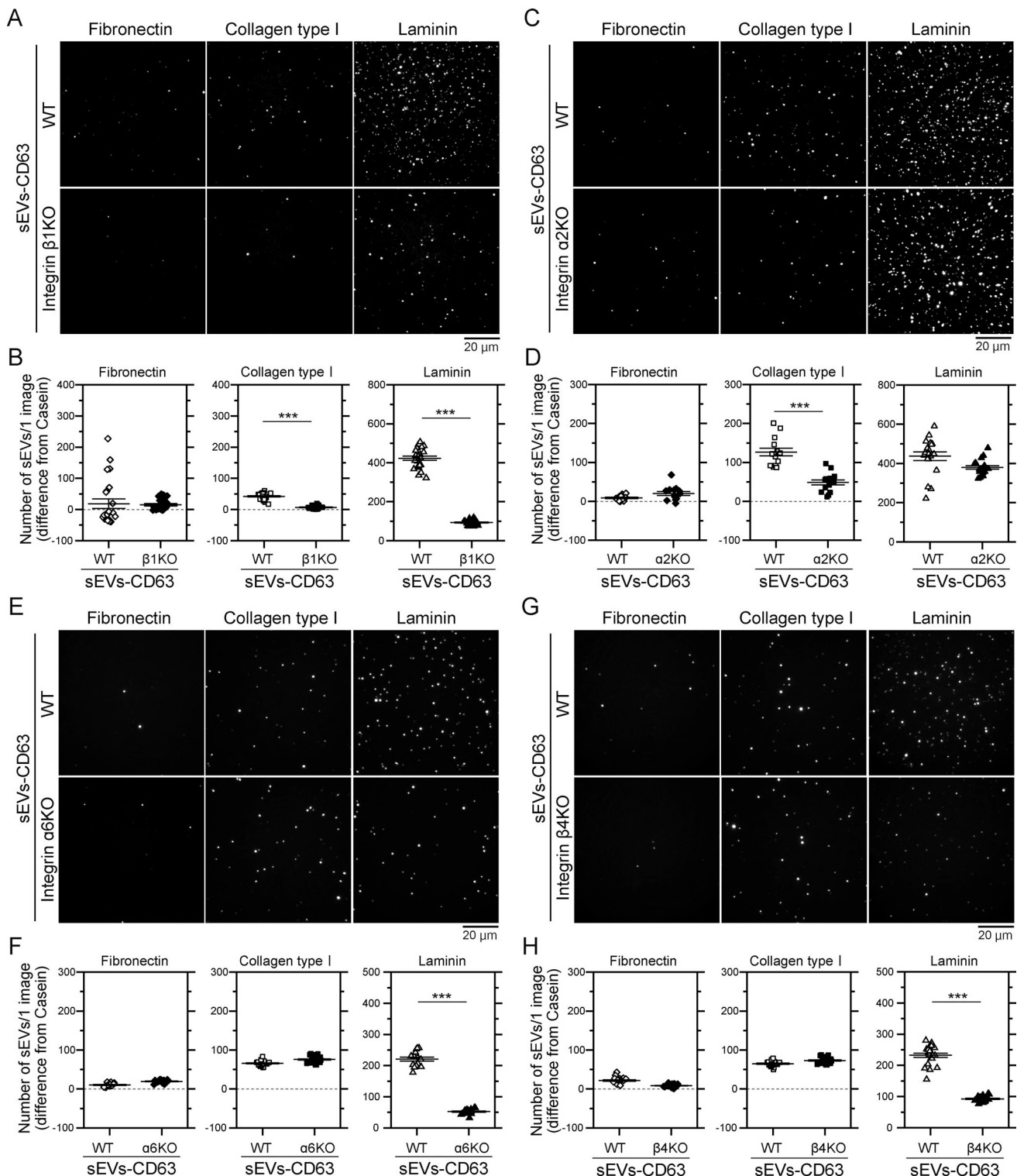

Figure 2. **Integrin α2β1 in sEVs derived from PC3 cells is responsible for the binding of CD63-containing sEVs to collagen type I, and integrins α6β1 and α6β4 are responsible for the binding to laminin. (A, C, E, and G)** Single-particle fluorescence images of sEV–CD63Halo7-SF650T on glass coated with fibronectin, collagen type I, and laminin before and after the KO of integrin β1 (A), integrin α2 (C), integrin α6 (E), and integrin β4 (G). **(B, D, F, and H)** The numbers of sEVs attached to glass coated with these ECM molecules before and after the KO of integrin β1 (B), integrin α2 (D), integrin α6 (F), and integrin β4 (H) (n = 16 images). Data are presented as the mean ± SE. n.s., nonsignificant difference; ***P < 0.001 according to Welch's t test (two-sided).

we focused on integrin heterodimers containing β1 subunit. Since integrin α2β1 is a receptor for collagen type I, we examined the α2 subunit. Fig. 2 C shows TIRFM images of individual particles of sEV–CD63Halo7-SF650T derived from PC3-WT and PC3-α2KO cells, attached to ECM components on glass. Notably, the expression levels of other integrin subunits in PC3-α2KO cells were comparable with those in PC3-WT cells (Fig. 1 A). Quantitative analysis showed a significant decrease in the number of all sEV subtypes from α2KO cells bound to collagen type I, while the number that bound to laminin was not significantly reduced (Fig. 2 D; Fig. S2, C and D; and Table S1). These findings, in conjunction with the results of β1KO, show the crucial involvement of integrin α2β1 in the sEV binding to collagen type I.

**Integrins α6 and β4 in tumor-derived sEVs are responsible for binding to laminin on glass**

We next investigated the involvement of the counterpart in β1-containing integrin heterodimers in the sEV binding to laminin. As integrin α6β1 is a laminin receptor, we examined whether the absence of α6 affects the sEV binding to laminin on glass. Western blotting showed a lack of integrin β4 in sEVs from PC3-α6KO cells (Fig. 1 A). In addition, we observed a considerable decrease in the expression level of integrin α6 in EVs from β4KO cells (Fig. 1 A). Therefore, it is plausible that the expression levels of integrins α6 and β4 in sEVs are closely linked. Fig. 2, E and G, show TIRFM images of single particles of sEV–CD63Halo7-SF650T derived from PC3-WT, α6KO, and β4KO cells, attached to ECM components on glass. We discovered that knocking out integrin α6 or β4 considerably impaired the binding of all sEV subtypes to laminin but did not significantly affect the binding to fibronectin or collagen type I (Fig. 2, F and H; Fig. S2, E–H; and Table S1). These findings explicitly show that integrin α6β1 and α6β4 in sEVs mediate sEV–laminin interactions on glass.

Thus, single-particle imaging revealed that integrin α2β1 in sEVs is essential for binding to collagen type I, while integrin α6β1 and α6β4 in sEVs mediate binding to laminin (Figs. 2 and S2; and Table S1). Interestingly, despite the presence of fibronectin receptors (α5β1, αVβ1, and αVβ5) in sEVs (Fig. 1, A and B; and Fig. 3 F), only small amounts of all the PC3-derived sEV subtypes bound to fibronectin on glass. Notably, these fibronectin receptors are functional in PC3 cells, as shown by their similar spreading on laminin, fibronectin, and collagen type I–coated glass (Fig. 1 E).

**GM1, but not other gangliosides, in sEVs specifically binds to laminin**

Our results showed that knockout of integrin α6, β4, or β1 greatly reduced but did not eliminate the binding affinity of PC3-derived sEVs for laminin, suggesting that other molecules may be involved. A previous study suggests that laminin may bind directly to GM1 on dorsal root ganglion neurons (Ichikawa et al., 2009). Our dot blot analysis showed that the relative density of GM1 (Fig. S4 A) in PC3-derived sEVs was about 29.7 ± 8.1 (mean ± SE, $n$ = 3) times greater than that in PC3 cells (Fig. S4 B), suggesting that GM1 might contribute to sEV-laminin

binding. To test this, we attempted to isolate cell lines expressing only one ganglioside by overexpressing (OE) ganglioside synthase in cells to produce or deplete gangliosides, as reported previously (Yamashiro et al., 1995; Yesmin et al., 2023). However, this strategy did not work in PC3 cells because the amounts of gangliosides produced or depleted by the synthase returned to the original levels during cell culture. Instead, we used B78 cells, a subclone of B16 melanoma cells that can predominantly express specific ganglioside (Fig. S5). The number of laminin-bound sEV particles, normalized to integrin α6/CD81 levels in the sEVs (Fig. S4 C), was approximately threefold higher for sEVs from B78 cells predominantly expressing GM1 than those expressing GM2, GM3, GD2, or GD2/GD3 (Fig. S4, A, D, and E). Regardless of ganglioside type, B78-derived sEVs showed low fibronectin binding (Fig. S4, F and G). These results suggest that GM1, but not the other examined gangliosides in sEVs, might bind to laminin, and all the gangliosides in sEVs might not bind to fibronectin.

To further examine whether GM1 binds directly to laminin, we investigated whether ~100-nm liposomes containing GM1 are bound to laminin but not to fibronectin. We used liposomes containing 1.0 mol% GM1, which was estimated to be equivalent to sEV levels for the binding assay (see the Materials and methods section). Fig. S4, H and I, show that GM1-containing liposomes bound extensively to laminin, whereas those with other gangliosides showed minimal binding. Conversely, liposomes hardly bound to fibronectin, regardless of ganglioside type (Fig. S4, H and J). These results confirm that GM1 in sEVs directly binds to laminin but not fibronectin.

We subsequently assessed the relative contributions of GM1 and integrin β1 in sEV-laminin binding using a competitive inhibition assay. The number of PC3-derived sEVs–CD63Halo7-TMR bound to laminin on glass decreased by 61% after integrin β1 KO (Fig. S4, K and L). In the presence of 0.5 mM GM1 glycan, the numbers of laminin-bound sEVs from PC3-WT and integrin β1 KO cells decreased by 18% and 34%, respectively (Fig. S4, K and L). These results show that while integrin heterodimers in the sEVs predominantly mediate laminin binding, GM1 glycan in the sEVs also contributes to the interaction.

**All the sEVs, mEVs, and MVs derived from four tumor cell lines exhibited marked binding affinity for laminin but minimal binding to fibronectin**

To investigate whether high laminin binding and low fibronectin binding are common in tumor-derived sEVs, we performed in vitro binding assays using sEV–CD63-Halo7-SF650T particles from four tumor cell lines. qNano measurements showed sEV diameters of 83 ± 16 nm (mean ± SD) (4175-LuT), 97 ± 9 nm (BxPC3), 83 ± 12 nm (HeLa), and 75 ± 8 nm (SKBR3) (Fig. 3 A). Our in vitro binding assay revealed that sEVs from 4175-LuT, PC3, BxPC3, and SKBR3 bound to laminin much more extensively than fibronectin on glass (Fig. 3, B and C). On the other hand, HeLa-derived sEVs showed minimal binding to both (Fig. 3 C). These results demonstrate that tumor-derived sEVs primarily bind to laminin on glass with low affinity for fibronectin.

**A**

| Cell lines | Diameters of EVs (mean ± S.D. nm) | | |
|---|---|---|---|
| | small EV | medium-sized EV | MV |
| 4175-LuT | 83 ± 16 | 90 ± 12 | 142 ± 14 |
| PC3 | 69 ± 17 | 90 ± 18 | 173 ± 28 |
| BxPC3 | 97 ± 9 | 103 ± 13 | 138 ± 27 |
| HeLa | 83 ± 12 | 97 ± 14 | 145 ± 31 |
| SKBR3 | 75 ± 8 | 85 ± 16 | 212 ± 47 |

Figure 3. **Tumor-derived sEVs bind much more strongly to laminin than to fibronectin. (A)** The diameters of sEVs, mEVs, and MVs derived from the 4175-LuT, PC3, BxPC3, HeLa, and SKBR3 cell lines were measured by qNano. **(B)** Single-particle fluorescence images of sEV–CD63Halo7-SF650T particles derived from the 4175-LuT, PC3, BxPC3, HeLa, and SKBR3 cell lines on glass coated with fibronectin or laminin. These sEVs were purified by ultracentrifugation

(200,000 × *g* for 4 h). **(C–E)** The numbers of EV–CD63Halo7-SF650T particles (sEVs [C], mEVs [D], and MVs [E]) attached to glass coated with ECM components (*n* = 16 images). mEVs and MVs were isolated by low-speed centrifugation (50,000 × *g* for 30 min) and (10,000 × *g* for 30 min), respectively. Data are presented as the mean ± SE. The numbers of sEVs, mEVs, and MVs derived from all cell lines applied to laminin- or fibronectin-coated glass were adjusted to the same. **(F)** Western blotting of integrin subunits in sEVs, mEVs, and MVs derived from 4175-LuT, PC3, BxPC3, HeLa, and SKBR3 cells. Source data are available for this figure: SourceData F3.

The diameter of EVs isolated by ultracentrifugation depends on variables such as sample volume and centrifugal duration (Théry et al., 2018). To determine whether the stronger binding affinity of EVs for laminin is a general phenomenon, we quantified the number of larger EVs bound to ECM-coated glass. This analysis was essential, as sEVs and MVs are reported to exhibit distinct membrane compositions (Jeppesen et al., 2019). Accordingly, we isolated larger EVs by ultracentrifugation at slower rates. The mEVs exhibited diameters of 90 ± 12 nm (4175-LuT), 90 ± 18 nm (PC3), 103 ± 13 nm (BxPC3), 97 ± 14 nm (HeLa), and 85 ± 16 nm (SKBR3) (Fig. 3 A). Our in vitro binding assay showed that these mEVs possess properties analogous to those of sEVs (Fig. 3 D). Western blotting revealed that the expression of fibronectin receptors, such as integrin α5β1 heterodimers, in the EVs derived from all examined tumor cell types (Fig. 3 F). Furthermore, MVs had diameters of 142 ± 14 (4175-LuT), 173 ± 28 (PC3), 138 ± 27 (BxPC3), 145 ± 31 (HeLa), and 212 ± 47 (SKBR3) (Fig. 3 A). As CD63 was not highly enriched in MVs (Fig. 3 F), MVs were labeled with a lipid probe, and their binding to laminin and fibronectin was quantified by TIRFM (Fig. 3 E). Our results reveal that MVs from 4175-LuT, PC3, and SKBR3 cells exhibited a stronger affinity for laminin than to fibronectin, whereas MVs from BxPC3 and HeLa cells showed minimal binding to both (Fig. 3 E). Despite the presence of fibronectin receptors, including integrins α5β1 and αVβ3, in all sizes of EV subtypes (Fig. 3 F), they exhibited minimal binding to fibronectin. Notably, tumor cells spread on fibronectin- and laminin-coated glass at comparable rates (Fig. 1 E), indicating that all receptors for ECM components on the PMs were functional. These results indicate that laminin receptors are markedly active, whereas fibronectin receptor activity is low in EVs from 4175-LuT, PC3, BxPC3, and SKBR3 cells.

### The integrin β1 and α6 subunits in sEVs are responsible for binding to cell PMs

Although the in vitro assay showed that integrin heterodimers are crucial for binding to laminin and collagen type I on glass (Figs. 2 and S2), their role in mediating sEV binding to recipient cell PMs remains unclear. To address this, we quantified sEV–CD63–SF650T particles bound to recipient cell PMs. Immunofluorescence imaging revealed that human fetal lung fibroblast (MRC-5) PMs were extensively coated with fibronectin, collagen type I, and laminin, whereas human marrow stromal (HS-5) cells exhibited minimal ECM coatings (Fig. 4, A and B). Laminin α5 isoform was significantly more abundant on MRC-5 PMs than α1 isoform, indicating that laminin-511, the same isoform used in the in vitro assay, represents the predominant isoform on MRC-5 PMs (Fig. 4 A). Then, the time course of sEVs binding to basal PMs of recipient cells was observed by TIRFM. After 30 min of incubation, the number of sEV–CD63–SF650T particles from

PC3-WT cells bound to the basal PM of MRC-5 cells was approximately twice that of sEVs from PC3-β1KO (P = 0.046; two-sided Welch's *t* test) or PC3-α6KO cells (P = 0.0082) (Fig. 4, C, E, F, and H). In contrast, PC3-β1KO–derived sEVs binding to HS-5 basal PMs was indistinguishable from that of PC3-WT–derived sEVs (P = 0.55) (Fig. 4, D and G). These results show that at least half of the sEVs bind to MRC-5 basal membranes due to active binding to pericellular laminin. To examine sEV binding to both apical and basal PMs of recipient cells, we performed confocal microscopy after chemical fixation. The number of PC3-WT–derived sEVs bound to MRC-5 PMs after 60 min incubation was approximately twice greater than that of PC3-β1KO–derived sEVs (P = 0.0022), while the numbers of these sEVs bound to HS-5 cell PMs were minimal and statistically indistinguishable (P = 0.30) (Fig. 4, I and J). The number of sEVs on MRC-5 PMs was ~11-fold greater than that on HS-5 cell PMs (Fig. 4 J). These results indicate that integrin β1 and α6 in sEVs are involved in binding to the MRC-5 PM, whose surface is densely covered by ECM components.

### Integrins in sEVs bind to laminin via the conventional molecular interface

We examined whether integrin β1 in PC3-derived sEVs adopts an active conformation upon laminin binding. To address this, we performed immunostaining of integrin β1 in sEV–CD63Halo7 bound to either uncoated or laminin-coated glass using antibodies specific for activated integrin β1 (clones HUTS-4 and HUTS-21, Luque et al., 1996) and a general anti-integrin β1 antibody (clone P5D2). Nearly all sEV–CD63Halo7-TMR particles bound to uncoated or laminin-coated glass were stained with P5D2 (Fig. 5, A–C). However, only 20–30% of the sEV particles on uncoated glass were stained with HUTS-4 and HUTS-21, whereas 60–70% of those bound to laminin-coated glass were positively stained with these activation-specific antibodies (Fig. 5, A–C). These results show that integrin β1 in sEVs adopts an active conformation upon laminin binding.

Next, we investigated whether integrin α6 in sEVs binds to laminin via the same molecular interface as the conventional interaction in cells. To this end, we compared the laminin-binding capacity of sEVs from integrin α6 KO-PC3 cells expressing the R155A dominant-negative integrin α6 mutant (Arimori et al., 2021) with sEVs from integrin α6-rescued cells. The levels of rescued integrin α6 and the R155A mutant in sEVs were comparable with the levels of endogenous integrin α6 in sEVs derived from PC3-WT cells (Fig. 5 D). The number of sEVs containing the integrin α6 R155A mutant bound to laminin was significantly lower than those containing endogenous or rescued WT integrin α6 but was comparable with that of sEVs from integrin α6 KO PC3 cells (Fig. 5, E and F). These results indicate that the interaction between integrin heterodimers in sEVs and

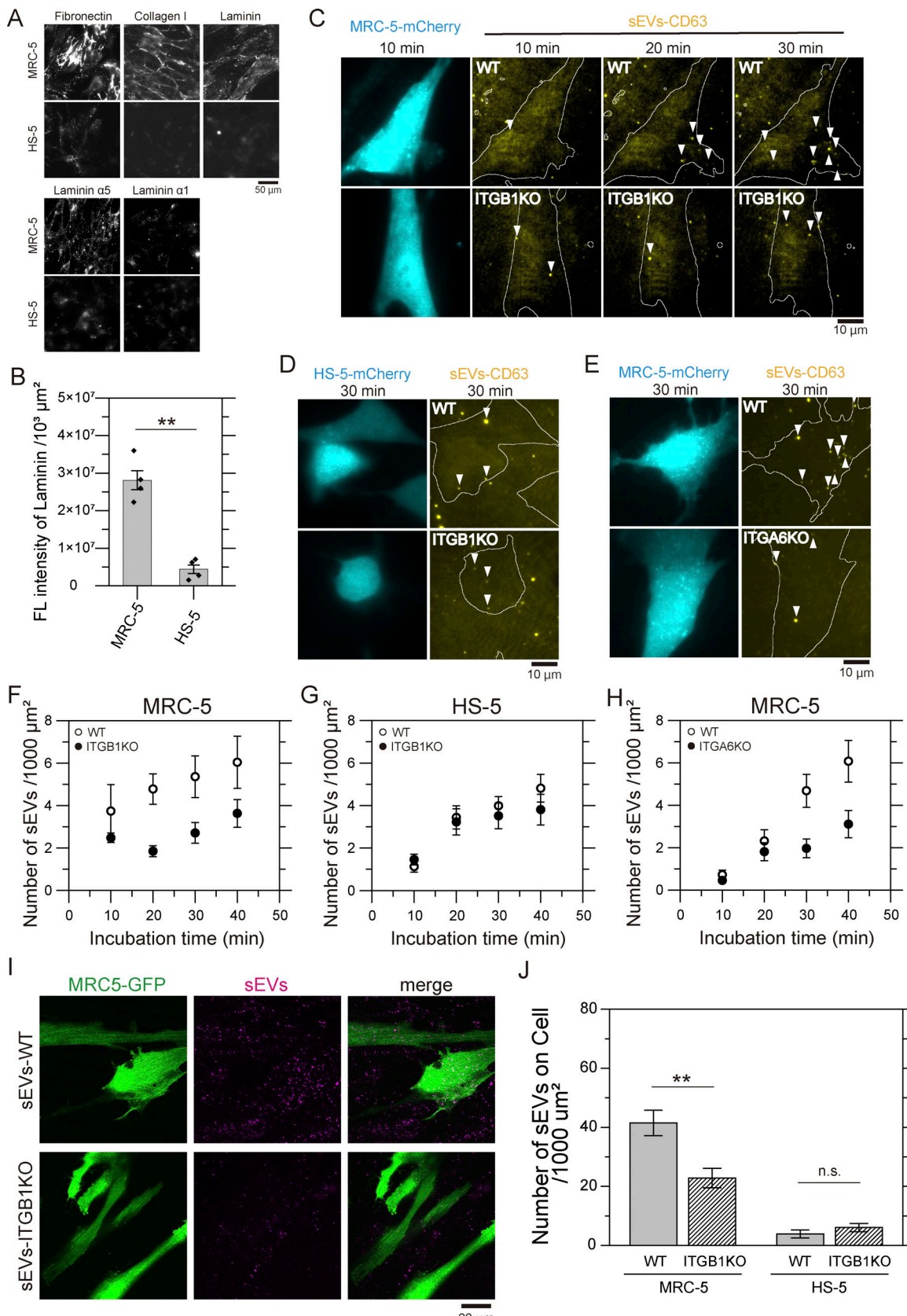

Figure 4. **Integrins β1 and α6 in sEVs and the ECM are responsible for the binding of sEVs to cell PMs. (A)** Immunofluorescence images of fibronectin, collagen type I, laminin, laminin α5, and laminin α1 in human normal embryonic lung fibroblast (MRC-5) cells and human bone marrow stromal (HS-5) cells. **(B)** The fluorescence intensities of laminin on cells in 1,000 μm² (n = 4 images). **(C)** TIRF images of MRC-5 cells expressing mCherry and sEV–CD63Halo7-

SF650T particles (arrowhead) attached to cell membranes after 10, 20, and 30 min of incubation. sEVs were isolated from the cell culture supernatant of intact PC3 cells (top panels) and from the cell culture supernatant of integrin β1 KO PC3 cells (bottom panels) bound to the basal surface of MRC-5 cells. **(D)** TIRF images of HS-5 cells and sEV–CD63Halo7-SF650T particles (arrowhead) attached to the basal surface of cell membranes after 30 min of incubation. sEVs were isolated from intact PC3 cells (top panels) and integrin β1 KO PC3 cells (bottom panels). **(E)** TIRF images of MRC-5 cells and the attached sEV–CD63Halo7-SF650T particles derived from intact PC3 cells (top panels) and from integrin α6 KO PC3 cells (bottom panels) after 30 min of incubation. **(F–H)** Time course of the numbers of intact sEVs and integrin β1 KO sEVs attached to the basal surface of the MRC-5 cell membrane ($n$ = 8 images) (F) and to the basal surface of the HS-5 cell membrane ($n$ = 15 images) (G) per 1,000 μm$^2$. **(H)** Time course of the numbers of intact sEVs and integrin α6 KO sEVs attached to the basal surface of the MRC-5 cell membrane per 1,000 mm$^2$ ($n$ = 12 images). Data are presented as the mean ± SE. **(I)** Fluorescence images of MRC-5 cells expressing GFP and sEV–CD63Halo7-TMR by confocal microscopy. sEVs were isolated from the cell culture supernatant of WT PC3 cells (top panels) and from that of integrin β1 KO PC3 cells (bottom panels). The cells were fixed after treatment of the sEVs for 1 h. **(J)** The numbers of sEVs attached to MRC-5 and HS-5 cell PMs by confocal fluorescence microscopy ($n$ = 10 images). In Fig. 4, since not all cells necessarily express mCherry or GFP, many fluorescent spots of sEVs were observed either in regions containing non-expression cells or potentially on the glass surface. Nevertheless, we can quantify the number of sEVs bound to both the apical and basal surfaces of the cell PM by counting the number of fluorescent EV spots on cells expressing mCherry or GFP. Data are presented as the mean ± SE. n.s., nonsignificant difference; **P < 0.01; ***P < 0.001 according to Welch's $t$ test (two-sided).

---

laminin relies on the same molecular interface as the conventional interaction in cells.

### Pseudo real-time super-resolution movie observation demonstrated that sEVs bind predominantly to laminin on the living cell PM

Having demonstrated the binding of integrin α2β1 in PC3-derived sEVs to collagen type I and the binding of both integrin α6β1 and α6β4 in sEVs to laminin on glass, we next examined whether these sEVs bind to ECM components on living recipient cell PMs. To directly assess this, we performed simultaneous single-particle tracking of PC3-derived sEVs–CD63-Halo7-TMR particles and super-resolution microscopy of ECM components on living MRC-5 PMs (Fig. 6, A–E). Since both sEVs and ECM components moved slowly on the PM and their positions changed throughout the observation, we sought to more accurately analyze the colocalization between sEVs and ECM components by acquiring super-

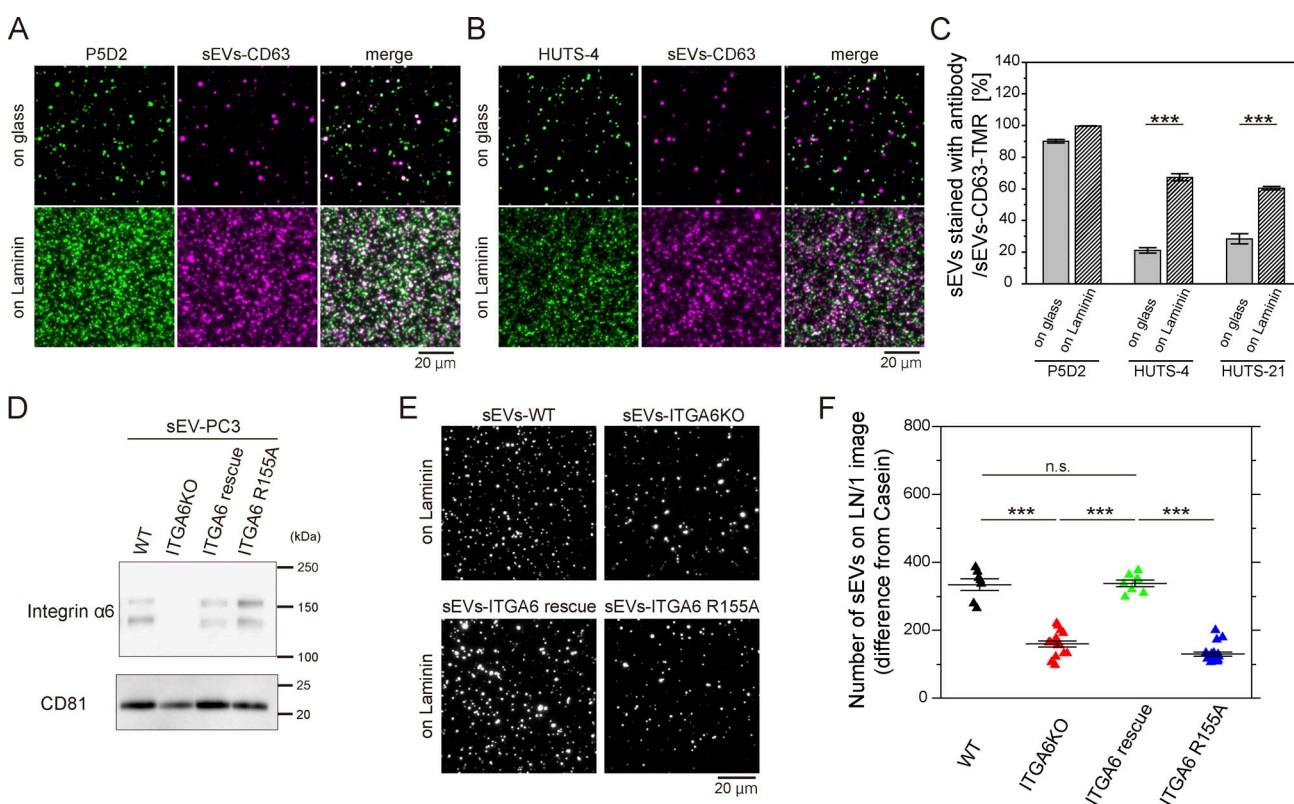

Figure 5. **Integrins in sEV bind to laminin through conventional integrin–laminin interactions. (A and B)** TIRF images of sEV-PC3-CD63Halo7-TMR stained with anti-activated integrin β1 HUTS-4 (A) and anti-integrin β1 P5D2 (B) on uncoated glass (top) or laminin-coated glass (bottom). **(C)** Colocalization ratio of fluorescent spots of sEVs stained with the antibody to those of sEV-PC3-CD63Halo7-TMR ($n$ = 10 images). **(D)** Western blot analysis of sEVs derived from WT, integrin α6 KO, integrin α6-rescued, and integrin α6 R155A-expressing PC3 cells. **(E)** Fluorescence images of the sEVs derived from WT, integrin α6 KO, integrin α6-rescued, and integrin α6 R155A-expressing PC3 cells bound to laminin on glass. **(F)** Quantification of sEVs bound to laminin on glass under the conditions described in E ($n$ = 8 images). Data are presented as the mean ± SE. n.s., nonsignificant difference; ***P < 0.001 according to Welch's $t$ test (two-sided). In F, due to the necessity of multiple statistical tests, the significance level was corrected by the Bonferroni method and determined to be 0.0125 (=0.05/4). Source data are available for this figure: SourceData F5.

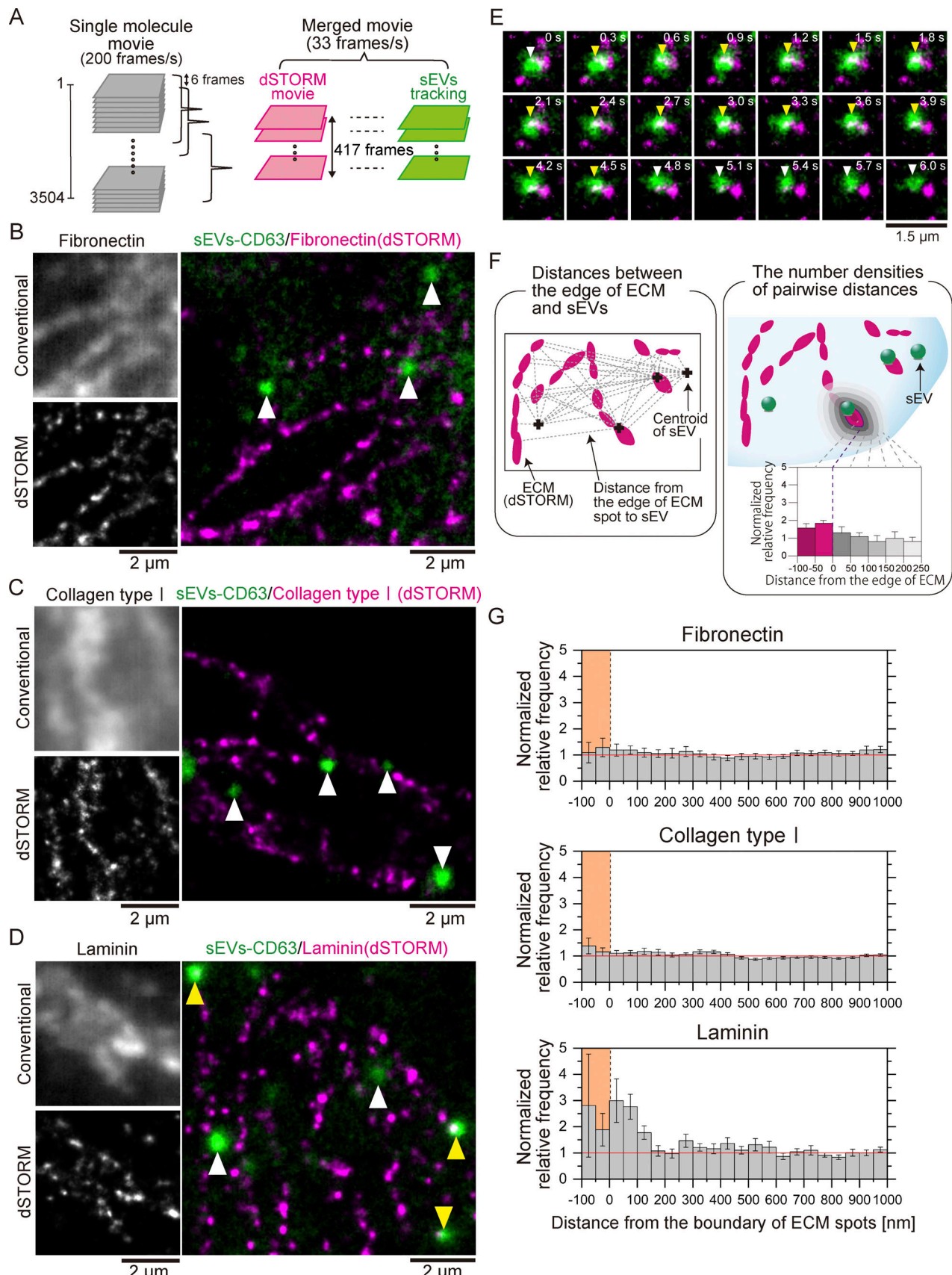

Figure 6. **sEVs bind to laminin on the PMs of living MRC-5 cells, as revealed by pseudo real-time super-resolution movie observation. (A)** Schematic diagram for generating merged movies of dSTORM images of the ECM structures and single-particle images of sEVs. Data acquisition was performed by

observing single-fluorescent molecules of ECM components immunostained with SF650B-conjugated antibodies at 200 frames/s (3,504 frames), and dSTORM images were reconstructed using the data acquired every 5.0 s (=1,002 frames). The first dSTORM image was reconstructed using frames 1–1,002, and the process was then repeated by shifting the initial frames backward by 6 frames; thus, 417 dSTORM images were obtained. These dSTORM still images were connected to construct the dSTORM movie. The single-particle movies of sEVs were subjected to a rolling average for 6 frames and synchronously merged with the dSTORM movie. **(B–D)** Conventional immunofluorescence images (top-left) and dSTORM images (bottom-left) of the ECM structures (B: fibronectin, C: collagen type I, and D: laminin) on the cells. dSTORM images of the ECM structures (magenta) and single-particle images of sEV–CD63Halo7-TMR particles (green) were merged (right). sEVs localized near (<100 nm) the boundary of ECM structures and sEVs localized alone are indicated by yellow and white arrowheads, respectively. **(E)** The image sequence (every 0.3 s) of laminin obtained by dSTORM (magenta) and a single sEV–CD63Halo7-TMR particle (green) on the living MRC-5 cell membrane. sEVs colocalized with laminin, as indicated by the yellow arrowhead. **(F)** Colocalization analysis method. We measured the nearest distance from an edge of the ECM structure to a centroid of the sEV spot and performed this measurement for all pairs of ECM structures and sEV particles. The normalized relative frequency was defined as the ratio of the average value of the spatial pair correlation function of the actual image to that of randomly distributed spots generated by a computer. We obtained histograms showing the distribution of the normalized relative frequency of sEVs at each distance from the edge of the ECM structures. Zero on the x axis indicates the contour of the ECM structures in the dSTORM images determined by the KDE method. When sEVs are enriched near the ECM structures, the normalized relative frequency is >1. **(G)** Probability density analysis of the sEVs and ECM structures. The colored areas indicate regions within the ECM structures. The sEV–CD63Halo7-TMR particles localized near the contour of laminin ($n$ = 20 cells).

resolution "dSTORM movies" of ECM components instead of still images as shown in Fig. 6 A. The pseudo real-time movies of the ECM components and sEV particles (33 frames/s) were superimposed (Fig. 6 A).

Conventional immunofluorescence microscopy failed to visualize the fibrillar ECM structure on living cells (left-top panels of Fig. 6, B–D). In contrast, live-cell dSTORM movie observation enabled clear visualization (left-bottom panels of Fig. 6, B–D). sEVs frequently colocalized with laminin structures on the PM (yellow arrowheads, Fig. 6 D, right; Video 1), and colocalization occasionally persisted over 4 s (Fig. 6 E and Video 2). However, colocalization between sEVs and fibronectin (Fig. 6 B, right, and Video 3) or collagen type I (Fig. 6 C, right, and Video 4) was rare. We quantitatively analyzed the colocalization events as shown in Fig. 6 F. Our analysis revealed that sEVs were significantly enriched near the edges of laminin structures at distances ranging from –100 to 150 nm on living MRC-5 PMs (Fig. 6 G, bottom). In contrast, sEVs were hardly detected near fibronectin- or collagen-type I structures (Fig. 6 G, top and middle, respectively). These results indicate that PC3-derived sEVs adhere predominantly to laminin on recipient cell PMs. Notably, we did not observe any binding of sEVs to collagen type I on the PM, although sEVs did bind to collagen I on glass (Fig. 2, A–D and Fig. S2, A–D). This difference may be attributed to the considerably lower sEV affinity for collagen type I than laminin, as shown in the glass assay (Figs. 2 and S2).

### Talin-1 and kindlin-2 do not facilitate the binding of integrin heterodimers in EVs to laminin

Talin binds to cytoplasmic tails of integrin β subunits and promotes the binding of integrin heterodimers to all ECM components (Sun et al., 2019). In PCa cells, talin-1, but not talin-2, plays a key role in integrin β1 activation, with Ser425 phosphorylation enhancing integrin β1 activity (Jin et al., 2015). Cdk5 kinase mediates talin-1 phosphorylation at Ser425 (Huang et al., 2009), triggering integrin β1 activation (Jin et al., 2015). The present study showed that the presence of fibronectin receptors in EVs does not correlate with fibronectin binding. Therefore, we hypothesized that factors other than the presence of ECM receptors are involved in regulating the binding of EVs with ECM components. We examined whether talin-1 and its phosphorylation

at Ser425 play pivotal roles in the activation of integrin heterodimers in sEVs.

Talin-1 knockdown (KD) significantly impaired PC3 and BxPC3 cell spreading on fibronectin, laminin, and collagen type I (Fig. 7, A–C). These results suggest that talin-1 in these cells plays a pivotal role in mediating cell adhesion via integrins and ECMs. sEVs from talin-1 KD or OE PC3 cells were isolated by ultracentrifugation, and we quantitatively analyzed the numbers of sEV-Halo7-integrinβ1-SF650T and sEV–CD63-Halo7-SF650T particles that bound to laminin-coated glass (Fig. 7, D–J). The expression level of talin-1 in sEVs was reduced to 40% after KD (top in Fig. 7 D). Notably, we observed that the binding of talin-1 KD cell-derived sEVs to laminin was similar to that of sEVs from intact cells (top in Fig. 7, E and F; top in Fig. 7, H and I). Furthermore, talin-1 OE in sEVs (2.3-fold increase; Fig. 7 D, bottom) did not alter the number of sEVs bound to laminin (Fig. 7 E, bottom; Fig. 7, G and H, bottom; Fig. 7 J). The number of sEVs attached to collagen type I was not changed by talin-1 KD or OE (Fig. 7, K and L). Additionally, sEVs from talin-1 KD BxPC3 cells (11% talin-1; Fig. 7 M) exhibited binding to laminin and collagen type I comparable with intact cell-derived sEVs (Fig. 7, N–P). These results suggest that talin-1 levels in sEVs are irrelevant to their ECM binding on glass.

Subsequently, we investigated whether talin-1 phosphorylation at Ser425 promotes the binding of integrin heterodimers in sEVs to ECM components. Western blotting revealed that talin-1 was highly phosphorylated at Ser425 in PC3 cells, but this phosphorylation was attenuated by treatment with roscovitine, an inhibitor of Cdk5 (Fig. 7 Q). Moreover, talin-1 at Ser425 was almost completely dephosphorylated in sEVs (Fig. 7 Q), suggesting that talin-1 phosphorylation at Ser425 does not play an important role in the binding of sEVs to ECM components. Furthermore, kindlin-2, a coactivator of integrins abundantly expressed in cancer cells (Montanez et al., 2008), was not detected in sEVs (Fig. 7 R). Combined with the finding from talin-1 KD/OE and Ser425 dephosphorylation experiments, these results indicate that talin-1 and kindlin-2 in sEVs do not increase the binding of integrin heterodimers in sEVs to ECM components. Moreover, the diminished binding of integrin heterodimers to fibronectin in EVs might stem from the absence of talin-1 and kindlin-2 functionality.

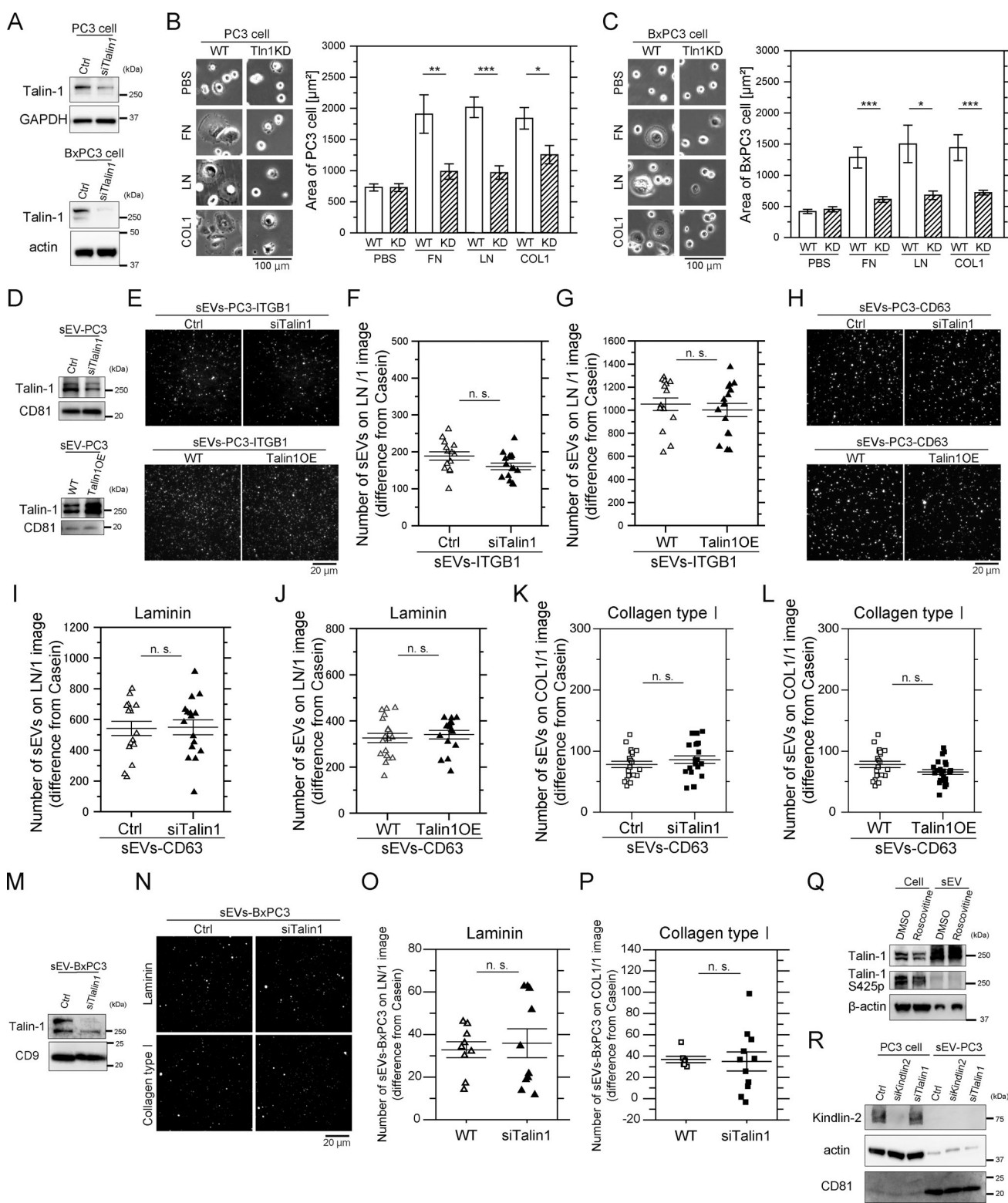

Figure 7. **Talin-1 in sEVs does not regulate the binding affinity of integrins for laminin. (A)** Western blot analysis of PC3 and BxPC3 cells after talin-1 KD by siRNA. **(B and C)** Cell spreading assay of WT and talin-1 (Tln1) KD PC3 cells (B) and BxPC3 cells (C) on glass coated with ECM components: fibronectin (FN), laminin (LN), or collagen typeI (COL1). Cells were observed after 2 h of incubation, and cell areas were quantified (n = 35 cells). **(D)** Western blot analysis of PC3 cell–derived sEVs after talin-1 KD by siRNA or overexpression of talin-1. **(E–G)** The fluorescence images (E) and the numbers of PC3-sEVs attached to glass coated with laminin before and after (F) talin-1 KD or (G) overexpression of talin-1 (n = 14 images). Halo7-integrin β1 in sEVs was labeled with SF650T. **(H–J)** The fluorescence images (H) and the numbers of CD63-labeled sEVs attached to glass coated with laminin before and after (I) talin-1 KD and (J) overexpression of talin-1 (n = 16 images). **(K and L)** The numbers of EV–CD63Halo7-SF650T particles attached to glass coated with collagen type I before and after (K) talin-1 KD or (L) overexpression of talin-1 (n = 21 images). **(M)** Western blot analysis of BxPC3 cell–derived sEVs after talin-1 KD by siRNA. **(N–P)** Fluorescence

images (N) and the numbers of sEVs-BxPC3 bound to laminin (O) or collagen type I (P) on glass before and after talin-1 KD (n = 7 images). The membranes of sEVs were stained with Exosparkler DeepRed. **(Q)** Western blot analysis of the phosphorylation of Ser425 on talin-1 in PC3 cells and sEVs. Roscovitine: an inhibitor of CDK5 that phosphorylates Ser425 of talin-1. **(R)** Western blot analysis of kindlin-2 in PC3 cells and PC3-derived sEVs before and after kindlin-2 KD and talin-1 KD. Data are presented as the mean ± SE. n.s., nonsignificant difference; *P < 0.05; **P < 0.01; ***P < 0.001 according to Welch's t test (two-sided). Source data are available for this figure: SourceData F7.

## Cholesterol attenuates the binding of sEVs to laminin and the PMs of recipient cells

Since talin-1 and kindlin-2 do not activate integrins in sEVs (Fig. 7), we sought to identify alternative factors that specifically maintain integrin binding to laminin even in sEVs where inside-out signaling is lacking. Previous reports showed that focal adhesions are highly ordered structures similar to rafts (Gaus et al., 2006) and that some integrin subunits, such as α6 and β4, undergo palmitoylation and partition into rafts (Gagnoux-Palacios et al., 2003; Yang et al., 2004). Moreover, a recent study demonstrated that sEV membranes are also raft enriched compared with parental cell PMs (Yasuda et al., 2022). Thus, we hypothesized that rafts enhance the binding of integrins in sEVs to laminin. To test this possibility, we used TIRFM to observe the binding of sEV–CD63-Halo7-SF650T to laminin on glass after depletion or addition of cholesterol. After cholesterol depletion by methyl-β-cyclodextrin (MβCD) or saponin, the number of sEVs bound to laminin on glass significantly increased by factors of 1.7 and 1.3, respectively (P < 0.01) (Fig. S6, A and B). Conversely, after cholesterol addition, the binding affinity was significantly reduced by ∼20% (P < 0.01) (Fig. S6 C). Furthermore, we also quantified the binding affinity of cholesterol-depleted and cholesterol-supplemented sEVs for MRC-5 PMs. In agreement with the findings on glass, cholesterol depletion from sEVs significantly increased the number of sEVs that bound to the recipient cell PM after 30 min of incubation (P = 0.039), whereas the addition of cholesterol markedly reduced it (P = 0.0059) (Fig. S6, D–F). These results, in contrast to our hypothesis, show that the presence of abundant cholesterol in sEV membranes impedes the binding of sEVs to both laminin and recipient cell PMs.

## CD151 preserves the binding of integrin heterodimers in sEVs to laminin

Since cholesterol inhibits the activity of laminin receptors in sEVs, we investigated the role of a membrane molecule that interacts with both cholesterol and integrin heterodimers in laminin binding in sEVs lacking inside-out signaling. CD151 is a ubiquitously expressed tetraspanin protein, and its large outer loop (Kazarov et al., 2002) specifically interacts with the C-terminal domain of the extracellular region of the integrin α7, α6, and α3 subunits, selectively strengthening the binding of integrin heterodimers such as α6β1 and α6β4 to laminin (Lammerding et al., 2003; Winterwood et al., 2006). CD151 contains a cholesterol-binding domain (Purushothaman and Thiruvenkatam, 2019), is palmitoylated (Yang et al., 2002), and interacts with many other membrane proteins via raft-lipid associations (Berditchevski, 2001; Charrin et al., 2003; Odintsova et al., 2006). We thus investigated whether CD151 preserves the binding activities of integrin heterodimers to

laminin in sEVs lacking inside-out signaling. After CD151 KD by siRNA (Fig. 8 A), integrin α6 and α3 levels in PC3 cells remained unchanged, yet cell spreading on laminin was significantly reduced, while spreading on fibronectin and collagen type I was unaffected (Fig. 8 B). CD151 KD also decreased the number of sEVs that bound to laminin by 41% (Fig. 8 C) but did not alter the binding affinity of sEVs to collagen type I (Fig. 8 E). Furthermore, CD151 KD attenuated the marked increase in the laminin-binding activity of sEVs caused by cholesterol depletion (Fig. 8, C and D). Meanwhile, a co-immunoprecipitation experiment showed that cholesterol depletion did not affect the interaction of integrin β1 with CD151, integrin α6, or α3 in sEVs (Fig. 8 F). These results suggest that cholesterol in sEVs suppresses sEV binding to laminin through a CD151-dependent mechanism, while it does not alter integrin heterodimer formation or the integrin–CD151 interaction. Therefore, integrin heterodimers that are not associated with cholesterol-dependent raft-like domains in sEVs might bind to laminin more effectively once raft-associated complexes are depleted.

## PC3-derived sEVs induce endothelial branching morphogenesis of human umbilical vein endothelial cells in a laminin-dependent manner

We next examined whether laminin- and integrin-mediated binding of sEVs to recipient cells plays a role in cellular responses. PCa-derived sEVs have been reported to promote an angiogenic phenotype in human umbilical vein endothelial cells (HUVECs) (Prigol et al., 2021; Wang et al., 2024). Thus, we investigated whether laminin-mediated binding of PC3-derived sEVs contributes to the morphological branching in HUVECs associated with angiogenesis (Myers et al., 2011). To determine whether laminin is essential for PC3-derived sEV binding to HUVECs, we knocked down laminin γ1 in HUVECs. As laminin γ1 is highly abundant among γ chain isoforms in HUVECs, according to the Human Protein Atlas (https://www.proteinatlas.org/), its KD should also reduce laminin α subunits (Fleger-Weckman et al., 2016). Indeed, western blotting confirmed that siRNA-mediated reduction of laminin γ1 to 13% (Fig. 9 A, top) resulted in a corresponding decrease in total laminin to 34% (Fig. 9 A, middle). The number of sEVs–CD63Halo7-TMR particles bound to both apical and basal membranes of HUVECs, quantified by confocal microscopy (Fig. 9 B), was dramatically decreased to 41% after laminin γ1 KD (Fig. 9 C). Similarly, the number of sEVs bound to the basal PM, observed by TIRFM, was markedly lower in laminin γ1 KD HUVECs than WT cells at prolonged incubation (>30 min) (Fig. 9 D). To assess morphogenesis, HUVECs were treated with sEVs for 12 h, and branched protrusions were analyzed using TIRFM. sEV-treated HUVECs exhibited significantly elongated protrusions compared with untreated cells (top-left and top-middle panels in Fig. 9 E).

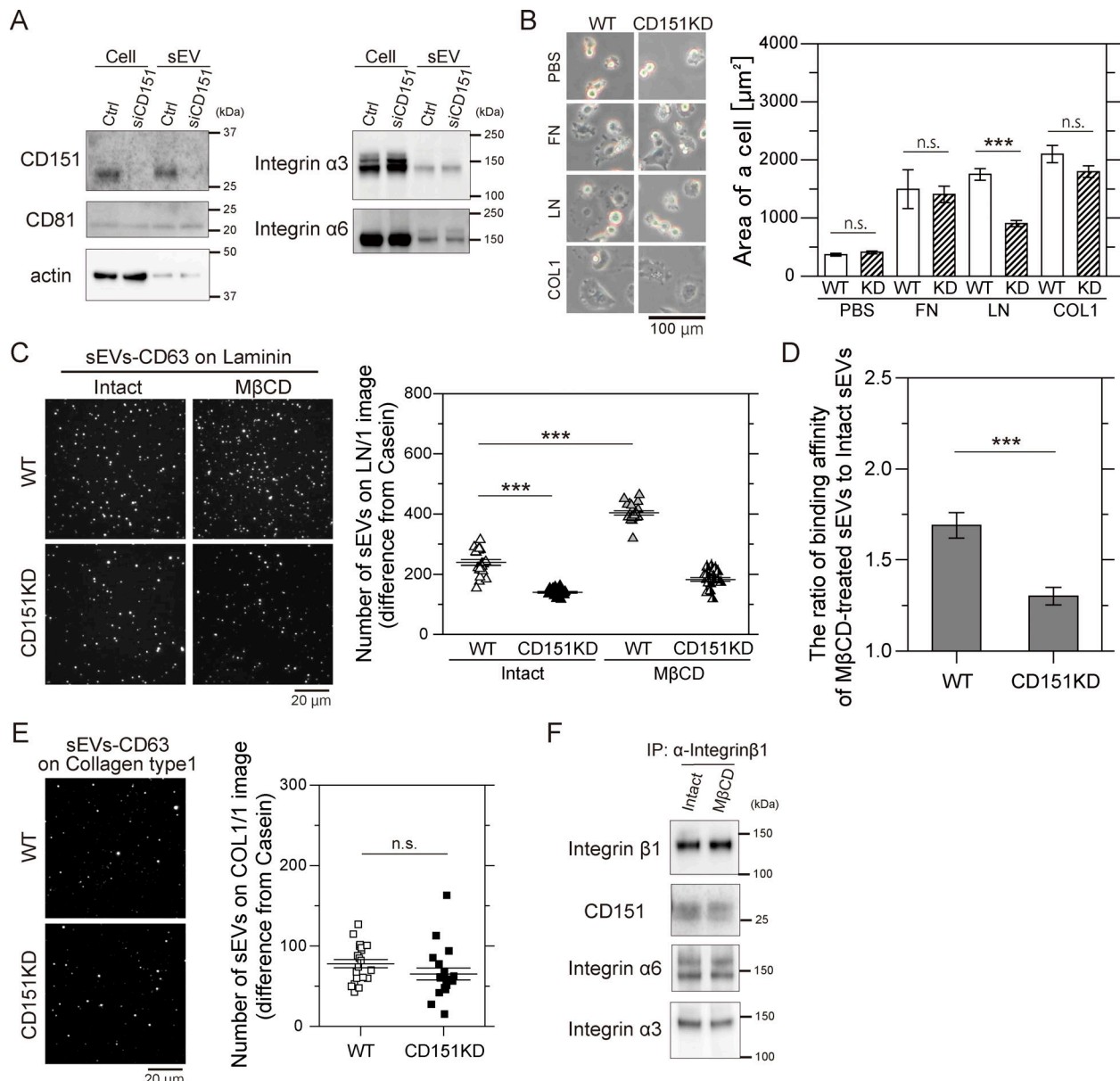

Figure 8. **CD151 and cholesterol regulate the binding affinity of sEVs for laminin. (A)** Western blot analysis of CD151 and integrin subunits in PC3 cells and sEVs after CD151 KD. The amount of cell proteins loaded in one lane was 2.5 times greater than that of the sEVs in the other lane. **(B)** Images of WT PC3 cells and CD151 KD cells on glass coated with ECM components (fibronectin [FN], laminin [LN], or collagen type I [COL1]) after 2 h of incubation. The areas of the cells were quantified ($n$ = 20 cells). **(C)** Single-particle fluorescence images of sEV–CD63Halo7-SF650T particles bound to laminin (LN) on glass before and after CD151 was knocked down and cholesterol was depleted by MβCD (left). The number of attached sEVs increased (right) ($n$ = 21 images). **(D)** The binding affinity ratio of cholesterol-depleted sEVs to intact sEVs was compared with that of CD151 KD sEVs. **(E)** Single-particle fluorescence images and the number of sEVs attached to collagen type I on glass before and after CD151 KD ($n$ = 19 images). **(F)** Western blot analysis of integrin–CD151 complex in PC3-derived sEVs before and after treatment with MβCD. The complex was immunoprecipitated with anti-integrin β1 and blotted with anti-integrin α6, α3, or CD151 antibodies. Data are presented as the mean ± SE. n.s., nonsignificant difference; ***P < 0.001 according to Welch's $t$ test (two-sided). In C, due to the necessity of multiple statistical tests, the significance level was corrected by the Bonferroni method and determined to be 0.025 (=0.05/2). Source data are available for this figure: SourceData F8.

Quantitative analysis of average protrusion lengths (yellow bars, Fig. 9 F) (Myers et al., 2011; Braun et al., 2014) in all examined cells and average total protrusion length per cell revealed that laminin γ1 KD significantly suppressed protrusion elongation (bottom-middle panel in Fig. 9, E and G–J), demonstrating that laminin-mediated sEV binding induces these morphological alterations. Interestingly, HUVEC stimulation with 0.26 nM VEGF,

an angiogenesis-inducing molecule, for 12 h produced protrusion lengths comparable with those observed after laminin γ1 KD (top-right and bottom-right in Fig. 9, E and G–J). These results indicate that while laminin is not intrinsically required for HUVEC branching morphogenesis, the binding of sEVs to laminin is essential for sEV-induced morphological changes in HUVECs.

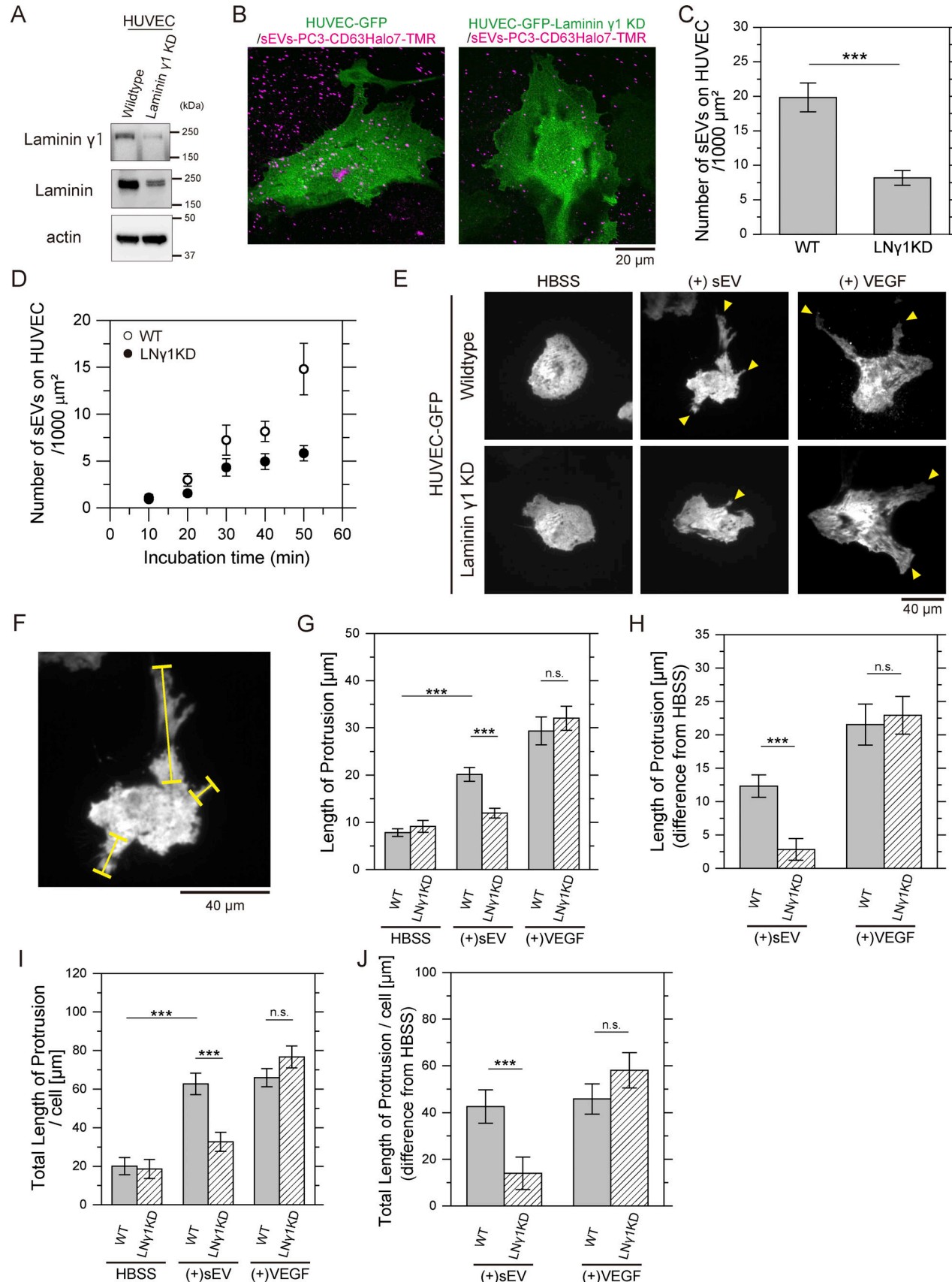

Figure 9. **Laminin-mediated binding of PC3-derived sEVs induces endothelial morphogenesis in HUVEC. (A)** Western blot analysis of laminin γ1 and total laminin levels in laminin γ1 KD HUVEC. **(B)** Fluorescence images of WT or laminin γ1 KD HUVECs expressing GFP and sEVs-PC3-CD63Halo7-TMR bound to the

cells, observed by confocal microscopy. Cells were fixed after 1-h incubation with sEVs. Since not all cells necessarily express GFP, many fluorescent spots of sEVs were observed either in regions containing non-expressing cells or potentially on the glass surface. Nevertheless, the number of sEVs bound to both the apical and basal surfaces of the cell PM can be quantified, as shown in Fig. 4. **(C)** Quantification of sEV particles bound to both apical and basal PM of WT or laminin γ1 KD HUVEC by confocal microscopy (*n* = 15 cells). **(D)** Time course analysis of the number of sEV particles bound to the basal PM of live WT or laminin γ1 KD HUVEC, monitored using TIRFM (*n* = 12 cells). **(E)** Fluorescence images of HUVEC-expressing GFP after 12 h of treatment with PC3-derived sEVs or 10 ng/ml VEGF. Yellow arrowheads show branched protrusions of HUVEC. **(F)** WT (+)sEV image from E showing protrusion lengths measured along yellow lines as indicated. **(G)** Average protrusion lengths across all examined cells before and after treatment with sEVs or VEGF. **(H)** Changes in average protrusion length after treatment with sEVs or VEGF, relative to untreated conditions (*n* = 16 cells). **(I)** Average total protrusion length per cell before and after treatment with sEVs or VEGF. **(J)** Variations in average total protrusion length per cell after treatment with sEVs or VEGF, relative to untreated conditions (*n* = 16 cells). Data are presented as mean ± SE. n.s., nonsignificant difference; *$P < 0.05$; **$P < 0.01$; ***$P < 0.001$ according to Welch's *t* test (two-sided). Source data are available for this figure: SourceData F9.

## Discussion

Our results demonstrated that laminin, not fibronectin, is the primary target of tumor-derived sEVs containing CD63, CD81, or CD9. sEV binding to laminin is mediated by integrin α6β1 and α6β4 heterodimers and GM1 in sEVs (Fig. 10 A). Tumor cell–derived mEVs and MVs also displayed higher binding affinities for laminin than fibronectin, despite containing receptors for both. While talin-1 and kindlin-2 do not enhance integrin binding activity in EVs, the tetraspanin CD151 considerably increased integrin-laminin binding, whereas cholesterol suppressed receptor function in a CD151-dependent manner. The binding of tumor-derived sEVs induced endothelial branching morphogenesis in HUVECs in a laminin-dependent manner.

Although inhibition of EV binding to recipient cells by integrin-targeting antibodies and ligands suggested that integrins in EVs mediate EV binding to recipient cells (Nazarenko et al., 2010; Carney et al., 2017; Zhang et al., 2022), direct evidence of the specific binding of integrins to ECM components has been lacking. The binding affinities of EVs for ECM components have not been quantitatively compared. Here, we developed an in vitro system to quantitatively measure the binding affinities of integrin heterodimers with ECM components, which enabled us to compare the binding affinities of EVs from different types of tumor cells with fibronectin, laminin, and collagen type I. The strong binding of all the sEVs, mEVs, and MVs to laminin via integrin heterodimers and the weak binding to fibronectin were unexpected because previous studies reported that the binding of EVs to recipient cells was inhibited by antibodies and inhibitors against fibronectin receptors (Huang et al., 2016; Altei et al., 2020). However, since our results do not indicate that tumor-derived EVs have no binding ability to fibronectin at all but rather that their binding to fibronectin is weaker than to laminin, our results are not inconsistent with previously reported findings. Since EV binding to PMs increases with EV concentration, differences in EV attachment after treatment with antibodies against integrin subunits of fibronectin receptors may have been detected in previous reports.

Our results showed that only GM1, which is ∼30-fold enriched in sEVs compared with their parental cells, bound to

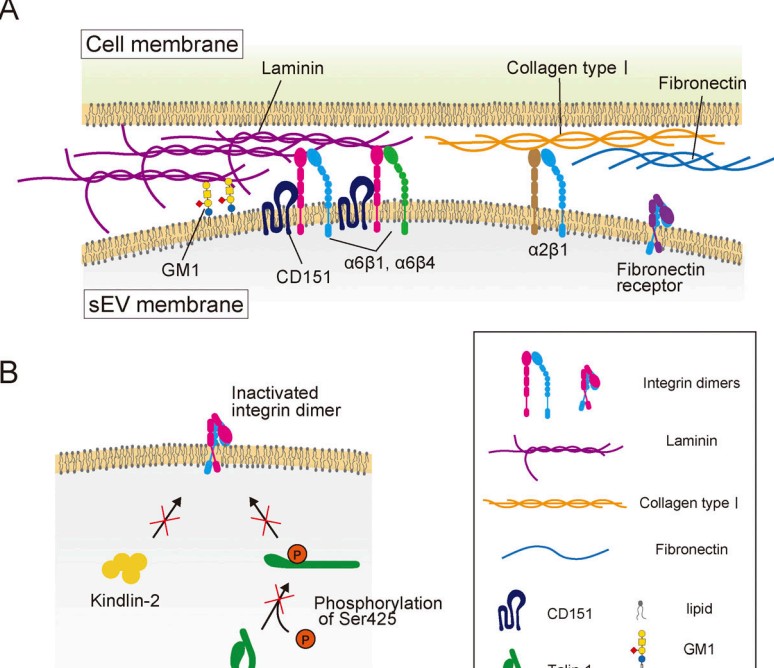

Figure 10. **Schematic model showing the binding of EVs to the recipient cell PM. (A)** Integrin heterodimers in EVs, α6β1/α6β4 and α2β1, can bind to laminin and collagen type I, respectively, but integrin heterodimers in EVs, such as α5β1, bind only weakly to fibronectin. CD151 enhances the binding of integrins α6β1 and α6β4 in sEVs to laminin. Cholesterol in sEVs suppresses laminin binding through a CD151-dependent mechanism. **(B)** Because Ser425 of talin-1 is hardly phosphorylated in EVs, talin-1 is inactive and is not involved in promoting the ECM-binding activity of integrin heterodimers. Kindlin-2 is also not involved in the activation of integrin heterodimers in EVs.

laminin on glass, while all the other examined gangliosides did not (Fig. S4). A competitive binding inhibition assay using the GM1 glycan moiety revealed that 18% and 61% of sEVs were bound to laminin through GM1–laminin and integrin β1–laminin interactions, respectively (Fig. S4 L). Meanwhile, none of the examined gangliosides, including GM1, bound to fibronectin. These results suggest that gangliosides may bind to laminin through specific glycan–protein interactions, not merely through electrostatic interactions. Since the binding of the glycan structure of GM1 to laminin does not require complicated regulation by other membrane molecules, it must constitute a robust binding system that is not prone to loss of activity unless GM1 is degraded.

Integrin heterodimers in cell membranes transition between bent-closed and extended-open conformations, maintaining the extended-open state by binding to ligands or activators such as talin (Moser et al., 2009; Li et al., 2024). Our study reveals that integrin β1 in sEVs adopts the extended-open conformation upon laminin binding (Fig. 5 C). Structural analysis revealed that integrin α6β1 interacts with laminin γ1's C-terminal tail via the β-propeller domain of integrin α6, with R155 being crucial for this interaction (Arimori et al., 2021). R155 mutations in integrin α6 markedly reduced sEV binding to laminin (Fig. 5 F), indicating the reliance on conventional integrin–ligand interactions. These findings suggest that integrin heterodimers in sEVs, like those in cells, bind to laminin through the extended-open conformation and that sEV integrin function may be modulated by established integrin-regulating molecules.

Neither talin-1 KD nor OE in sEVs significantly altered their binding to laminin (Fig. 7). Moreover, Ser425 of talin-1, an activating phosphorylation site (Jin et al., 2015), was almost completely dephosphorylated in sEVs, while phosphorylated in the cells that secreted sEVs. Talin-1 is ubiquitously expressed in all cell types in all tissues, whereas talin-2 is expressed mostly in the brain, heart muscle, and kidney (Monkley et al., 2001; Thul et al., 2017) and is absent in PC3 cells. Additionally, kindlin-2 was undetectable in sEVs (Fig. 7 R). These results indicate that inside-out integrin signaling is absent in sEVs, placing their integrins in a distinct environment from cell PMs. The lack of inside-out integrin signaling in sEVs seems reasonable because there is no cytoskeletal actin filament network inside the sEVs, which may be one of the reasons that integrin heterodimers of fibronectin receptors exhibit relatively low activity in sEVs (Fig. 10 B). On the other hand, our results show that α6β1 and α6β4 integrin heterodimers, laminin receptors in sEVs, remain active (Figs. 2, 4, 6, and S2), with CD151 preserving their activity (Fig. 8 C). CD151 forms a complex with integrins early in biosynthesis; this complex is delivered to tetraspanin-enriched microdomains in the cell PM (Stipp, 2010) and selectively strengthens integrin-mediated adhesion to laminin (Lammerding et al., 2003). CD151 contains a cholesterol-binding domain (Purushothaman and Thiruvenkatam, 2019) and links integrin heterodimers of laminin receptors with other tetraspanins by cholesterol- and raftophilic lipid–dependent mechanisms (Berditchevski et al., 2002; Franco et al., 2010). Our results revealed that cholesterol inhibited the binding of integrin heterodimers in sEVs to laminin in a CD151-dependent manner

(Fig. 8, C and D), suggesting that CD151-dependent activation of integrin heterodimers in sEVs may be attenuated by its association with lipid rafts. This finding aligns with the fact that CD151 depalmitoylation increases the downstream signaling of integrins in the cell PM (Berditchevski et al., 2002). Since PC3-derived sEV membranes possess a greater abundance of Lo (raft)-like phase than cell PMs (Yasuda et al., 2022), attenuation of CD151-dependent activation of integrin heterodimers may occur more extensively in sEVs.

The higher binding affinity of tumor-derived sEVs for laminin over fibronectin on recipient cell PMs may influence recipient cell responses. For example, embryonic neural crest cells (NCCs) exhibit ligand- and receptor-specific integrin regulation (Strachan and Condic, 2004). On laminin-coated glass, NCCs accumulate internalized laminin receptors (integrin α6) but not fibronectin receptors (integrin α5). Internalized laminin receptors colocalize with recycling vesicles and return to the cell surface, while fibronectin receptors do not. Blocking receptor recycling inhibited NCC motility on laminin, indicating that laminin receptor recycling is essential for rapid migration. Our recent study revealed that sEVs bound to laminin on recipient cells accumulate integrin subunits underneath, which are internalized via caveolae and clathrin-independent pathways (Hirosawa et al., 2025). Notably, 20–40% of PC3 cell–derived sEVs accumulate in recycling endosomes, suggesting that sEV-laminin binding promotes efficient integrin recycling.

Another example of a specific response to integrin-laminin binding involves signaling pathways distinct from those induced by fibronectin. Laminin-10/11 (same as laminin 511 used in this study) activates Rac via p130Cas-CrkII–DOCK180, enhancing cell migration, while fibronectin activates Rho via FAK, reducing cell migration (Gu et al., 2001). Since sEV binding to recipient cell PMs rapidly activates integrins, talin-1, and Src family kinases (Hirosawa et al., 2025), our results suggest laminin-specific signaling may occur during sEV binding.

Intriguingly, our results revealed that the binding of PC3-derived sEVs to HUVECs induced branching morphogenesis in a laminin-dependent manner, comparable with the level induced by VEGF in a laminin-independent manner (Fig. 9). As these morphological changes occurred 12 h after sEV treatment, they are likely driven by internalized sEVs rather than prompt sEV-induced signaling at the PM. A previous study suggested that PC3-derived EVs promote branching morphogenesis in HUVECs, potentially mediated by microRNA miR2–7a-3p (Prigol et al., 2021). For these cellular responses to occur, the microRNA must first be internalized into HUVECs and subsequently released from internalized EVs into the cytosol. Given that laminin KD significantly decreased the binding of sEVs to HUVECs, the amount of internalized microRNA available for cytosolic release is also likely diminished. Another critical factor contributing to the attenuated morphological changes after laminin KD may involve alterations in subcellular transport pathways after internalization, as previously discussed, potentially impairing the efficiency of microRNA release into the cytosol. The mechanisms underlying the enhanced cellular responses induced by laminin-bound EV internalization warrant further investigation.

In summary, our findings demonstrate that sEVs bind to laminin through robust mechanisms that do not require inside-out integrin signaling, leading to laminin-dependent morphological changes in HUVECs. The mechanism of EV binding to laminin revealed in this study will also provide useful information for applied research, including clinical applications.

## Materials and methods

### Materials

The human PCa cell line (PC3), human breast cancer cell line (SKBR3), human pancreas cancer cell line (BxPC-3), human fetal lung fibroblast line (MRC-5), and human marrow stromal cell line (HS-5) were purchased from the American Type Culture Collection (ATCC). The human breast cancer cell line (4175-LuT) was kindly provided by Dr. Joan Massagué and Dr. Hoshino (Hoshino et al., 2015). HUVECs were purchased from PromoCell.

All antibodies were obtained from commercial sources. The following primary antibodies were used for western blotting: mouse monoclonal anti-β actin 15G5A11/E2 (1:5,000, Cat# MA1–140; Thermo Fisher Scientific), mouse monoclonal anti-CD63 8A12 (1:1,000, Cat# SHI-EXO-M02; Cosmo Bio), mouse monoclonal anti-CD81 B11 (1:500, Cat# SC166029; Santa Cruz Biotechnology), rabbit monoclonal anti-CD9 EPR23105-125 (1:1,000, Cat# ab263019; Abcam), mouse monoclonal anti-CD151 11G5a (1:250, Cat# ab33315; Abcam), rabbit polyclonal anti-fibronectin (1:1,000, Cat# F3648; Sigma-Aldrich), rabbit monoclonal anti-GAPDH 14C10 (1:2,000, Cat# 2181S; Cell Signaling), rabbit monoclonal anti-integrin α2 EPR5788 (1:2,500, Cat# ab133557; Abcam), rabbit polyclonal anti-integrin α3 (1:500, Cat# ab190731; Abcam), rabbit polyclonal anti-integrin α5 (1:250, Cat# 4705S; Cell Signaling), rabbit monoclonal anti-integrin α5 clone EPR7854 (1:500, Cat# ab150361; Abcam), rabbit polyclonal anti-integrin α6 (1:500, Cat# 3750S; Cell Signaling), rabbit polyclonal anti-integrin α7 (1:500, Cat# ab203254; Abcam), rabbit polyclonal anti-integrin αV (1:500, Cat# ab117611; Abcam), mouse monoclonal anti-CD29 Clone 18 (1:2,000, Cat# 610467; BD Bioscience), rabbit polyclonal anti-integrin β3 (1:500, Cat# AB2984; Millipore), rabbit polyclonal anti-integrin β4 (1:500, Cat# 4707S; Cell Signaling), rabbit polyclonal anti-integrin β5 (1:500, Cat# 4708S; Cell Signaling), rabbit polyclonal anti-laminin 1+2 (1:1,000, Cat# ab7463; Abcam), mouse monoclonal anti-laminin clone 2E8 (1:500, Cat# MAB1920; Millipore), mouse monoclonal anti-talin-1 97H6 (1:500, Cat# GTX38972; Gene Tex), rabbit polyclonal anti-phospho-talin (Ser425), (1:250, Cat# TP4171; ECM Biosciences), mouse monoclonal anti–kindlin-2 clone 3A3 (1:500, Cat# MAB2617; Sigma-Aldrich), and anti-GM3 antibody clone GMR6 (1:1,000, Cat# A2582; Tokyo Chemical Industry), along with biotin-conjugated cholera toxin B subunit (CTXB; 1:500, Cat# 112; List Labs). Additionally, goat anti-mouse IgG-HRP (1:5,000, Cat# 12–349; Millipore), donkey anti-rabbit IgG-HRP (1:4,000, Cat# NA934; Cytiva), goat anti-rabbit IgG-HRP (1:10,000, Cat# A0545; Sigma-Aldrich), goat anti-mouse IgM antibody-HRP (1:10,000, Cat# 31440; Invitrogen), and rabbit anti-biotin antibody-HRP clone D5A7 (1:10,000, Cat# 5571; Cell Signaling) were used as secondary antibodies. For immunoprecipitation, mouse monoclonal IgG2a-negative control

C1.18.4 (Cat# MABF1080Z; Millipore), mouse monoclonal IgG1-κ–negative control MOPC-21 (Cat# MABF1081Z; Millipore), and rabbit polyclonal IgG-negative control (Cat# ab37415; Abcam) were used. For the observation of EVs and ECM components, mouse monoclonal anti-CD81 JS-81 (Cat# 555675; BD Pharmingen), mouse monoclonal anti-CD9 M-L13 (Cat# 555370; BD Pharmingen), rabbit polyclonal anti-collagen type 1 (Cat# ab34710; Abcam), rabbit polyclonal anti-fibronectin (Cat# ab23750; Abcam), rabbit polyclonal anti-laminin (Cat# L9393; Sigma-Aldrich), goat polyclonal anti-rabbit IgG-rhodamine (Cat# 55666; Cappel), and goat anti-rabbit IgG (Cat# 0212-0081; Cappel) were used.

We purchased the following purified ECM proteins: fibronectin from Roche (Cat# 11051407001), laminin from Sigma-Aldrich (Cat# L6274), and collagen type I from Sigma-Aldrich (Cat# C7774). For the experiments involving cholesterol and CD151, we used laminin-511 (Biolaminin 511 LN; BioLamina). The recombinant laminin (L6274; Sigma-Aldrich) we used was almost identical to Biolaminin 511 LN because nearly all the laminins in L6274 were laminin-511 (Wondimu et al., 2006).

### Cell culture

Mycoplasma contamination was not detected in any of the cell lines used in this study.

PC3 human PCa cells (CRL-1435; ATCC) were cultured in HAM F12 medium (Sigma-Aldrich) supplemented with 10% FBS (Gibco), 100 U/ml penicillin, and 100 µg/ml streptomycin (Gibco). Human fetal lung fibroblasts (MRC-5; CCL-171; ATCC) and human uterine cervix cancer cells (HeLa; CCL-2; ATCC) were cultured in minimum essential medium (Gibco) supplemented with 10% FBS, 100 U/ml penicillin, and 100 µg/ml streptomycin. Human marrow stromal cells (HS-5; CRL-11882; ATCC) and human breast cancer cells (4175-LuT; Hoshino et al., 2015) were cultured in Dulbecco's modified Eagle's medium (Sigma-Aldrich) supplemented with 10% FBS, 100 U/ml penicillin, and 100 µg/ml streptomycin. Human breast cancer cells (SKBR3; HTB-30; ATCC) were cultured in McCoy's 5a medium (Gibco) supplemented with 10% FBS, 100 U/ml penicillin, and 100 µg/ml streptomycin. Human pancreatic cancer cells (BxPC-3; CRL-1687; ATCC) were cultured in RPMI-1640 medium (Sigma-Aldrich) supplemented with 10% FBS, 100 U/ml penicillin, and 100 µg/ml streptomycin. HUVEC were cultured in endothelial cell growth medium (PromoCell).

The mouse melanoma cell line B78 (kindly provided by Dr. A. Albino at Memorial Sloan-Kettering Cancer Center, New York, NY, USA) was cultured in Dulbecco's modified Eagle's medium (Sigma-Aldrich) supplemented with 10% FBS, 100 U/ml penicillin, 100 µg/ml streptomycin, and 400 µg/ml G418. B78 cells expressed only GM3 (Fig. S4). pM2T1–1 cDNA transfection induced the overexpression of GM2/GD2 synthase, resulting in GM3 depletion because this enzyme uses GM3 as a substrate for synthesizing GM2 (Fig. S4). The transfection of B78 cells with pM1T-9 and pM2T1–1 induced the overexpression of GM1 (Fig. S4), resulting in GM3 and GM2 depletion because these enzymes use GM3 and GM2 as substrates for the synthesis of GM1. Transfection of B78 cells with pM2T1–1 and subsequently with pD3T31 induced overexpression of GD2, while transfection with

pD3T31 and subsequently with pM2T1–1 induced overexpression of both GD2 and GD3 (Fig. S4) (Yesmin et al., 2023).

## Flow cytometry of cells expressing gangliosides

The expression of gangliosides on the cell surface was analyzed by flow cytometry with a FACSCalibur (Becton-Dickison) as previously reported (Yamashiro et al., 1995; Yesmin et al., 2023). Intact and glycosyltransferase-transfected B78 cells were stained by incubation with monoclonal antibodies against GM3 (M2590 and IgM) (Cosmo Bio), GM2 (10–11 and IgM) (Natoli et al., 1986), GD3 (R24 and IgG3) (Vadhan-Raj et al., 1988) (10–11 and R24 antibodies kindly provided by P.O. Livingston and L.J. Old at Memorial Sloan-Kettering Cancer Center), GD2 (220-51 and IgG1), GD1a (D-266 and IgM), or GD1b (370, IgM); FITC-labeled goat anti-mouse IgG (H and L chain) antibodies (Cat# 55514; Cappel), which also recognize anti-mouse IgM. Anti-GD2, anti-GD1a, and anti-GD1b antibodies were generated by K. Furukawa (Zhao et al., 1999). Alternatively, GM1 on the cells was stained by applying biotinylated CTXB (Cat# 112; Sigma-Aldrich) and then FITC-labeled avidin (Cat# A2050; Sigma-Aldrich).

## Knockout of integrin subunits

The integrin β1, β4, α2, and α6 subunits were knocked out in PC3 cells via CRISPR-Cas9 gene editing. The single gRNA (sgRNA) target sequence was selected via the online tool CHOPCHOP (https://chopchop.cbu.uib.no/). The sgRNA target sequences were subsequently incorporated into the multicloning site of pSpCas(BB)-2A-Puro (Addgene) using BbsI-HF (New England Biolabs). The sgRNA target sequences used were 5′-TCATCACAT CGTGCAGAAGT-3′ and 5′-ATACAAGCAGGGCCAAATTG-3′ for the knockout of integrin β1; 5′-GTGCAGGTTCTCGTATCCCT-3′ and 5′-GCGCTCAGTCAAGGTAAGC-3′ for the knockout of integrin α2; 5′-CCGGGTACTTATAACTGGAA-3′ and 5′-ACACCG CCCAAAGATGTCTC-3′ for the knockout of integrin α6; and 5′-CTGCGAGATCAACTACTCGG-3′ for the knockout of integrin β4. PC3 cells were transfected with plasmids with a 4D-Nucleofector (LONZA). The transfected cells were selected and cloned using 1 µg/ml puromycin (final concentration) to achieve the knockout of integrin β1 or α2. For the knockout of integrin α6 or β4, the transfected cells were sorted by flow cytometry (BD, FACS-Melody) using a PE-conjugated anti-integrin α6 antibody (Invitrogen) or a PE-conjugated anti-integrin β4 antibody (BD). The knockout of each integrin subunit was validated by western blotting.

## Preparation of cDNA plasmids and transfection

WT PC3 cells and integrin KO PC3 cells were transfected with 1,500 ng of cDNA-encoding CD63-Halo7, CD81-Halo7, or CD9-Halo7 using a 4D-Nucleofector (LONZA) according to the manufacturer's recommendations. cDNA plasmids for human CD63 (NCBI accession no. NM_001780), human CD81 (NM_004356 and pFN21AE5228), and human CD9 (NM_001769 and pFN21AE1546) were purchased from Kazusa DNA Res. Inst. The DNA sequences encoding CD63, CD81, or CD9 were cloned and inserted into the pEGFP-N1 vector, and the EGFP sequence was replaced with the Halo7 sequence (Promega). The linker sequence 5′-ACCGGTGGTGGGCGCGCCTCTGGTGGCGGATCCGGG

GGT-3′ was incorporated between CD63, CD81, CD9, and Halo7. Cells stably expressing the molecules of interest were selected using a flow cytometer (BD, FACSMelody) and were cloned by treatment with 0.2 mg/ml (final concentration) G418 (Nacalai Tesque). To generate an expression vector for Halo7-integrin β1, cDNA-encoding Halo7-integrin β1 was cloned and inserted into the PiggyBac transposon vector. Halo7 was inserted into the hybrid domain of human integrin β1 (NM_002211) between residues Gly101 and Tyr102. The tagging of integrin β1 with Halo7 at this position is known to have no detrimental effects on integrin activity (Huet-Calderwood et al., 2017). WT or integrin β1 KO PC3 cells were cotransfected with cDNA-encoding Halo7 integrin β1 and transposase, yielding cell lines that exhibited stable expression of Halo7 integrin β1 (Yusa et al., 2009; Nagai et al., 2021). Cells exhibiting high expression of Halo7-integrin β1 were selected by flow cytometry. cDNA plasmids encoding integrin α6 and the dominant-negative integrin α6 R155A mutant were cloned and inserted into the PiggyBac transposon vector (Yusa et al., 2009). The cDNA plasmid for human integrin α6 (NM_000210.2 and pRK α6) was purchased from Addgene. The integrin α6 R155A cDNA was generated from the integrin α6 cDNA via PCR. Integrin α6 KO PC3 cells were transfected with cDNA encoding either integrin α6 or the α6 R155A mutant. Stable cell lines expressing these constructs were selected using 0.4 mg/ml zeocin and subsequently cloned.

4175-LuT, PC3, BxPC3, HeLa, and SKBR3 cells were transiently transfected with a plasmid-encoding CD63-Halo7 using polyethylenimine MAX (Polysciences) to isolate tumor-derived sEVs containing CD63.

To obtain PC3 cells OE human talin-1, PC3 cells were transfected with cDNA-encoding human talin-1 (NM_006289) using Lipofectamine 3000 reagent (Invitrogen) according to the manufacturer's recommendation. cDNA-encoding talin-1 was cloned and inserted into the pEGFP-N1 vector, after which the EGFP sequence was deleted. Moreover, talin-1 was knocked down by transfecting PC3 cells and BxPC3 cells with talin-1 siRNA (sc-36610; Santa Cruz Biotech). siRNA–A (sc-37007; Santa Cruz Biotech) was used as a negative control. CD151 or kindlin-2 was knocked down by transfecting PC3 cells with CD151 siRNA (D-003637-04 and D-003637-13; Dharmacon) or kindlin-2 siRNA (Sc-106786; Santa Cruz Biotech) using Lipofectamine 3000.

## Isolation of sEVs from cell culture supernatant

Cells that secreted sEVs were cultured either in two 100-mm dishes for observing sEVs on glass or in two 150-mm dishes for western blotting analysis and the observation of sEVs on cell PMs. The cells were grown to ∼80% confluence ($0.3–1.6 \times 10^8$ cells/150-mm dish). These cells were then transfected with plasmids or siRNAs and cultured for 1–2 days. Next, the cell culture medium in the 100- or 150-mm dishes was replaced with 10 or 30 ml of FBS-free medium, respectively. After 48 h of incubation, the cell culture supernatant was collected. For roscovitine treatment, cells were incubated with 20 µM roscovitine (186692-46-6; Santa Cruz Biotech) in FBS-free medium. The collected supernatant was then subjected to centrifugation at $300 \times g$ for 10 min at 4°C to remove cells. Subsequently, the

supernatant was further centrifuged at 2,000 × g and 4°C for 10 min to remove sediment and eliminate apoptotic bodies. Then, the supernatant was centrifuged at 10,000 × g and 4°C for 30 min (himac CF16RN, T9A31 angle rotor, and 50-ml Falcon tube) to pellet the MVs. Finally, the supernatant was concentrated by ultrafiltration using an Amicon Ultra 15 100K (Millipore) or a Centricon Plus 70 100K (Millipore) filter unit. The collected sEVs were incubated with 50–100 nM (final concentration) HaloTag TMR ligand (Promega) or HaloTag Sara-Fluor650T (SF650T) ligand (Goryo Chemical) for 1 h at 37°C. The sEVs were then pelleted by ultracentrifugation at 200,000 × g and 4°C for 4 h (himac CS100FNX with an S55A2 angle rotor and S308892A microtubes for ultracentrifugation). Alternatively, mEVs were pelleted by ultracentrifugation at 50,000 × g and 4°C for 30 min. The resulting pellet was resuspended in HBSS for microscopic observation. For western blotting, the pellet was suspended in RIPA buffer containing protease inhibitor cocktail set III (Millipore).

The ExoSparkler Exosome Membrane Labeling Kit-Deep Red (Dojindo) was also utilized as a fluorescent probe. MVs purified by lower-speed centrifugation (10,000 × g for 30 min) were successfully labeled according to the Exosparkler Technical Manual. Subsequently, the labeled MVs were purified by ultrafiltration using an Amicon Ultra 0.5 100K filter unit (Millipore).

**Partial cholesterol depletion/addition in sEVs**
sEVs were isolated from the cell culture supernatant of two 150-mm dishes using ultracentrifugation. To deplete cholesterol, the sEVs were incubated with 5 mM MβCD (Sigma-Aldrich) overnight at room temperature with gentle shaking or incubated with 200 µM saponin (Kanto Chemical) for 1 h on ice. These solutions were then concentrated by ultrafiltration using Amicon Ultra 2 ml 100K. The concentrated sEVs were incubated with 50–100 nM (final concentration) HaloTag SF650T ligand for 1 h at 37°C. Next, the sEVs were pelleted by ultracentrifugation at 200,000 × g and 4°C for 4 h. The pellet was resuspended in PBS or RIPA buffer. To add cholesterol, sEVs were incubated with 3.5 mM MβCD–cholesterol complex (1:1) for 2 h at 37°C, then isolated by the same method used in cholesterol depletion. The amount of cholesterol in these sEVs was measured by a LabAssay Cholesterol Kit (Fujifilm Wako Pure Chemical).

**Immunoprecipitation of sEVs**
The initial step involved treating 0.6 mg of Protein G–conjugated Dynabeads (Invitrogen) with 200 µl of 15 µg/ml anti-CD63 antibody (SHI-EXO-M02; Cosmo Bio), 8 µg/ml anti-integrin β1 (ab183666; Abcam), or mouse IgG2a isotype control (MABF1080Z; Millipore) in PBS containing 0.02% Tween 20. Alternatively, 0.6 mg of Protein A–conjugated Dynabeads (Invitrogen) was treated with 200 µl of 15 µg/ml anti-fibronectin antibody (F3648; Sigma-Aldrich) or rabbit IgG isotype control (ab172730; Abcam) in PBS containing 0.02% Tween 20. The mixture was incubated with rotation for 10 min at room temperature, and then the supernatant was removed. The beads were washed once with PBS containing 0.02% Tween 20 and

twice with 20 mM sodium phosphate (pH 7.0) and 150 mM NaCl using a magnetic rack. Then, the antibody-conjugated beads were resuspended in 5 mM BS3, 20 mM sodium phosphate (pH 7.0), and 150 mM NaCl for cross-linking between the antibody and the beads. After incubation with rotation for 10 min at room temperature, 12.5 µl of 1 M Tris-HCl (pH 7.5) was added to the solution, which was then incubated with rotation for an additional 15 min at room temperature. Subsequently, the supernatant was removed, and the beads were washed once with PBS containing 0.02% Tween 20 and then twice with PBS using a magnetic rack.

The antibody-conjugated beads were mixed with 300 µl of PC3-derived sEVs (0.1 mg/ml protein concentration) in PBS or 150 µl of PC3-derived sEVs' lysate (0.15 mg/ml protein concentration) in lysis buffer containing 0.5% Nonidet P40 substitute. After the mixture was incubated with rotation overnight, the supernatant was removed, and the beads were washed three times with PBS or lysis buffer. Then, the beads were resuspended in Laemmle's SDS sample buffer and incubated at 95°C for 5 min. The supernatant was collected using a magnetic rack and analyzed by western blotting.

**TEM of sEVs after negative staining**
A copper grid with a carbon-coated acetylcellulose film (EM Japan) was washed briefly (~1 min). 10 µl of water was placed on the grid and then blotted to eliminate the excess liquid. Next, 5 µl of the sEV sample was pipetted onto the grid and incubated for 1 min. The excess liquid was then removed by blotting. The grid was then briefly (~45 s) treated with 5 µl of 2% phosphor tungstic acid (TAAB Laboratory and Microscopy), followed by blotting to remove excess liquid. Next, the grid was air-dried at room temperature for 3 days in a desiccator containing silica gel desiccant and subsequently subjected to TEM observation.

sEV images were obtained using a single TEM instrument (JEM-2100F; JEOL) at 200 kV. These images were produced by computing the mean of 3-s acquisitions captured by a side-mounted CCD camera (Gatan) and processed by image solution software (Gatan Digital Micrography).

**Determination of sEV size by qNano**
The diameter of the suspended sEVs in HBSS was analyzed using a tunable resistive pulse sensing instrument, qNano (Izon Science) (Coumans et al., 2014), according to the manufacturer's protocol. Izon Control Suite 3.3 (Izon Science) was used for the data analysis. The measurements were performed with an NP100 pore (particle detection range: 40–320 nm) equipped with a stretch of 47.00 mm, a voltage of 1.2 V, and a pressure of 0.8 kPa; an NP150 pore with a stretch of 46.85 mm, a voltage of 1.2 V, and a pressure of 1.4 kPa; or an NP200 with a stretch of 48.00 mm, a voltage of 1.4 V, and a pressure of 1.4 kPa. The samples were calibrated using 70-, 95-, and 210-nm polystyrene calibration beads (CPC100; Izom Science).

**Western blotting analysis**
The quantities of integrin subunits were evaluated in WT and integrin-KO PC3 cells, as well as in sEVs derived from these cells, by western blotting. The cells and sEVs were suspended in RIPA

buffer containing protease inhibitor cocktail set III (Millipore). The protein concentrations of the samples were determined using a Thermo BCA protein assay kit (Thermo Fisher Scientific) and adjusted to 1 mg/ml for SDS-PAGE. Then, 5× concentrated Laemmli's SDS sample buffer was added to the samples, and the mixture was incubated at 95°C for 5 min in a blocking incubator. Next, 10 µl of this mixture was loaded into the lanes of a precast 4–12% gradient polyacrylamide gel (4–12% Bolt Bis-Tris Plus Gels; Thermo Fisher Scientific). Molecular weights were determined using Precision Plus Protein Prestained Standards (Bio-Rad Laboratories). After electrophoresis, the proteins were transferred onto a 0.45-µm polyvinylidene difluoride membrane (Millipore). Next, after blocking for 30 min at room temperature and washing, the membrane was incubated with the primary antibody in TBS (20 mM Tris and 150 mM NaCl, pH = 7.4) supplemented with 0.1% Tween20 (TBS-T) containing 5% nonfat milk or Blocking One (Nacalai Tesque) overnight at 4°C. After washing with TBS-T, the membranes were incubated with HRP-conjugated secondary antibody in TBS-T containing 5% nonfat milk or Blocking One solution at room temperature for 1 h. After washing with TBS-T, the membranes were treated with either the ECL start reagent (GE Healthcare) or the ECL select reagent (GE Healthcare) according to the manufacturer's guidelines. The chemiluminescent images of the membranes were acquired using FUSION-SOLO.7S (Vilber-Loumat) and analyzed using ImageJ.

## Dot blot analysis
Lysates of PC3 cells and PC3 cell–derived sEVs were prepared as described above. The phospholipid content of these lysates was quantified using a LabAssay Phospholipid (Fujifilm). Subsequently, 100 µl of these lysates (phospholipid concentration = 6.4 µM) was adsorbed to a nitrocellulose membrane using a Bio-Dot (Bio-Rad). After blocking with TBS-T containing 3% BSA for 30 min at room temperature, followed by washing, the membrane was incubated with biotin-conjugated CTXB for the detection of GM1 (1:500, Cat# 112; list labs) or with the primary IgM antibody against GM3 clone GMR6 (1:1,000, Cat# A2582; Tokyo Chemical Industry) in TBS-T containing 3% BSA overnight at 4°C. After washing with TBS-T, the membranes were incubated with the HRP-conjugated primary antibody against biotin (1:10,000, Cat# 5571; Cell Signaling) or the HRP-conjugated secondary antibody against IgM (1:10,000, Cat# 31440; Invitrogen) in TBS-T containing 3% BSA solution at room temperature for 1 h. After washing with TBS-T, the membranes were incubated with the ECL start reagent (GE Healthcare). Chemiluminescence images of the membranes were acquired using FUSION-SOLO.7S (Vilber-Loumat) and analyzed using ImageJ.

## Preparation of ganglioside-containing liposomes
The gangliosides GM1 (bovine brain), GD1a (bovine brain), and GD2 (human brain) were purchased from Avanti Polar Lipids, AdipoGen Life Sciences, and Santa Cruz, respectively. GM2, GM3, and GD3 were prepared by total chemical synthesis (Koikeda et al., 2019; Takahashi et al., 2020). In an eggplant-shaped flask, 50 µl of 1 mM ganglioside (GM1, GM2, GM3,

GD1a, GD2, or GD3) in methanol, 500 µl of 10 mM DMPC in chloroform, and 0.5 µl of 1 mM Bodipy-SM (Invitrogen) in methanol were mixed in a solution of 99:1:0.01 mol% ratio DMPC:ganglioside:Bodipy-SM. Since the molar ratio of GM1 to lipids in PC3 cells is known to be 0.0158 mol% (Llorente et al., 2013), and since the relative density of GM1 in the sEVs derived from PC3 was 29.7 ± 8.1 (mean ± SE, $n$ = 3) times greater than that in PC3 cells (Fig. S4 B). Since gangliosides are distributed in both the outer and inner leaflets of liposomal membranes, the density of GM1 in sEVs is similar to that in the outer leaflet of liposomes containing 0.94 mol% GM1. Therefore, we prepared liposomes containing 1 mol% ganglioside. The solvent was first dried with nitrogen gas and further dried with a CVE-3000 (EYELA) evaporator with VT-2000 (EYELA). Then, 5 ml of PBS was added to the flask, which was incubated at room temperature for 1 h. The solution was completely frozen in liquid nitrogen and then thawed in hot water (45°C). The freeze-thaw process was repeated five times. Using an extruder (LiposoFAST, AVESTIN), 500 µl of the vesicle suspension was passed through 100-nm pore-sized polycarbonate membrane filters (10 times). The amount of sialic acid in the liposomes was determined using a Sialic Acid Assay Kit (MAK314-1KT; Sigma-Aldrich), and the concentration of liposomes containing a ganglioside was adjusted to the same level for use in microscopic observation.

## Immunofluorescence imaging of the ECM
MRC-5 and HS-5 cells were cultured on glass-bottom dishes (IWAKI) for 2 days. The cells were then incubated with 10 µg/ml primary antibodies against fibronectin (F3648; Sigma-Aldrich), laminin (L9393; Sigma-Aldrich), collagen type I (ab34710; Abcam), laminin α1 (sc-74418; Santa-cruz biotech), or laminin α5 (ab17107; Abcam) in MEM at 37°C for 30 min. Next, the cells were washed twice with HBSS and incubated with 12 µg/ml rhodamine-conjugated anti-rabbit IgG antibody (55666; Cappel) or 10 µg/ml FITC-conjugated anti-mouse IgG antibody (55514; Cappell) in MEM at 37°C for 30 min. After washing twice with HBSS, fresh HBSS was added to the glass-based dishes. Subsequently, the cells were observed by epi-fluorescence microscopy (ECLIPSE Ts2-FL; Nikon, 60× 1.40 NA oil objective), which was performed with a CMOS camera (DS-Qi2; Nikon). A 560-nm LED lamp (Nikon) was used to excite rhodamine. The fluorescence intensities of laminin in the regions of cells (cell contours were determined using bright-field images) were quantified using ImageJ. The actual fluorescence intensities of laminin in the region of cells were measured by subtracting the average fluorescence intensity without the primary antibody from the fluorescence intensity of laminin stained with both the primary and secondary antibodies.

## Determination of the concentrations of sEVs containing fluorescently labeled marker proteins by TIRFM
The concentrations of protein and lipids in the sEV suspension (on the order of nM) were too low to determine very quantitatively via conventional spectrophotometry. Therefore, we employed single-fluorescent particle tracking to directly quantify the number of sEVs. In brief, the glass windows of single-well or triple-well glass-based dishes (IWAKI) were coated with 100 or

50 µl of 10 µg/ml anti-CD63 IgG antibody (8A12; Cosmo Bio), anti-CD81 IgG antibody (JS-81; BD Biosciences), or anti-CD9 antibody (M-L13; BD Biosciences) in HBSS and then incubated for 2 h at 37°C. Subsequently, the antibody solution was removed, and the glass window was coated with either 100 or 50 µl of 50 µg/ml casein (Sigma-Aldrich) in HBSS for 1 h at 37°C. This coating process was essential for mitigating the nonspecific binding of sEVs to the glass surface during subsequent experiments.

Either 100 or 50 µl of three distinct concentrations of sEVs containing CD63-Halo7, CD81-Halo7, and CD9-Halo7, all conjugated with SF650T in HBSS, was applied to glass windows precoated with antibodies against CD63, CD81, and CD9, respectively. The samples were then incubated at 37°C for 1 h. After removing the sEVs and washing twice with HBSS, the individual fluorescent particles of the sEVs were observed at 37°C with single-molecule detection sensitivity by TIRFM using an Olympus IX-83 microscope (60× 1.49 NA oil objective) equipped with a high-speed gated image intensifier (C9016-02MLG; Hamamatsu Photonics) coupled to an sCMOS camera (ORCA-Flash4.0 V2; Hamamatsu Photonics), as previously described (Komura et al., 2016; Kinoshita et al., 2017; Morise et al., 2019). SF650T was excited using a 647-nm laser (LuxXPlus647-140, 140 mW, Omicron Laserrange) at an intensity of 0.3 µW/µm².

All the acquired sEV movies were analyzed using ImageJ (Fiji). First, a noise reduction process was carried out, wherein 10 frames of sEV images observed at 30 frames/s were averaged. Subsequently, each pixel was replaced by the average of the 3 × 3 neighborhood pixels using a smooth filter. Single particles of sEVs whose fluorescence intensity exceeded that of free SF650T were identified, and the numbers of sEVs were counted. By performing these observations at three distinct sEV concentrations, we obtained a calibration curve. Thereafter, sEVs derived from both intact cells and integrin subunit KO cells were prepared at identical concentrations (0.8–4 × 10¹⁰ particles/ml).

### Determination of the number of sEVs bound to ECM components on glass

To prevent nonspecific binding of sEVs to the glass surface, 50 or 100 µl of 50 µg/ml casein solution in HBSS was added to glass windows in single-well or triple-well glass-based dishes (IWAKI) and incubated for 1 h at 37°C. Then, the casein solution was replaced with 50 or 100 µl of 100 µg/ml fibronectin (Roche), 50 µg/ml laminin (Sigma-Aldrich), 20 µg/ml laminin-511 (Bio-Lamina), 100 µg/ml collagen type I (Sigma-Aldrich), or 50 µg/ml casein (Sigma-Aldrich) for 2 h at 37°C. After removal of the ECM or casein solution, 50 or 100 µl of sEVs containing CD63/CD81/CD9-Halo7 labeled with SF650T at the same concentration were applied to the glass and incubated for 1 h at 37°C. To assess the relative contributions of integrin and GM1 to sEV binding to laminin, we performed a competitive inhibition assay using the glycan moiety of GM1, synthesized as previously reported (Yanaka et al., 2019). sEVs were applied to laminin-coated glass in the presence of a high concentration (0.5 mM final) of the GM1 glycan moiety. After removal of the sEV suspension and two washes with HBSS, individual fluorescent particles of the sEVs in HBSS were visualized at 37°C with single-molecule

detection sensitivity by TIRFM. The number of sEVs bound to the ECM or casein on glass was determined using ImageJ (Fiji) as described above. The number of sEVs bound to ECM-coated glass was estimated by subtracting the number of sEVs bound to casein-coated glass in a 1,024 pixel × 1,024 pixel (81.9 × 81.9 µm) image.

### Determination of the number of sEVs bound to cell membranes

sEVs containing CD63-Halo7 labeled with SF650T (sEV–CD63Halo7-SF650T) were incubated with MRC-5 or HS-5 cells expressing mCherry or GFP in the cytosol at 37°C. sEV–CD63Halo7-SF650T was visualized with single-molecule detection sensitivity by TIRFM, and mCherry or GFP was observed by oblique illumination. The number of sEVs on the cell membrane (the cell contour was determined using binarized mCherry or GFP images) was quantified every 10 min by ImageJ (Fiji) as described above.

sEVs bound to cell membranes were visualized using confocal fluorescence microscopy. sEVs–CD63Halo7-TMR were incubated with MRC-5 or HS-5 cells expressing GFP at 37°C for 1 h. After incubation, the cells were fixed with 4% paraformaldehyde at room temperature for 30 min. After removing the paraformaldehyde solution, the cells were treated with 0.1 M glycine for 5 min to quench residual fixation. The samples were then washed with HBSS, and an aqueous mounting medium (TA-030-FM; Thermo Fisher Scientific) was applied to the sample for imaging. sEVs–CD63Halo7-TMR and GFP within the cells were observed at room temperature using an FV-1000 confocal microscope (Olympus; 60× 1.49 NA oil objective). Imaging was performed with an extended detector exposure time. Fluorescent images were acquired using the FV10MP-SU-TI scanning unit (Olympus) and FV10-ASW4.2 acquisition software (Olympus). The number of sEVs associated with cell membranes was quantified as described above.

### Observation of sEVs stained with antibodies targeting activated integrin β1

sEVs containing CD63-Halo7 labeled with TMR (sEV–CD63Halo7-TMR) were incubated on either uncoated glass or laminin-511–coated glass for 1 h. After incubation, the samples were washed with HBSS, and 10 µg/ml of general anti-integrin β1 antibody (clone P5D2 from Abcam) or antibodies specific for activated integrin β1 (clone HUTS-4 from Millipore and clone HUTS-21 from BD Pharmingen) (Luque et al., 1996) was added and incubated at 37°C for 30 min. After two washes with HBSS, 1 µg/ml AlexaFlour488-conjugated anti-mouse IgG antibody was applied and incubated at 37°C for 30 min. The glass was then washed five times with HBSS. Single-fluorescent particles of sEV–CD63Halo7-TMR and AlexaFlour488-conjugated antibody on uncoated glass or laminin 511-coated glass were simultaneously observed by TIRFM. Colocalization between sEV–CD63Halo7-TMR and anti-integrin β1 antibody was analyzed using ImageJ (Fiji). The brightest point of each sEV–CD63Halo7-TMR signal was identified as the center of the sEV. The fluorescence intensity of AlexaFlour488 within a circular region with a radius of 3 pixels, centered on the detected

coordinates of the sEV, was quantified. Instances where the AlexaFlour488 intensity exceeded the signal of a single Alexa-Flour488 molecule were recorded as colocalization events. Finally, the percentage of colocalization events relative to the total number of sEVs was calculated, representing the ratio of sEVs stained with anti-integrin β1 antibody.

## Simultaneous dual-color observation of dSTORM movie of ECM components and single-fluorescent particles of sEVs on living cell PMs

For immunostaining of the ECM components for dSTORM movie observation, 0.67 µl of 5.6 µg/ml SaraFluor650B (SF650B; Goryo Chemical) NHS ester was incubated with 100 µl of 1 mg/ml anti-rabbit IgG antibody (0212-0081; Cappel) in 0.1 M NaHCO$_3$ for 60 min at room temperature. SF650B-conjugated anti-rabbit IgG was then isolated using a Sephadex G-25 column (GE Healthcare). Since SF650B spontaneously blinks, it enables the observation of dSTORM movie observations in living-cell PMs. MRC-5 cells cultured on glass-based dishes for 2 days were subsequently incubated with 10 µg/ml anti-fibronectin IgG (F3648; Sigma-Aldrich), anti-laminin IgG (L9393; Sigma-Aldrich), or anti-collagen type I IgG (ab34710; Abcam) for 30 min at 37°C. After washing twice with HBSS, the cells were incubated with 10 µg/ml SF650B-conjugated anti-rabbit IgG antibody for 30 min at 37°C. After washing with HBSS three times, sEVs containing CD63–Halo7-TMR were incubated with the cells for another 30 min at 37°C. Individual fluorescent particles of the sEVs and single molecules of SF650B on the apical surface of the cells at 512 × 512 pixel (25.6 µ × 25.6 µm) were observed at 5-ms resolution (200 frames/s) for 3504 frames by oblique-angle illumination using an Olympus IX-83 inverted microscope (100× 1.5 NA oil objective) equipped with two high-speed gated image intensifiers (C9016-02MLG; Hamamatsu Photonics) coupled to two sCMOS cameras (ORCA-Flash4.0 V2; Hamamatsu Photonics), as described previously (Komura et al., 2016; Kinoshita et al., 2017; Morise et al., 2019; Kemmoku et al., 2024). TMR and SF650B were excited using a 561-nm laser (Excelsior-561-100-CDRW, 100 mW, Spectra-Physics) and a 647-nm laser (140 mW; Omicron Laserrange) at 2.8 and 16 µW/µm², respectively. The final magnification was 133×, yielding pixel sizes of 47.1 nm (square pixels). To obtain a single dSTORM image, data acquisition was performed for 1,002 frames and then repeated by shifting the initial frames backward by 6 frames, resulting in a total of 417 dSTORM images (Fig. 6 A). By connecting these dSTORM image sequences, we generated a pseudo real-time "dSTORM movie." Images of individual sEV particles were concurrently recorded at 200 frames/s, and images averaged over 6 frames were subsequently combined to produce the movie. The pseudo real-time movies of ECM components and sEV particles (33 frames/s) were superimposed (Fig. 6 A).

More details regarding the data acquisition process for dSTORM imaging and the generation of the dSTORM movie are provided below. The dSTORM super-resolution video data were generated using frame information and the x and y coordinates of all the spots, which were incorporated into the CSV data output by the ThunderSTORM plugin for ImageJ (Ovesný et al., 2014) installed in the Fiji package (Schindelin et al., 2012).

dSTORM image reconstructions were conducted with a pixel size of 10 nm. Gaussian rendering with a localization precision of 24 nm was utilized in this process. Furthermore, we used the uncertainty that represents the spatial precision of the spot. A Gaussian distribution was generated for each spot, with the x and y coordinates serving as the center and the SD being equivalent to six times the uncertainty of the spot. This distribution indicates the existence probability of the spot. The existence probability distribution for all structures at time t was obtained as a summation of all the distributions for spots that appeared in [6t+1 6t+1002] frames. The dSTORM super-resolution video data were generated by repeating this process for time t = 0, 1, 2, ... If the value of each time and coordinate in the dSTORM video data exceeded a threshold value θ, we considered a structure to exist at the time and coordinate. The optimal threshold value θ was determined for each dSTORM video data point to minimize the average in-class variance for the two classes when every value in the dSTORM video data was classified into two classes using the threshold value θ (Chen et al., 2014; Kemmoku et al., 2024). To synchronize the dSTORM movie of ECM structures with the single-particle movie of sEVs, the temporal resolution of the single-particle movie was converted to 33.3 frames/s by averaging 6 frames, and the averaged single-particle image was merged with the dSTORM image created from 1,002 frames, of which the middle frame was synchronized with the averaged single-particle image (Fig. 6 A). Single-particle tracking in the movie was performed by in-house computer software based on previously reported methods (Gelles et al., 1988; Kusumi et al., 1993). More details have been described previously (Suzuki et al., 2007a, 2007b, 2012; Kasai et al., 2011; Komura et al., 2016; Kinoshita et al., 2017; Morise et al., 2019; Fujiwara et al., 2023a, 2023b).

## Colocalization analysis between ECM structures in dSTORM movies and single sEV particles

The distances between the centroids of the sEVs and the boundaries of the ECM structures were quantified using single-particle tracking data and binarized dSTORM movies. To generate binarized images of the ECM structures in the dSTORM movie, we employed the kernel density estimation (KDE) method (Kemmoku et al., 2024). KDE is a traditional image segmentation method that can rapidly determine the criterion for segmenting PALM and dSTORM images, as reported previously (Chen et al., 2014; Kemmoku et al., 2024). Furthermore, a recent study demonstrated that KDE is one of the most appropriate methods for segmenting images of clearly outlined structures, such as protein clusters (Nieves et al., 2023). KDE provides a way to interpolate object boundaries without bias by using random fluctuations of activated boundary fluorophores.

The ThunderSTORM datasets were imported into in-house KDE software (MATLAB) for image segmentation and colocalization analysis. We used not only the localization coordinates but also the uncertainty, which reflects the spatial detection precision of the spots. Let $x_i$, $y_i$ and $u_i$, i = 1,2, . . . ,$N$ be the horizontal and vertical localization coordinates and the uncertainty of spot i, respectively. $N$ represents the total number of spots. Considering the uncertainty of the location measurement

of the spot $i$, the existence probability $p_i(x,y)$ of the spot $i$ is considered to spread to the Gaussian distribution with the center coordinate $x_i$, $y_i$ and the SD that is proportional to the uncertainty $u_i$ as follows:

$$p_i(x,y) = \frac{1}{2\pi(Au_i)^2} e^{-\frac{(x-x_i)^2+(y-y_i)^2}{2(Au_i)^2}},$$

in which $A$ is the proportional coefficient of SD to the uncertainty $u_i$. This coefficient was estimated as $A \approx 6$ by some preliminary experiments. The Gaussian distribution for constructing each existence probability $p_i(x,y)$ is called the Gaussian kernel. The existence probability distribution for all spots, i.e., the dSTORM image, is obtained by averaging the existence probability distribution $p_i(x,y)$ as follows:

$$p(x,y) = \alpha \frac{1}{N} \sum_{i=1}^{N} p_i(x,y) + (1-\alpha)p_{bg}(x,y),$$

excluding spots having extremely small or large uncertainties $u_i$ because such spots are likely to contain noise (e.g., electrical shot noise from the camera). $p_{bg}(x,y)$ represents the background noise generated by the probability density distribution $f(v) = \frac{1}{2\pi\sigma_{bg}^2} e^{-\frac{(v-\mu_{bg})^2}{2\sigma_{bg}^2}}$, independent of $x,y$. $\alpha$ is a weight determined by the respective occurrence probabilities of the spots and background noise.

The structures can be considered to exist where the dSTORM image, which represents the probability of structure existence, is above a certain threshold value $\theta$ because the spots appear uniformly inside the structures with a certain probability. It is necessary to determine the optimal threshold value $\theta$ for each obtained dSTORM image because the optimal value differs depending on the target molecule and the experimental environment. In general, the histogram of $p(x,y)$ values for all $x,y$ forms a bimodal shape consisting of both larger values created by dense spots appearing in structures and smaller values created by sporadic noise. Therefore, the threshold value $\theta$ should be determined to separate these two structures. Otsu's method is a well-known approach for determining such a threshold value (Otsu, 1979). Let $S_{in}$ and $S_{out}$ be sets of coordinates $(x,y)$ where $p(x,y) \geq \theta$ and $p(x,y) < \theta$, respectively. The sets $S_{in}$ and $S_{out}$ indicate the inside and outside of the clusters, respectively. The numbers of elements in sets $S_{in}$ and $S_{out}$ are represented by $N_{in}$ and $N_{out}$, respectively. The intraclass variances of the sets $S_{in}$ and $S_{out}$ are calculated by

$$\sigma^2[S_{in}] = \frac{1}{N_{in}} \sum_{(x,y) \in S_{in}} (p(x,y) - \mu[S_{in}])^2,$$

$$\sigma^2[S_{out}] = \frac{1}{N_{out}} \sum_{(x,y) \in S_{out}} (p(x,y) - \mu[S_{out}])^2.$$

In the above equations, $\mu[S_{in}]$ and $\mu[S_{out}]$ are the respective averages of the set $S_{in}$ and $S_{out}$ obtained by

$$\mu[S_{in}] = \frac{1}{N_{in}} \sum_{(x,y) \in S_{in}} p(x,y),$$

$$\mu[S_{out}] = \frac{1}{N_{in}} \sum_{(x,y) \in S_{out}} p(x,y).$$

The average of these intraclass variances is

$$\sigma^2[S_{in}, S_{out}] = \frac{N_{in}\sigma^2[S_{in}] + N_{out}\sigma^2[S_{out}]}{N_{in} + N_{out}}.$$

In Otsu's method, the optimal threshold value $\hat{\theta}$ to minimize the average of the intraclass variances $\sigma^2[S_{in},S_{out}]$ is determined as follows:

$$\hat{\theta} = \underset{\theta}{argmin}\,\sigma^2[S_{in}, S_{out}].$$

In this study, the structures were determined using Otsu's method.

Thereafter, the distance from the contour of the determined structures to the center of the sEVs on the PM was measured in all the frames of the superimposed movies of the ECM components and sEVs, and the number densities of pairwise distances at a given distance were plotted. If an sEV particle was within an ECM structure, the measured distance was expressed as a negative value. Furthermore, we simulated the distances between random coordinates and the boundaries of ECM structures in silico and calculated their corresponding relative frequency distributions using a bin width of 50 nm. To normalize the relative frequency of sEVs within each bin, we divided the data by the corresponding relative frequency of random spots. A ratio exceeding 1 indicated a higher incidence of sEVs at the measured distance.

## dSTORM imaging of ECM-coated glass
The ECM molecules were coated on triple-well glass-bottom dishes and incubated with 5 µg/ml anti-fibronectin IgG (F3648; Sigma-Aldrich), anti-laminin IgG (L9393; Sigma-Aldrich), or anti-collagen type I IgG (ab34710; Abcam) for 2 h at room temperature. After washing twice with HBSS, the ECM molecules were incubated with 2.5 µg/ml SF650B-conjugated anti-rabbit IgG antibodies for 1 h at room temperature. After washing with HBSS five times, single molecules of SF650B at $512 \times 512$ pixels ($25.6 \times 25.6$ µm) were observed via TIRFM at 5-ms resolution (200 frames/s) for 1,008 frames. The dSTORM images were reconstructed from the movie using the ThunderSTORM plugin for ImageJ.

## Cell spreading assay
The cells that secreted sEVs were cultured for 3–5 days and subsequently collected using a solution of 0.5% BSA and 5 mM EDTA in PBS. A total of $4 \times 10^5$ cells/ml were then evenly dispersed onto an ECM-coated single-well glass-bottom dish. The cells were observed after incubation for 0, 30, 60, or 120 min with an epi-illumination microscope (CKX53; Olympus, 20× 0.40 NA objective) equipped with a camera (WAT-01U2; WA-TEC). The cell area was quantified using ImageJ software.

## Observation of branched protrusions in HUVEC after sEV treatment
HUVECs were transfected with a plasmid-encoding GFP and siRNA-targeting laminin γ1 (SIO2757475; Qiagen). After 3 days of

culture, the transfected HUVECs were seeded onto fibronectin-coated glass-bottom dishes and cultured for an additional day. Subsequently, the cells were incubated with PC3-derived sEVs or 0.26 nM (10 ng/ml) VEGF (Peprotech) for 12 h. The HUVEC-GFP and HUVEC-GFP–LNγ1KD were then visualized using TIRFM. The position of the greater curvature on the HUVEC membrane was designated as the origin of the branched protrusion. The length of the protrusion, defined as the distance between the branched protrusion origin and the tip of the protrusion (Fig. 9 F), was measured following the previously reported protocol (Myers et al., 2011; Braun et al., 2014).

### Online supplemental material

Fig. S1 shows the preparation of sEVs derived from PC3 cells and the determination of their size and concentration. Fig. S2 shows the number of sEV particles attached to glass coated with ECM components. sEVs containing CD81 or CD9-Halo7 were derived from WT, integrin β1, integrin α2, integrin α6, or integrin β4 KO PC3 cells and labeled with SF650T. Fig. S3 shows the results of western blot analysis to determine whether sEVs derived from PC3 cells were covered by fibronectin and/or laminin. Fig. S4 shows the number of sEVs derived from B78 cells expressing several glycosyltransferases and the number of liposomes containing gangliosides bound to laminin. Fig. S5 shows the flow cytometry analysis of B78 cells expressing several glycosyltransferases to examine the expression levels of gangliosides. Fig. S6 shows the number of PC3-derived sEVs bound to laminin on glass and MRC-5 PMs before and after cholesterol depletion or addition. Table S1 presents the numbers of sEVs labeled with tetraspanin–Halo7-TMR attached to glass coated with ECM components. Table S1 is related to Figs. 3 and S2. Videos 3, 4, and 1 (Video 3: fibronectin, Video 4: collagen type I, and Video 1: laminin) show the simultaneous observation of single sEV–CD63Halo7-TMR particles (green) and dSTORM ECM movies (magenta) on an MRC cell. Video 2 shows an enlarged movie of the colocalization of an sEV–CD63Halo7-TMR particle (green) and laminin (dSTORM movie, magenta) on an MRC cell.

### Data availability

The data supporting the findings of this study are available from the corresponding author upon reasonable request.

## Acknowledgments

We thank Joan Massagué (Memorial Sloan-Kettering Cancer Center) and Ayuko Hoshino (University of Tokyo, Tokyo, Japan) for kindly providing the human 4175-LuT cells. We also thank Yumi Matsuno for supporting the preparation of EVs, Shinobu Kawaguchi for constructing various cDNAs, and Takahiro Fujiwara and Akihiro Kusumi for developing the analysis software for single-molecule imaging.

This work was supported in part by Japan Science and Technology Agency (JST) grants from the Core Research for Evolutional Science and Technology program in the field of "Extracellular Fine Particles" (JPMJCR18H2) (K.G.N. Suzuki, K.M. Hirosawa, and H. Ando) and "Cell Control" (JPMJCR24B3) (K.G.N. Suzuki), Grants-in-Aid for Scientific Research from the Japan Society for the Promotion of Science (JSPS) (24K01974 and 24K21944) (K.G.N. Suzuki), a Grant-in-Aid for Core-to-Core Program from JSPS (JSCCA202000007) (K.G.N. Suzuki and H. Ando), the National Cancer Center Research and Development Fund (2023-A-03) (K.G.N. Suzuki), the Japan Agency for Medical Research and Development (AMED) (JP21km0908001 and JP24ym0126134) (K.G.N. Suzuki), and the Takeda Science Foundation (K.G.N. Suzuki), as well as a Grant-in-Aid from Support for Pioneering Research Initiated by the Next Generation in JST (JST SPRING) (JPMJSP2125) (T. Isogai) and a Grant-in-Aid for JSPS Fellows (23KJ1045) (T. Isogai).

Author contributions: T. Isogai: conceptualization, funding acquisition, investigation, and writing—original draft, review, and editing. K.M. Hirosawa: conceptualization, data curation, methodology, project administration, software, supervision, and writing—review and editing. M. Kanno: investigation. A. Sho: conceptualization, data curation, formal analysis, funding acquisition, investigation, methodology, project administration, resources, software, supervision, validation, visualization, and writing—original draft, review, and editing. R.S. Kasai: methodology and resources. N. Komura: investigation. H. Ando: formal analysis, investigation, methodology, and writing—original draft, review, and editing. K. Furukawa: investigation and resources. Y. Ohmi: investigation and resources. K. Furukawa: investigation and resources. Y. Yokota: formal analysis, methodology, software, and writing—original draft, review, and editing. K.G.N. Suzuki: conceptualization, data curation, funding acquisition, methodology, project administration, software, supervision, visualization, and writing—original draft, review, and editing.

Disclosures: The authors declare no competing interests exist.

Submitted: 12 April 2024

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

# Supplemental material

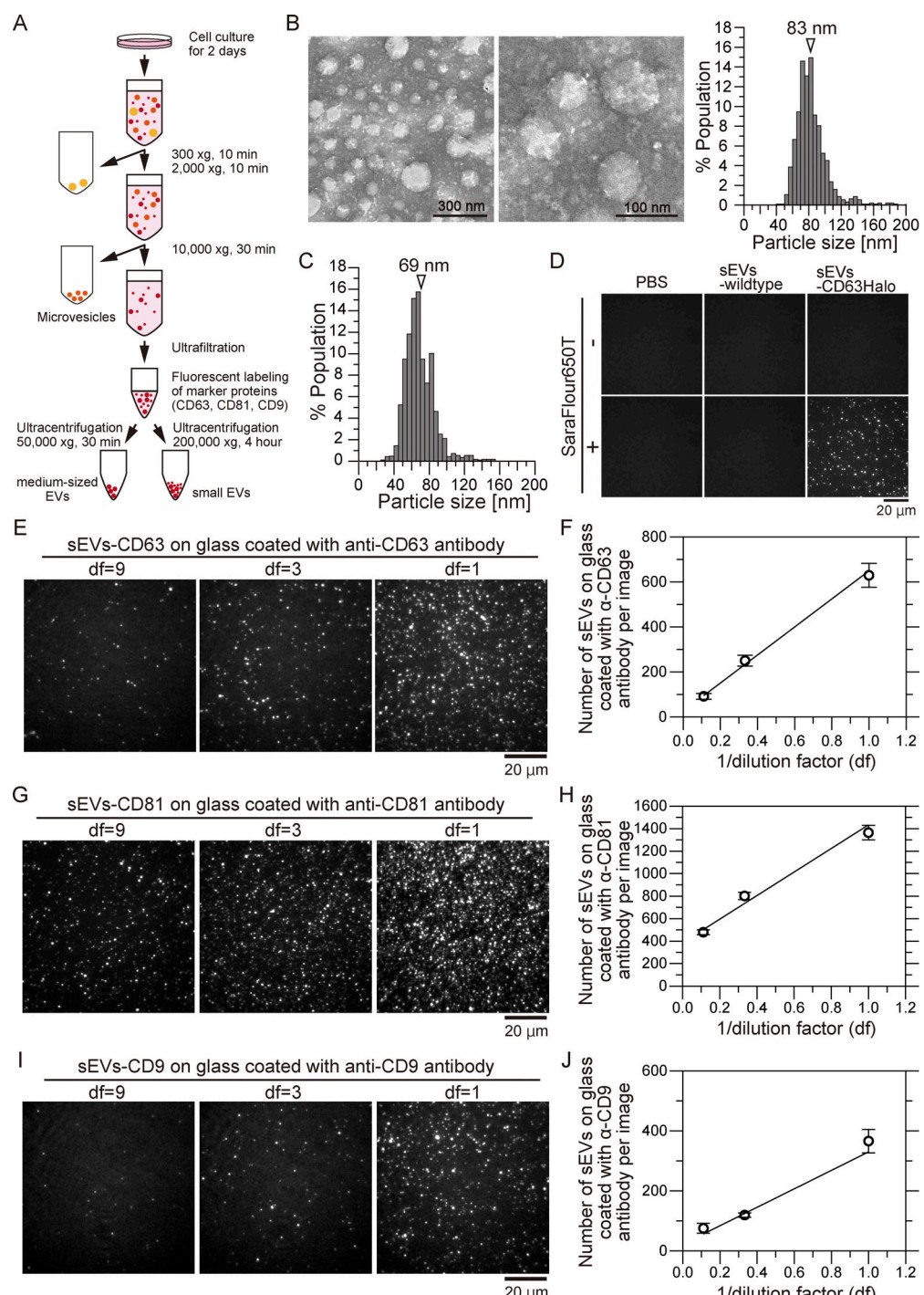

Figure S1. **Preparation of sEVs derived from PC3 cells and determination of their size and concentration. (A)** sEVs from PC3 cells were isolated from the cell culture supernatant by ultrafiltration and ultracentrifugation. Tetraspanins tagged with Halo7 in sEVs were fluorescently labeled with SaraFluor650T (SF650T). **(B)** Negative-staining TEM images of sEVs revealed that the mean size of the sEVs was 83 ± 19 nm (mean ± SD). **(C)** The mean size of the sEVs determined by qNano was 69 ± 17 nm, as indicated by the arrowhead. **(D)** Single-particle fluorescence images of sEVs by TIRFM. Only when sEVs expressed CD63-Halo7, single particles labeled with SF650T (sEVs–CD63Halo7-SF650T) could be observed. **(E, G, and I)** We directly measured the number of sEV-tetraspanin-Halo7-SF650T particles bound to the antibody-coated glass. After incubating the sEV solution ($2 \times 10^{10}$ particles/ml) on antibody-coated glass at a dilution factor (df) of 1, 3, or 9, followed by three washes with HBSS, we obtained TIRFM images of single sEV–CD63-Halo7-SF650T particles. Single-particle fluorescence images of three concentrations of sEV–CD63Halo7-SF650T particles (E), sEV–CD81Halo7-SF650T particles (G), and sEV–CD9Halo7-SF650T particles (I), which attached to glass coated with anti-CD63 antibody, anti-CD81 antibody, and anti-CD9 antibody, respectively. df indicates the dilution factor. **(F, H, and J)** The number of sEVs bound to the glass decreased in accordance with the dilution factor (df), which allowed us to obtain a calibration curve. The numbers of sEV–CD63Halo7-SF650T particles (F), sEV–CD81Halo7-SF650T particles (H), and sEV–CD9Halo7-SF650T particles (J) at three df values attached to the CD63 antibody, anti-CD81 antibody, and anti-CD9 antibody-coated glass, respectively ($n$ = 16 images). Data are presented as the mean ± SE. The sEV concentration was adjusted according to the calibration line.

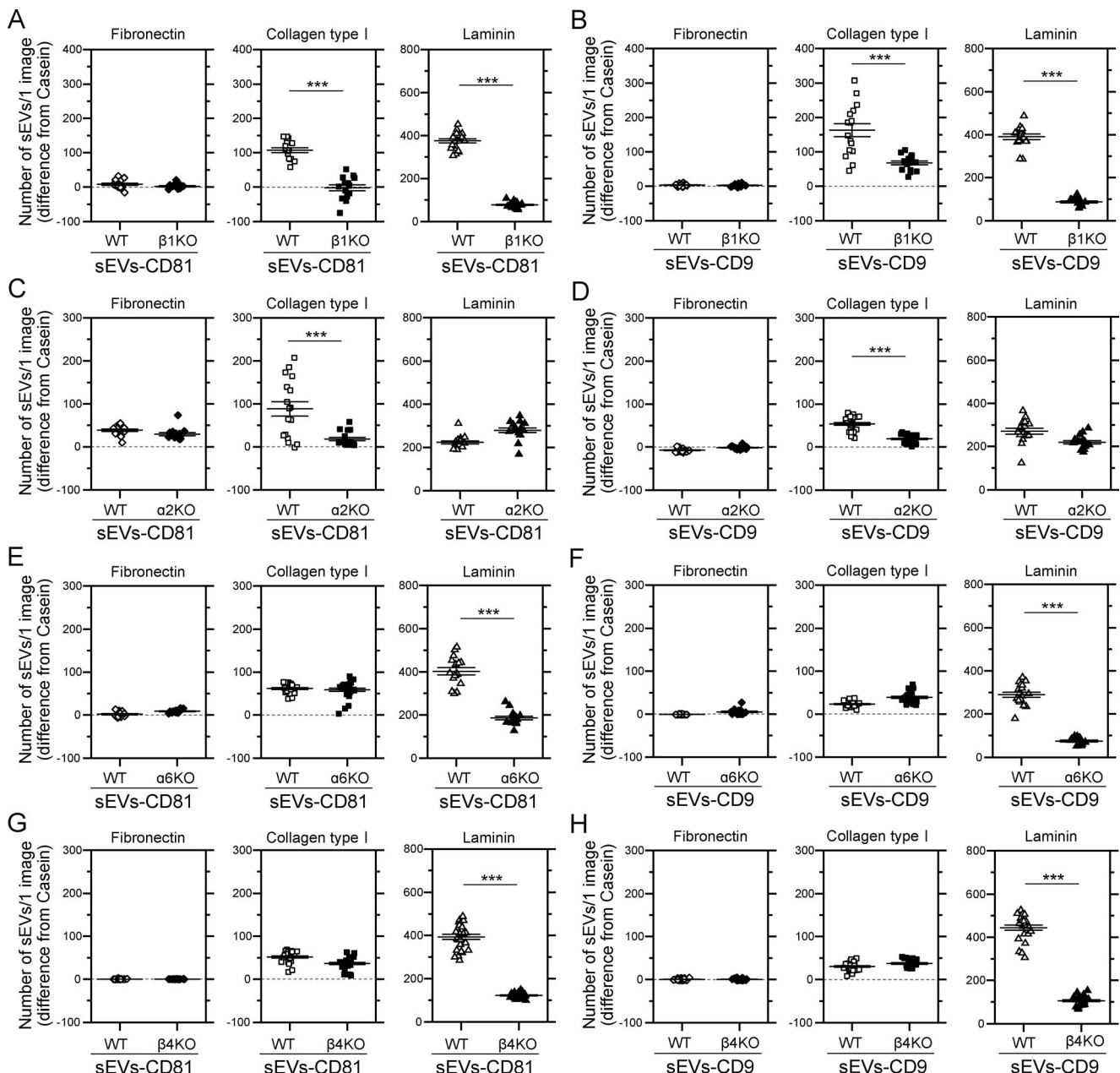

Figure S2. **The integrin β1, integrin α6, and integrin β4 subunits in sEVs containing CD81 or CD9 derived from PC3 cells are responsible for the binding of the sEVs to laminin, and the integrin α2 subunit is responsible for the binding of the sEVs to collagen type I. (A–H)** The numbers of intact PC3 cell–derived sEVs attached to glass coated with fibronectin, collagen I, and laminin were compared with those of sEVs derived from integrin β1 (A and B), integrin α2 (C and D), integrin α6 (E and F), and integrin β4 (G and H) KO cells. CD81-Halo7 (A, C, E, and G) and CD9-Halo7 (B, D, F, and H) in sEVs were labeled with SF650T. Data are presented as the mean ± SE. n.s., nonsignificant difference; ***P < 0.001 according to Welch's *t* test (two-sided).

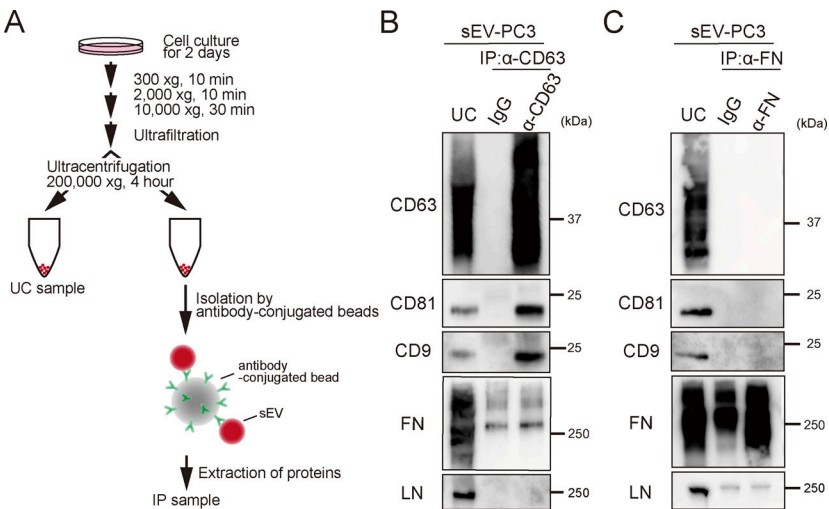

Figure S3. **Neither laminin nor fibronectin is present on PC3-derived sEV surfaces. (A)** Schematic diagram of the isolation of specific sEVs. sEVs were isolated by ultracentrifugation at 200,000 × *g* for 4 h (UC sample). Then, special sEVs were isolated from sEVs by a bead-conjugating antibody (IP: immunoprecipitation sample). **(B)** Western blot analysis of tetraspanin (CD63, CD81, and CD9), fibronectin (FN), and laminin (LN) in UC and IP samples. sEVs were isolated by IP using an anti-CD63 antibody. **(C)** Western blot analysis of sEVs isolated by IP using anti-fibronectin antibody. Source data are available for this figure: SourceData FS3.

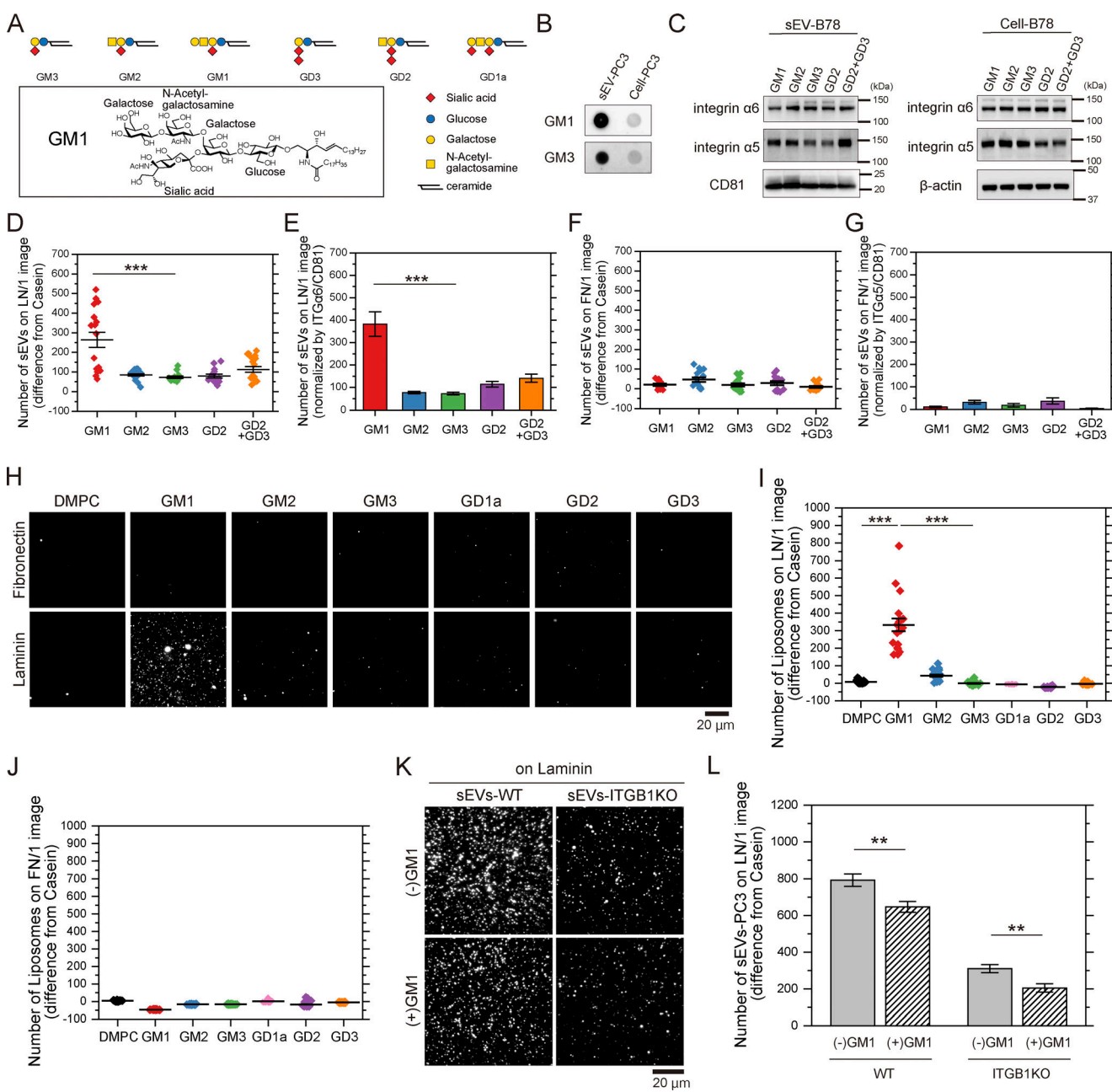

Figure S4. **GM1 is responsible for the binding of sEVs to laminin. (A)** (top) Schematic diagram of gangliosides and (bottom) chemical structure of GM1. Monosaccharide symbols follow the Symbol Nomenclature for Glycans (SNFG). **(B)** Dot blotting of GM1 and GM3 in PC3 cells and PC3-derived sEVs. **(C)** Western blot analysis of integrin subunits in B78 cell lines with abundant expression of one type of ganglioside—GM1, GM2, GM3, GD2, or GD2/GD3—and sEVs derived from these cells. **(D and E)** The numbers of sEVs attached to glass coated with laminin (D) and the numbers normalized to the ratio of integrin α6/CD81 in the sEVs (E). **(F and G)** The numbers of sEVs attached to glass coated with fibronectin (F) and the numbers normalized to the ratio of integrin α5/CD81 in the sEVs (G). **(H)** Single-particle fluorescence images of DMPC liposomes containing GM1, GM2, GM3, GD1a, GD2, or GD3 on glass coated with fibronectin or laminin. **(I and J)** The numbers of liposomes attached to glass coated with laminin (I) or fibronectin (J). **(K)** Single-particle fluorescence images of sEVs-PC3-CD63Halo7-TMR and sEVs-PC3-ITGB1KO-CD63Halo7-TMR on laminin before or after treatment of the GM1's glycan. **(L)** The numbers of sEVs bound to laminin before or after treatment with a high concentration of the GM1 glycan moiety (0.5 mM final). Data are presented as the mean ± SE. n.s., nonsignificant difference; ***P < 0.001 according to Welch's t test (two-sided). In I and L, due to the necessity of multiple statistical tests, the significance level was corrected by the Bonferroni method and determined to be 0.025 (=0.05/2). Source data are available for this figure: SourceData FS4.

Figure S5. **Expression levels of gangliosides in B78 cell PMs analyzed by flow cytometry.** The cells were stained as described in the Materials and methods. Control specimens were prepared without the primary antibodies or biotinylated cholera toxin B (black traces). The red traces indicate the molecules that were stained with antibodies (positive), and the blue traces indicate molecules that were not stained (negative) or were only slightly stained (weakly positive).

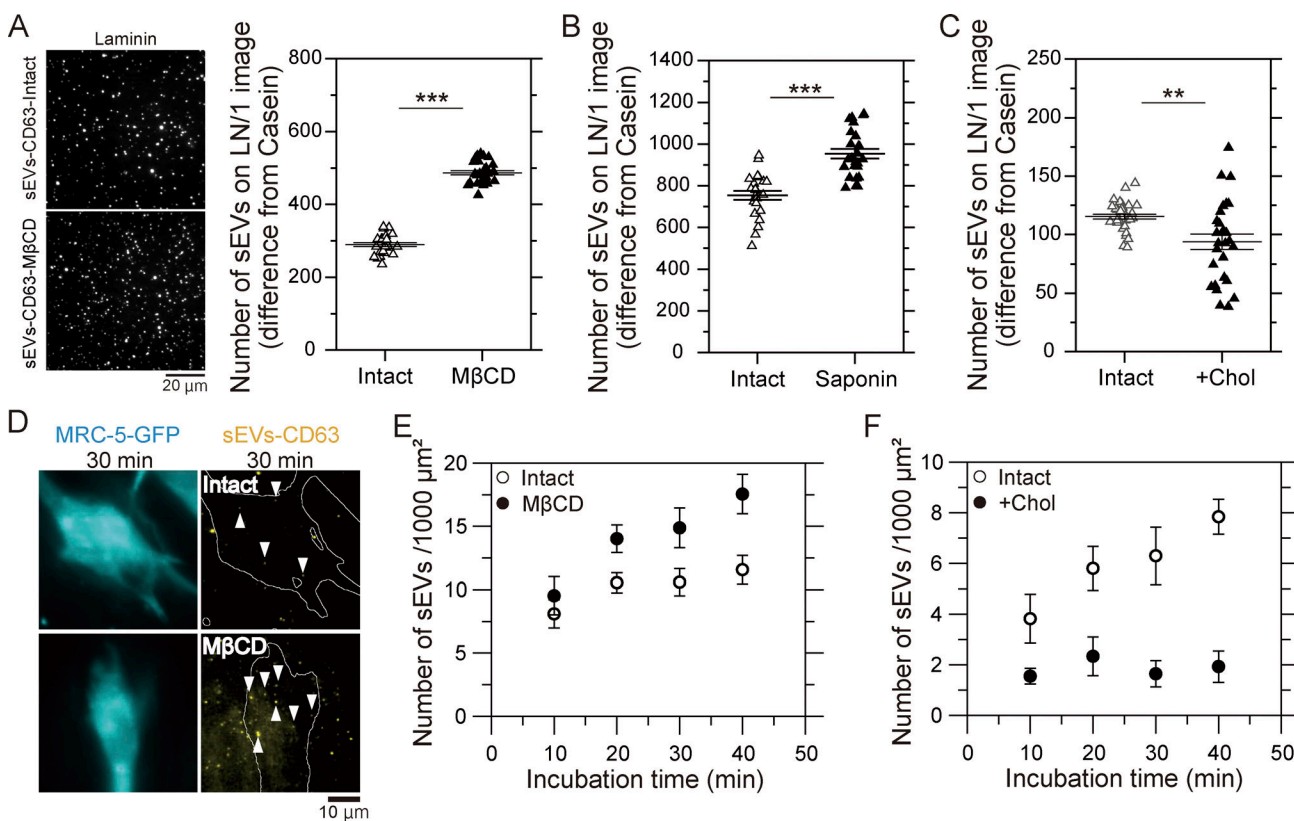

Figure S6.  **Cholesterol impairs the binding of sEVs to laminin and the MRC-5 cell PM. (A)** Fluorescence images of sEV–CD63Halo7-SF650T particles bound to laminin (LN) on glass before and after cholesterol depletion by MβCD and the numbers of attached sEVs per image (82 × 82 μm). The cholesterol content in PC3 cell–derived sEVs was reduced to 16% after treatment with MβCD. **(B and C)** The numbers of sEV–CD63-Halo7-SF650T particles attached to glass coated with laminin before and after treatment with saponin (B) and the addition of cholesterol by MβCD–cholesterol complex (C).The cholesterol content was increased to 186% after treatment with the MβCD–cholesterol complex. **(D)** Fluorescence images of an MRC-5–GFP cell and sEV–CD63Halo7-SF650T particles on the MRC-5 cell after 30 min of incubation. **(E and F)** Time course of the number of sEV–CD63-Halo7-SF650T particles per 1,000 μm² attached to the MRC-5 cell membrane before and after treatment with MβCD (n = 16 cells) (E) or the MβCD–cholesterol complex (n = 8 cells) (F). Data are presented as the mean ± SE. n.s., nonsignificant difference; **P < 0.01; ***P < 0.001 according to Welch's t test (two-sided).

Video 1.  **Video showing the simultaneous observation of sEV-PC3-CD63Halo7-TMR particles (green) and immunostained laminin (magenta) on a living MRC-5 cell.** By connecting the dSTORM image sequences of laminin structure, we generated a pseudo real-time dSTORM movie, which was superimposed with a movie of sEV particles (33 frames/s). sEVs localized near (<100 nm) the boundary of laminin structure and sEVs localized alone are indicated by yellow and white arrowheads, respectively. Real-time replay; frame rate, 33 frames/s. Related to Fig. 6 D, right.

Video 2.  **Enlarged video of the simultaneous observation of laminin structure by dSTORM (magenta) and a single sEV-PC3-CD63Halo7-TMR particle (green) on a living MRC-5 cell membrane.** The field of view in this video is different from that of Video 3. Real-time replay; frame rate, 33 frames/s. Related to Fig. 6 E.

Video 3.  **Video showing the simultaneous observation of sEV-PC3-CD63Halo7-TMR particles (green) and immunostained fibronectin (magenta) on a living MRC-5 cell.** By connecting the dSTORM image sequences of fibronectin structure, we generated a pseudo real-time dSTORM movie, which was superimposed with a video of sEV particles (33 frames/s). sEVs localized near (<100 nm) the boundary of fibronectin structure are not observed and sEVs localized alone are indicated by white arrowheads. Real-time replay; frame rate, 33 frames/s. Related to Fig. 6 B, right.

Video 4.  **Video showing the simultaneous observation of sEV-PC3-CD63Halo7-TMR particles (green) and immunostained collagen type I (magenta) on a living MRC-5 cell.** By connecting the dSTORM image sequences of collagen type I structure, we generated a pseudo real-time dSTORM movie, which was superimposed with a video of sEV particles (33 frames/s). sEVs localized near (<100 nm) the boundary of fibronectin structure are not observed and sEVs localized alone are indicated by white arrowheads. Real-time replay; frame rate, 33 frames/s. Related to Fig. 6 C, right.

**Provided online is Table S1. Table S1 shows the the numbers of sEVs labeled with tetraspanin–Halo7-TMR attached to glass coated with the ECM.**

