## [Peer Review File · The Journal of Cell Biology]

Extracellular vesicles adhere to cells primarily by interactions of integrins and GM1 with laminin

Tatsuki Isogai, Koichiro Hirosawa, Miki Kanno, Ayano Sho, Rinshi Kasai, Naoko Komura, Hiromune Ando, Keiko Furukawa, Yuhsuke Ohmi, Koichi Furukawa, Yasunari Yokota, and Kenichi Suzuki

Corresponding Author(s): Kenichi Suzuki, Gifu University

Review Timeline:

Submission Date:	2024-04-12
Editorial Decision:	2024-07-12
Revision Received:	2024-12-09
Editorial Decision:	2025-02-05
Revision Received:	2025-02-20
Editorial Decision:	2025-02-28
Revision Received:	2025-03-07

Monitoring Editor: Kenneth Yamada

Scientific Editor: Dan Simon

Transaction Report:

DOI: <https://doi.org/10.1083/jcb.202404064>

July 12, 2024

Re: JCB manuscript #202404064

Prof. Kenichi G.N. Suzuki
Gifu University
iGCORE
Yanagido 1-1
Gifu, Gifu 501-1193
Japan

Dear Prof. Suzuki,

Thank you for submitting your manuscript entitled "Extracellular vesicles adhere to cells primarily by interactions of integrins and GM1 with laminin". We apologize for the delayed reviewing process due to a third reviewer who had promised but never delivered their review. Consequently, your manuscript was assessed by two highly expert reviewers, whose reports are appended below. As you can see from the reviews provided by these leaders in overlapping research areas spanning the elements of this paper, there were unfortunately some major concerns about the definitiveness and impact of the conclusions. Consequently, unfortunately, after an assessment of the reviewer feedback, our editorial decision at this point must be against publication in JCB.

We should note that in the initial editorial evaluation process, this paper was considered borderline, but because of the intriguing conclusions, we decided to send the paper out for full peer reviewing. As you can see, a key point raised by both reviewers was the need for evidence that depleting laminin on the target cells can prevent EVs from producing effects on the behavior of these cells. Also needed was stronger evidence for laminin-receptor integrin-dependent binding, better evaluation of biological relevance using endothelial cells, and resolution of other specific concerns including a concern about the level of talin knockdown. As noted by one of the reviewers, JCB would require either citation of a publication or reviewing of the data in the cited preprint.

Although your manuscript is intriguing, we feel that the points raised by the reviewers are more substantial than can be addressed in a typical revision period. If you wish to expedite publication of the current data, it may be best to pursue publication at another journal.

However, given interest in the topic, we would be open to resubmission to JCB of a significantly revised and extended manuscript that fully addresses the reviewers' concerns and is subject to further peer-review. If you would like to resubmit this work to JCB, please submit an appeal with a revision plan explaining how you will address each of the reviewer comments. Please note that priority and novelty would be reassessed at resubmission of the revised manuscript.

Although we very much regret that our decision must currently be negative for this manuscript, we hope that you find the reviews constructive, and we thank you for allowing us to see this work. Of course, this decision does not imply any lack of interest in your work, and we hope that you will still consider JCB for your highest quality work in the future. We would be happy to discuss the reviewer comments further once you've had a chance to consider the points raised in this letter. You can contact the journal office with any questions at cellbio@rockefeller.edu.

Thank you for your interest in the Journal of Cell Biology.

With kind regards,

Kenneth Yamada, MD, PhD
Senior Editor
Journal of Cell Biology

Dan Simon, PhD
Scientific Editor
Journal of Cell Biology

Reviewer #1 (Comments to the Authors (Required)):

In this manuscript, the authors provide evidence that extracellular vesicles (EVs) can adhere to extracellular matrix proteins

laminin and to a lesser extent collagen I, but not very well to fibronectin. CRISPR deletion of laminin binding integrin subunits (alpha6, beta1, or beta4) significantly reduced adhesion of EVs to a commercial laminin preparation thought to consist primarily of laminin-511, while deletion of collagen-binding subunits (alpha2 or beta1) reduced EV adhesion to collagen I. Ganglioside GM1 might also contribute to EV adhesion to laminin-coated glass. Some evidence is presented to support the hypothesis that EVs bind to pericellular laminin on target cells in an integrin-dependent manner. EV adhesion on laminin-coated glass slides depended partially on tetraspanin CD151, but was not influenced by manipulating talin expression in EVs. Overall, the study data support that integrin-containing EVs can bind to laminin. The use of CRISPR in source cells to create EVs depleted of integrin subunits is clean and convincing (for example, depleting integrin beta1 also depletes integrin alpha2 for which beta1 is an obligate partner to form the alpha2/beta1 heterodimer). However, the significance of the study is unclear since no EV-dependent cell behavior or phenotype is investigated, only EV binding. This and several other issues outlined below would prevent publication in JCB at this time, in this reviewer's opinion.

Major issues

- If the authors could identify an important EV-dependent target cell behavior and then show that depleting laminin in the target cells both reduces EV binding and blocks the EV-dependent cell behavior then the potential significance and impact of the study would be much more firmly established.
- A significant proportion of the potential significance depends on data from a preprint that has not yet completed peer review.
- Although the data on EV binding to purified matrix components is convincing, the data on EV binding to live cell-associated matrix proteins is less so. The number of EVs binding per cell appears relatively low. It is difficult to judge if enough EVs would bind in an integrin-dependent manner to affect target cell phenotype, behavior, or gene expression. The number of EVs that bind to matrix-poor HS-5 cells is not that different than the number that bind to matrix-rich MRC-5 cells.
- The potential role of cholesterol in regulating EV binding to laminin is rather speculative, invoking a "molecule X."

Moderate/minor issues

- Depleting beta1 integrin did not result in depleting alpha5 integrin in Figure 1A, despite that it did result in depleting alpha2 integrin. This raises a question of whether the alpha5 integrin band shown in the immunoblot is authentic alpha5 integrin.
- There is a lot of variability in the absolute number of EVs that bind to matrix proteins in different experiments. For example, binding of WT sEVs to laminin shown in Table S1 varies from 131 plus/minus 5.3 to 1029 +/- 18.3. Is this a reflection of the quality of different preps of sEVs? If so, how does that influence the interpretation of the data?
- The commercial source of laminin used for most of the purified matrix experiments is placental laminin that is thought to be mostly laminin-511, but also known to be proteolyzed such that the globular G-domain may be partially degraded to greater or lesser extents in different preps. In addition, although laminin-511 is a good ligand for alpha6/beta1 and alpha6/beta4 integrin, the antibody used to detect laminin in the live cell experiments was raised against EHS laminin, which contains a lot of laminin-111. It is not clear which isoform of laminin the MRC-5 cells are expressing that might support binding of integrin-containing EVs.
- No analysis of the proportion of EVs that express different integrin heterodimers is provided. Do all EVs express alpha6/beta1, alpha6/beta4, alpha2/beta1 and various fibronectin receptors?
- How do the GM1 data relate to the integrin data? If GM1 is sufficient to mediate EV attachment to laminin, why does depleting different integrins make such a big difference?

Reviewer #2 (Comments to the Authors (Required)):

In this work, the authors set out to investigate the role of integrins in the binding of small extracellular vesicles (sEVs) to extracellular matrix components and fibroblasts. This question is interesting because integrins have been reported to confer organ specificity on EVs, but the molecular details of how this might actually work are not known. Through a range of in vitro assays, the authors conclude that laminin-binding integrins are critical, notably ITGA6, ITGB4, and ITGB1. The binding assays and microscopy are done to a high standard. However, the main message currently falls short of what I would expect for JCB. In particular, it remains not properly resolved if integrin-laminin binding is via the same molecular mechanism and interface as 'conventional' integrin-mediated adhesions. Also, there is no demonstration that interference with laminin binding prevents sEVs from having a functional effect on recipient cells.

Major comments

1. The authors need to determine if the integrin-laminin interaction relies on the same molecular interface as conventional integrin α 6 β 1-laminin interaction. To do this they should engineer in some of the mutations reported by Arimori et al (Nat Comms 2021) to disrupt integrin-laminin interaction.

2. The authors focus heavily on MRC5 fibroblasts. To improve the biological relevance, they should test if the same mechanism is involved in sEV binding to endothelial cells, which will represent the main cell type that circulating sEVs will encounter.
3. The talin experiment is interesting, but not satisfactorily performed. The extent of knockdown is rather modest and the authors do not demonstrate that the level of knockdown is sufficient to perturb integrin-mediated adhesion to laminin in a conventional cellular context. This is critical - without this the lack of effect cannot be interpreted. The authors should also test the effect of Kindlin depletion/deletion with a similar set of experiments.
4. The authors should try staining sEVs bound to matrix components with integrin conformation specific antibodies - HUTS4 and HUTS21 would be informative in certain contexts.
5. The manuscript would be greatly enhanced if the authors could determine if laminin depletion in recipient cells prevent sEVs having a biological effect on the them.

Minor comments

6. The STORM timelapse set up is not well explained.
7. The manuscript is rather oddly structured and the first few figures could be compressed. The GM1 and M β CD experiments could be moved to supplementary figures. This would aid conveying the main messages to the readers.
8. The rationale for the transition between cholesterol and CD151 is not clear to me.

Reviewer #1 (Comments to the Authors (Required)):

In this manuscript, the authors provide evidence that extracellular vesicles (EVs) can adhere to extracellular matrix proteins laminin and to a lesser extent collagen I, but not very well to fibronectin. CRISPR deletion of laminin binding integrin subunits (alpha6, beta1, or beta4) significantly reduced adhesion of EVs to a commercial laminin preparation thought to consist primarily of laminin-511, while deletion of collagen-binding subunits (alpha2 or beta1) reduced EV adhesion to collagen I. Ganglioside GM1 might also contribute to EV adhesion to laminin-coated glass. Some evidence is presented to support the hypothesis that EVs bind to pericellular laminin on target cells in an integrin-dependent manner. EV adhesion on laminin-coated glass slides depended partially on tetraspanin CD151, but was not influenced by manipulating talin expression in EVs. Overall, the study data support that integrin-containing EVs can bind to laminin. The use of CRISPR in source cells to create EVs depleted of integrin subunits is clean and convincing (for example, depleting integrin beta1 also depletes integrin alpha2 for which beta1 is an obligate partner to form the alpha2/beta1 heterodimer). However, the significance of the study is unclear since no EV-dependent cell behavior or phenotype is investigated, only EV binding. This and several other issues outlined below would prevent publication in JCB at this time, in this reviewer's opinion.

Thank you very much for your insightful and constructive comments. In response, we performed additional experiments and revised the manuscript in accordance with all the reviewer's suggestions. We believe that these revisions have significantly strengthened the manuscript.

We fully agree with the reviewer's comments. In our study, we investigated the mechanism of EV binding to recipient cells; however, we did not examine EV-dependent cellular behaviors or phenotypes. To address the reviewer's concerns, we explored whether laminin- and integrin-mediated binding of sEVs to recipient cells is involved in any cellular responses. It has been known that prostate cancer cell-derived sEVs promote angiogenic phenotypes in endothelial cells such HUVECs (Human umbilical vein endothelial cells). Therefore, we examined whether laminin-mediated binding of PC3-derived sEVs induces angiogenesis-associated morphogenic changes in HUVECs.

The HUVEC membrane surface is densely covered with laminin. We found that laminin γ 1 knockdown in HUVECs, which reduced its expression to 13% of its original level (Fig.

9A), correspondingly decreased total laminin to 34% (Fig. 9A) and resulted in a substantial reduction of sEV binding to 41% of the original level (Fig. 9B-D). Interestingly, branching morphological changes in HUVECs cultured on fibronectin-coated glass were observed 12 hours after the addition of PC3 cell-derived sEVs, comparable to the changes induced by VEGF treatment (Fig. 9E). Quantitative analysis of cell branching morphogenesis, based on the total length of protrusions (Fig. 9F), revealed that laminin $\gamma 1$ knockdown dramatically suppressed EV-induced cellular morphogenesis but had no significant effect on VEGF-induced morphogenesis (Fig. 9G-J).

These results demonstrate that while laminin is not required for HUVEC branching morphogenesis under these experimental conditions, the binding of sEVs to laminin is critical for sEV-induced morphogenesis. These results have been incorporated into Fig. 9 of the revised version and are described from the second paragraph on page 28 to the first paragraph on page 30, and discussed in the first paragraph on page 36.

Major issues

- If the authors could identify an important EV-dependent target cell behavior and then show that depleting laminin in the target cells both reduces EV binding and blocks the EV-dependent cell behavior then the potential significance and impact of the study would be much more firmly established.

Thank you for your thoughtful and constructive comments. As outlined above, we explicitly showed that depletion of laminin $\gamma 1$ in target cells significantly reduced EV binding and inhibited HUVEC morphogenesis. On the other hand, VEGF-induced morphogenesis remained unaffected after laminin $\gamma 1$ knockdown. These results are incorporated into the revised manuscript as noted above.

- A significant proportion of the potential significance depends on data from a preprint that has not yet completed peer review.

Thank you very much for this comment. Indeed, it is important to show that the peer review process for the preprint has been completed. Our manuscript, currently under review at Nature Communications, has received support for publication from four out of

five reviewers. We are trying to respond to the comment by the remaining reviewer. The additional experiments have been completed, and we obtained results to address concerns raised by the reviewer. We will resubmit the revised version within a week.

- Although the data on EV binding to purified matrix components is convincing, the data on EV binding to live cell-associated matrix proteins is less so. The number of EVs binding per cell appears relatively low. It is difficult to judge if enough EVs would bind in an integrin-dependent manner to affect target cell phenotype, behavior, or gene expression. The number of EVs that bind to matrix-poor HS-5 cells is not that different than the number that bind to matrix-rich MRC-5 cells.

Thank you for these thoughtful comments. EVs on cell membranes were visualized using total internal reflection fluorescence microscopy (TIRFM), a technique comparable to the observation of EVs bound to laminin on glass. Consequently, only EVs attached to the basal side of the plasma membrane were detected, as they are located between the basal membrane and the glass substrate. We fully agree with the reviewer's comment.

Previously, individual EV particles on living cell membranes could not be resolved using confocal fluorescence microscopy, as their slight movement and the short detector exposure time limited the detection. In the revised version, we employed chemical fixation and extended the detector exposure times to successfully observe individual sEV particles on both the apical and basal surfaces of cell membranes using confocal fluorescence microscopy.

Figs. 4I and 4J show the number of PC3-derived sEVs bound to MRC-5 cell membranes after incubation for 60 min. The number of sEVs on MRC-5 cell membranes observed by confocal fluorescence microscopy was approximately 6-fold higher than that observed with TIRFM (Fig. 4F vs. Fig. 4J) and 11-fold greater than that observed on HS-5 using confocal fluorescence microscopy (Fig. 4J). Furthermore, the number of PC3 cell-derived sEVs bound to MRC-5 cell membranes observed by confocal microscopy was approximately double that of sEVs from integrin β 1 KO PC3 cells (Fig. 4J). In contrast, the number of PC3-derived sEVs bound to HS-5 cell membranes was comparable to the number from integrin β 1 KO PC3 cells (Fig. 4J). These results explicitly show that PC3-derived sEVs bind to MRC-5 cell membranes in an integrin-1-dependent manner. These new results are described from the fourth line from the bottom of page 19 to the fourth line of page 20. (While not all the cells necessarily express GFP and many fluorescent spots of sEV-CD63Halo7-TMR are observed outside GFP-expressing cells in Fig. 4I, the number of sEV-CD63Halo7-TMR bound to both the apical and basal surfaces of the cell

PM can be quantified by counting the fluorescent EV spots on cells expressing GFP)

Additionally, both confocal fluorescence microscopy and TIRFM were used to quantify the number of sEV particles bound to HUVECs before and after laminin knockdown, as shown in Figs. 9B-D. These analyses revealed that sEVs binding to cells occurs in a laminin-dependent manner.

- The potential role of cholesterol in regulating EV binding to laminin is rather speculative, invoking a "molecule X."

Thank you for this comment. It is evident that EV binding to laminin was significantly enhanced after cholesterol depletion and that the effect of cholesterol on increased EV binding to laminin is diminished in the absence of CD151. However, as molecule X has not yet been identified, we have removed Fig. 10C and the associated description of membrane component(s) X on pages 30-31 of the original manuscript. Additionally, we have shortened the discussion regarding the attenuation of CD151-dependent activation of integrin heterodimers by association of CD151 with lipid rafts (as described in the latter half of page 34 in the revised manuscript).

Moderate/minor issues

- Depleting beta1 integrin did not result in depleting alpha5 integrin in Figure 1A, despite that it did result in depleting alpha2 integrin. This raises a question of whether the alpha5 integrin band shown in the immunoblot is authentic alpha5 integrin.

We used a rabbit polyclonal anti-integrin $\alpha 5$ antibody (Cell Signaling, Cat# 4705S), which has been validated for western blotting in 89 publications (e.g., Chen et al., *Dev. Cell*, 56, 3250-3263, 2021). To further validate our results in the revised version, we also employed a different antibody against integrin $\alpha 5$. Specifically, we used a rabbit monoclonal anti-integrin $\alpha 5$ antibody (EPR7854) (Abcam, ab150361), which has been validated for western blotting in 68 publications (e.g., Biering et al., *Nat. Commun.*, 13, 7630, 2022). In the revised version, Fig. 1A more explicitly shows the presence of integrin $\alpha 5$ in both integrin $\beta 1$ -KO PC3 cells and sEVs derived from these integrin $\beta 1$ -KO cells.

- There is a lot of variability in the absolute number of EVs that bind to matrix proteins in

different experiments. For example, binding of WT sEVs to laminin shown in Table S1 varies from 131 plus/minus 5.3 to 1029 +/- 18.3. Is this a reflection of the quality of different preps of sEVs? If so, how does that influence the interpretation of the data?

The number of sEVs containing CD81 bound to laminin was 131 ± 5.3 , compared to the number of EVs derived from $\alpha 6$ KO cells. On the other hand, the number of sEVs containing CD9 bound to laminin was 1029 ± 18.3 , compared to the number of sEVs derived from $\beta 4$ KO cells (Table S1). While the numbers of EVs in these two experimental systems varied significantly, the numbers in other systems were similar. This large discrepancy is attributed to variations in EV density ($0.8-4 \times 10^{10}$ particles/ml) used in the experiments, as described in the Methods section (the second paragraph on page 55 of the revised manuscript). To enable direct comparisons of the number of EVs bound to ECMs, the density of EVs derived from intact cells was adjusted to match that of EVs derived from integrin KO cells. Although all EVs were isolated using the same method, the resulting EV density varied depending on the experimental system. To confirm whether integrin KO significantly reduces the number of EVs bound to laminin in these systems (CD81-containing EVs before and after $\alpha 6$ integrin KO, and CD9-containing EVs before and after $\beta 4$ integrin KO), we performed single-particle imaging of sEVs after equalizing their densities. As shown in Table S1 of the revised manuscript, the number of sEVs containing CD81 bound to laminin increased to 403 ± 16.4 , which was larger than that derived from $\alpha 6$ KO cells (186 ± 7.9). Similarly, the number of sEVs containing CD9 bound to laminin increased to 444 ± 11.5 , which was larger than that derived from $\beta 4$ KO cells (106 ± 4.4). We believe these results effectively address the issue. These results are incorporated into Fig. S2E right, Fig. S2H right, and Table S1 of the revised manuscript.

- The commercial source of laminin used for most of the purified matrix experiments is placental laminin that is thought to be mostly laminin-511, but also known to be proteolyzed such that the globular G-domain may be partially degraded to greater or lesser extents in different preps. In addition, although laminin-511 is a good ligand for $\alpha 6/\beta 1$ and $\alpha 6/\beta 4$ integrin, the antibody used to detect laminin in the live cell experiments was raised against EHS laminin, which contains a lot of laminin-111. It is not clear which isoform of laminin the MRC-5 cells are expressing that might support binding of integrin-containing EVs.

Thank you for this thoughtful comment. We performed immunostaining of laminin-111

and laminin-511 using anti-laminin α 1 and α 5 antibodies, respectively, and analyzed their densities by epi-fluorescence microscopy (Fig. 4A). The density of laminin α 5 on MRC-5 cells was significantly higher than that of laminin α 1, whereas both were scarcely detectable on HS-5 cells. These results indicate that laminin-511 is considerably more abundant than laminin-111 on MRC-5 cells and that the laminin isoform on MRC-5 cells closely resembles the commercially sourced laminin used in our experiments. We have included these new results in Fig. 4A and described in lines 6 to 9 of page 19 in the revised manuscript.

- No analysis of the proportion of EVs that express different integrin heterodimers is provided. Do all EVs express α 6/ β 1, α 6/ β 4, α 2/ β 1 and various fibronectin receptors?

We performed western blot analysis of integrin subunits across all five cell lines used in Fig. 5 of the original manuscript (now Fig. 3 in the revised version). The results for integrin α 2, α 3, and β 4 have been incorporated into the revised manuscript (shown in the bottom three rows of Fig. 3F).

- How do the GM1 data relate to the integrin data? If GM1 is sufficient to mediate EV attachment to laminin, why does depleting different integrins make such a big difference?

We were unable to deplete GM1 in integrin subunit KO-PC3 cells by inhibiting GM1 synthase, as GM1 levels returned to their original state during cell culture. Consequently, we could not directly assess the relative contributions of integrin and GM1 to EV binding to laminin using cells with both integrin subunit knockout and GM1 depletion. Instead, we synthesized the GM1 glycan moiety and performed a competitive binding inhibition assay using EVs derived from wild-type cells or integrin β 1 KO cells in the presence of a high concentration of GM1 glycan (0.5 mM). As shown in Fig. S4L of the revised version, the number of EVs bound to laminin decreased to 39% of the original levels after integrin β 1 KO, indicating that more than half of the EVs likely bind to laminin via integrins. This experiment parallels the one shown in the right panel of Fig. 2B, with consistent results across both datasets.

Furthermore, we quantified the contribution of GM1 to sEV binding to laminin. The addition of a high concentration of GM1 glycan (0.5 mM final concentration) significantly reduced the number of wild-type PC3 cell-derived sEVs bound to laminin by 18% and that of integrin β 1-KO cell-derived sEVs by 34% ($p < 0.01$, Welch's t-test;

Fig. S4L). These results show that while integrin heterodimers in sEVs predominantly mediate laminin binding, GM1 glycan in sEVs also partially facilitates the interaction with laminin. These new results were incorporated into Fig. S4L, and described in the second paragraph on page 16 of the revised manuscript.

Reviewer #2 (Comments to the Authors (Required)):

In this work, the authors set out to investigate the role of integrins in the binding of small extracellular vesicles (sEVs) to extracellular matrix components and fibroblasts. This question is interesting because integrins have been reported to confer organ specificity on EVs, but the molecular details of how this might actually work are not known. Through a range of in vitro assays, the authors conclude that laminin-binding integrins are critical, notably ITGA6, ITGB4, and ITGB1. The binding assays and microscopy are done to a high standard. However, the main message currently falls short of what I would expect for JCB. In particular, it remains not properly resolved if integrin-laminin binding is via the same molecular mechanism and interface as 'conventional' integrin-mediated adhesions. Also, there is no demonstration that interference with laminin binding prevents sEVs from having a functional effect on recipient cells.

Thank you very much for your thoughtful and constructive comments. In response, we performed additional experiments and revised the manuscript in accordance with all the reviewer's suggestions. We believe that the revisions have significantly strengthened the manuscript.

Major comments

1. The authors need to determine if the integrin-laminin interaction relies on the same molecular interface as conventional integrin $\alpha 6 \beta 1$ -laminin interaction. To do this they should engineer in some of the mutations reported by Arimori et al (Nat Comms 2021) to disrupt integrin-laminin interaction.

Thank you very much for your thoughtful and constructive comment. We fully agree with the reviewer's comment. Therefore, we compared the number of integrin $\alpha 6$ KO PC3 cell-derived sEVs bound to laminin with those derived from cells expressing similar levels of either wild-type integrin $\alpha 6$ or the R155A mutant. Integrin $\alpha 6$ KO PC3 cells were transfected with a plasmid encoding either wild-type integrin $\alpha 6$ or the R155A mutant using a PiggyBac transposon vector system (Yusa et al., Nat. Methods, 6, 363-

369, 2009) to achieve expression levels comparable to endogenous integrin $\alpha 6$.

As shown in Fig. 5D, the levels of rescued integrin $\alpha 6$ and the R155A mutant in sEVs were comparable to the levels of endogenous integrin $\alpha 6$ in sEVs derived from PC3-WT cells. Figs. 5E and 5F show that the number of sEVs containing the integrin $\alpha 6$ R155A mutant bound to laminin was significantly lower than those containing endogenous or rescued wild-type integrin $\alpha 6$, but comparable to that of sEVs from integrin $\alpha 6$ KO PC3 cells.

These results explicitly show that the interaction between integrin heterodimers in sEVs and laminin relies on the same molecular interface as the conventional interaction observed in cells. These new results are described from the third paragraph on page 20 to the first paragraph on page 21 and discussed from the third paragraph on page 32 to the first paragraph on page 33 of the revised manuscript.

2. The authors focus heavily on MRC5 fibroblasts. To improve the biological relevance, they should test if the same mechanism is involved in sEV binding to endothelial cells, which will represent the main cell type that circulating sEVs will encounter.

Thank you very much for this thoughtful comment. We fully agree with the reviewer's comment. As described in our response to reviewer 1's comment, we examined the binding of PC3-derived sEVs to laminin on an endothelial cell, HUVEC. We found that laminin $\gamma 1$ knockdown in HUVEC, reducing its expression to 13% of the original level (Fig. 9A), significantly reduced the number of sEVs binding to the cell surface to 41% of the original level (Fig. 9B-D). The results are described from the second paragraph on page 28 to the second line from the bottom on page 29 of the revised manuscript.

3. The talin experiment is interesting, but not satisfactorily performed. The extent of knockdown is rather modest and the authors do not demonstrate that the level of knockdown is sufficient to perturb integrin-mediated adhesion to laminin in a conventional cellular context. This is critical - without this the lack of effect cannot be interpreted. The authors should also test the effect of Kindlin depletion/deletion with a similar set of experiments.

Thank you very much for your thoughtful comment. We agree that the knockdown efficiency is modest. Our attempts to achieve a complete knockout of talin-1 in PC3 cells were unsuccessful and therefore we employed a knockdown approach instead. As the reviewer correctly noted, the reduction in talin-1 content in sEVs after the knockdown was modest (approximately 40%), as shown in Fig. 7D of the original manuscript. To

address this, we additionally performed talin-1 knockdown experiments in BxPC3 cells, which resulted in a substantial reduction of talin-1 content in sEVs to 11% of the original level (Fig. 7K of the revised manuscript). Importantly, the laminin-binding affinity of sEVs derived from talin-1 KD BxPC3 cells was not significantly different from that of sEVs derived from intact cells (Fig. 7, L and M of the revised manuscript). These results are described in the first paragraph on page 25 of the revised manuscript.

Furthermore, we performed a cell spreading assay using PC3 and BxPC3 cells both before and after talin-1 knockdown. We found that talin-1 knockdown dramatically suppressed the spreading of PC3 and BxPC3 cells compared to their pre-knockdown counterparts. These new results are incorporated in Fig. 7A-C and described in the second paragraph on page 24 of the revised manuscript.

In response to the reviewer's suggestion, we also attempted a knockdown of kindlin-2, which is abundantly expressed in cancer cells. However, as shown in Fig. 7O, kindlin-2 was not detected in sEVs, demonstrating that kindlin-2 does not contribute to the activation of integrin in sEVs derived from PC3 cells. This new result is incorporated in Fig. 7O and described in the first paragraph on page 26 of the revised manuscript.

4. The authors should try staining sEVs bound to matrix components with integrin conformation specific antibodies - HUTS4 and HUTS21 would be informative in certain contexts.

Thank you very much for your thoughtful and valuable comment. Following the reviewer's suggestion, we performed immunostaining of EVs bound to laminin-coated glass or uncoated glass using anti-integrin β 1 antibodies, clones P5D2, HUTS4, and HUTS21. The P5D2 antibody binds to both inactive and activated forms of integrin β 1, whereas HUTS4 and HUTS21 specifically recognize the activated conformation of integrin β 1. Nearly all sEV-CD63Halo7-TMR particles on uncoated glass and laminin-coated glass were stained with the P5D2 antibody (Fig. 5A-C). In contrast, only 20~30% of the sEV particles on uncoated glass were stained with HUST4 and HUTS21, which detect activated integrin β 1, whereas 60~70% of the sEV particles bound to laminin-coated glass were stained with these antibodies (Fig. 5A-C). These results explicitly indicate that integrin β 1 in sEVs adopts an active conformation upon binding to laminin. These new results are described in the second paragraph on page 20 and discussed from

the third paragraph on page 32 to the first paragraph on page 33 of the revised manuscript.

5. The manuscript would be greatly enhanced if the authors could determine if laminin depletion in recipient cells prevent sEVs having a biological effect on the them.

Thank you very much for your thoughtful and constructive comment. We fully agree with the reviewer's comment. We analyzed the endothelial cell branching morphogenesis of HUVECs with laminin knockdown, both before and after the addition of sEVs. The results were highly intriguing and are described in the initial part of our response to reviewer 1.

Minor comments

6. The STORM timelapse set up is not well explained.

We appreciate the reviewer's comment. The section on dSTORM data acquisition and the generation of a pseudo-real-time dSTORM movie in the Methods section was difficult to follow. Therefore, we revised this section, spanning from page 59 to the first paragraph on page 62, as outlined follows.

- 1) We elaborated on the rationale for using SF650B as a probe. Since SF650B spontaneously blinks, it enables the observation of dSTORM movies in living cells. (in the middle of page 59)
- 2) We described the final magnification of the image (the first paragraph on page 60).
- 3) We re-measured and updated the laser power values for TMR and SF650B at the sample plane (the first paragraph on page 60).
- 4) We explained the data acquisition process for dSTORM images and the subsequent connection of dSTORM images to create a dSTORM movie in the Methods section (the first paragraph on page 60). While the same description exists in the main text (the first paragraph on page 22), including it in the Methods section improves clarity and accessibility.
- 5) To maintain contextual flow, we added the sentence "More details regarding the data acquisition process for dSTORM imaging and the generation of the dSTORM movie are provided below." following the description in point 4 at the beginning of the second paragraph on page 60.

7. The manuscript is rather oddly structured and the first few figures could be compressed.

The GM1 and M β CD experiments could be moved to supplementary figures. This would aid conveying the main messages to the readers.

We agree with the reviewer's comment and have addressed it accordingly. Fig. 1 and Fig. 2 have been combined, and Fig. 4 and Fig. 9A-F have been relocated to Figs. S4 and Fig. S6, respectively. In addition, we shortened the content related to Fig. S4 (spanning from the second paragraph on page 14 to the second paragraph on page 16 of the revised version) and Fig. S6 (from the second paragraph on page 26 to the first paragraph on page 27 of the revised version). Some sentences in these sections have been also relocated to the Methods section (page 52) and the caption of Fig. S6 in the revised manuscript to better emphasize the primary message to the readers.

8. The rationale for the transition between cholesterol and CD151 is not clear to me.

Thank you for this thoughtful comment. We fully agree with this comment. In the revised manuscript, we have updated the introductory sentence at the beginning of the section titled "CD151 preserves the binding of integrin heterodimers in sEVs to laminin" in the second paragraph on page 27 as follows:

"Since cholesterol inhibits the activity of laminin receptors in sEVs, we investigated the role of a membrane molecule that interacts with both cholesterol and integrin heterodimers in laminin binding in sEVs lacking inside-out signaling."

Cholesterol depletion from sEV membranes significantly increased the number of sEVs bound to laminin (Figs. S6, A, B, and E of the revised version), whereas the addition of cholesterol to sEVs markedly reduced their binding to laminin (Figs. S6, C and F of the revised version). Meanwhile, CD151 knockdown substantially diminished sEV binding to laminin (Fig. 8C). Since CD151 is a cholesterol-binding protein, we hypothesized the presence of a mechanism that regulates the activity of the integrin-CD151 complex in a cholesterol-dependent manner. Indeed, the enhancement of sEV binding to laminin induced by cholesterol depletion was attenuated in the absence of CD151 (Fig. 8C and D). However, as the identity of molecule X remains unknown, we removed Fig. 10C, omitted references to molecule X, and shortened the discussion on the attenuation of CD151-dependent activation of integrin heterodimers via its association with lipid rafts, as described in the latter half of page 34 in the revised manuscript.

February 5, 2025

Re: JCB manuscript #202404064R-A

Kenichi Suzuki
Gifu University

Dear Prof. Suzuki,

Thank you for submitting your revised manuscript entitled "Extracellular vesicles adhere to cells primarily by interactions of integrins and GM1 with laminin" to the Journal of Cell Biology. The manuscript has now been assessed by the expert reviewers, whose reports are appended below. As you can see from the reviews provided by leaders in overlapping research areas spanning the elements of this paper, there was appreciation for the effectiveness of your revisions and encouragement to provide some final revisions. Our general policy is that papers are considered through only one revision cycle; however, given that the suggested changes are relatively minor we are open to one additional short round of revision.

The reviewers requested some additional information and discussion that we ask you to consider carefully. For example, it would indeed be helpful to see statistics for Fig. 5f, as well as clarification - to the extent practical - of the findings that puzzled the reviewers. We would also like to hear the status of the related manuscript you have had under review.

We look forward to receiving a revised manuscript within one month. Please also submit a cover letter that includes a point by point response to the remaining reviewer comments. It will be re-reviewed at the senior JCB Editor level for a final determination about its appropriateness for publication.

Thank you for your interest in the Journal of Cell Biology and for your conscientious revisions. We look forward to seeing you final revisions. You can contact me or the scientific editor listed below at the journal office with any questions at cellbio@rockefeller.edu.

With kind regards,

Kenneth M. Yamada, MD, PhD
Senior Editor
Journal of Cell Biology

Dan Simon, PhD
Scientific Editor
Journal of Cell Biology

Reviewer #1 (Comments to the Authors (Required)):

In this revised manuscript, the authors provide evidence to strengthen their conclusion that extracellular vesicles (EVs) can bind to cell-associated laminin (and collagen) via an integrin-dependent mechanism and thereby influence cell behavior. EV binding to laminin appears to depend on tetraspanin CD151, which would fit with the well-documented, preferential association of CD151 with laminin-binding integrins. Another interesting aspect of the study is the apparent lack of contribution to talin and kindlin to laminin-binding by EVs. This suggests a potential contribution of tetraspanin-mediated integrin avidity to EV-laminin binding. Overall, as described below, the authors provide significant new data to address several of the major concerns this reviewer raised about the original manuscript; however, there remain some issues that would be good to address.

Original major issue 1: the significance of the study is unclear since no EV-dependent cell behavior or phenotype is investigated, only EV binding.

In response to this critique, the authors provided significant new data showing that EV binding to HUVEC endothelial cells could be suppressed by knockdown of laminin in HUVEC cells. Cellular protrusions elicited by EV exposure were also suppressed in the laminin-depleted cells, and importantly, cellular protrusions elicited by VEGF exposure were not suppressed by laminin depletion. This latter observation was an important control that suggests the depletion of laminin specifically impacted EV-dependent cellular responses. Although the biological significance of the HUVEC cell protrusions in these experiments is not entirely clear, these experiments at least provide proof-of-concept results that integrin-dependent EV binding to pericellular matrix may be able to induce a phenotypic response. In my opinion, these new data adequately address one of the major concerns raised during the first review.

Original major issue 2: A significant proportion of the potential significance depends on data from a preprint that has not yet completed peer review.

The authors report progress on getting the preprint manuscript published in a peer-reviewed journal; however, this has not yet been achieved. The importance of this is that the preprint manuscript contains data that could shed light on the mechanisms by which EVs binding to pericellular matrix might actually influence cell behavior. In the Discussion, the authors cite preprint results including EV endocytosis, and the activation of integrin signaling in target cells upon EV binding as contributing to EV-induced responses in target cells. The JCB editors may decide the extent to which having these data peer reviewed is a prerequisite for publishing this study in JCB.

Original major issue 3: Although the data on EV binding to purified matrix components is convincing, the data on EV binding to live cell-associated matrix proteins is less so. The number of EVs binding per cell appears relatively low.

The authors clarify that the low number of cell-associated EVs in the original study is due to the use of TIRFM, which can only detect EVs associated with the basal plasma membrane of the target cells. Using a different method in which cells were fixed and then analyzed via confocal microscopy, the authors provide new evidence that a larger number of EVs can bind per cell when both apical and basal EVs are imaged, and that integrin beta1 knockdown can reduce the number of EVs that bind to matrix rich MRC-5 cells, while matrix poor HS-5 cells still bind relatively few EVs. This is important because a low number of EVs associated with the basal plasma membrane could at least in part be explained by cell membrane protrusion over EVs that had already bound the cell-free substrate, rather than active EV binding to pericellular matrix. The data in Figure 4 micrographs show significant numbers of EVs on the substrate outside of cell boundaries. If possible, the authors should comment on any evidence or experimental detail that helps to show that basally located EVs result from active binding to pericellular matrix rather than being simply overspread by the cellular plasma membrane.

Original major issue 4: The potential role of cholesterol in regulating EV binding to laminin is rather speculative, invoking a "molecule X."

The authors responded to this comment by changing the text to remove some of the speculative aspects. However, the results of the cholesterol depletion experiments in Figure 8 and S6 are still difficult to interpret. EVs treated with MBCD to deplete cholesterol from their membranes do appear to bind better to laminin-coated glass and MRC-5 cells. CD151-depleted EVs bind less well to laminin-coated glass, whether they were MBCD-treated or not. The authors interpret this to suggest that "an unidentified molecule interacts with CD151 in a cholesterol-dependent manner to inhibit the binding of EVs to laminin." It is not clear why such an unidentified molecule would necessarily need to physically interact with CD151 to have such an effect. In addition, MBCD treatment can cause the shedding of a subset of integrin-tetraspanin complexes (and GM1) from lipid membranes (Claas et al., JBC (276) 7974-7984 2001). The effect of MBCD treatment on the integrin-tetraspanin content of the EVs should therefore be evaluated. Potentially, integrins that are not associated with cholesterol-dependent raft-like domains might more efficiently bind ligands, once raft-associated complexes are depleted. Upon revisiting these data, it also occurs to this reviewer that testing CD151-depleted EVs for binding to collagen might reveal a specificity of CD151 contribution to laminin-binding integrins. The prediction based on the model in Figure 10 is that CD151 depletion would not alter EV binding to collagen-coated glass. Conversely, it would be interesting to determine whether talin depletion did affect EV binding to collagen, despite its apparent lack of effect of EV binding to laminin.

The mild/moderate issues raised in the preliminary review have largely been addressed in the revision.

Reviewer #2 (Comments to the Authors (Required)):

In this revised manuscript, the authors extend their arguments regarding the criticality of integrin binding to laminin for extracellular vesicle activity. New data are added addressing the conformation of the integrin, molecular dependencies of binding, and the functional consequence of perturbations on endothelial cells. The work is significantly improved and could be published following some clarifications.

1. Fig 5F - stats should be provided for WT vs ITGA6 KO
2. Why do there appear to be different levels of EV binding to the surrounding coverslip in the laminin depleted HUVEC image?

Reviewer #1 (Comments to the Authors (Required)):

In this revised manuscript, the authors provide evidence to strengthen their conclusion that extracellular vesicles (EVs) can bind to cell-associated laminin (and collagen) via an integrin-dependent mechanism and thereby influence cell behavior. EV binding to laminin appears to depend on tetraspanin CD151, which would fit with the well-documented, preferential association of CD151 with laminin-binding integrins. Another interesting aspect of the study is the apparent lack of contribution of talin and kindlin to laminin-binding by EVs. This suggests a potential contribution of tetraspanin-mediated integrin avidity to EV-laminin binding. Overall, as described below, the authors provide significant new data to address several of the major concerns this reviewer raised about the original manuscript; however, there remain some issues that would be good to address.

Thank you very much for your thoughtful comments. In response, we have conducted additional experiments and revised the manuscript in accordance with your suggestions. We believe that these revisions have addressed all the issues.

Original major issue 1: the significance of the study is unclear since no EV-dependent cell behavior or phenotype is investigated, only EV binding.

In response to this critique, the authors provided significant new data showing that EV binding to HUVEC endothelial cells could be suppressed by knockdown of laminin in HUVEC cells. Cellular protrusions elicited by EV exposure were also suppressed in the laminin-depleted cells, and importantly, cellular protrusions elicited by VEGF exposure were not suppressed by laminin depletion. This latter observation was an important control that suggests the depletion of laminin specifically impacted EV-dependent cellular responses. Although the biological significance of the HUVEC cell protrusions in these experiments is not entirely clear, these experiments at least provide proof-of-concept results that integrin-dependent EV binding to pericellular matrix may be able to induce a phenotypic response. In my opinion, these new data adequately address one of the major concerns raised during the first review.

We sincerely appreciate your thoughtful comment regarding EV-dependent cellular responses. The findings on HUVEC protrusion induced by EV binding to laminin have substantially strengthened the manuscript.

Original major issue 2: A significant proportion of the potential significance depends on data

from a preprint that has not yet completed peer review.

The authors report progress on getting the preprint manuscript published in a peer-reviewed journal; however, this has not yet been achieved. The importance of this is that the preprint manuscript contains data that could shed light on the mechanisms by which EVs binding to pericellular matrix might actually influence cell behavior. In the Discussion, the authors cite preprint results including EV endocytosis, and the activation of integrin signaling in target cells upon EV binding as contributing to EV-induced responses in target cells. The JCB editors may decide the extent to which having these data peer reviewed is a prerequisite for publishing this study in JCB.

The manuscript of the study presented in the preprint (Hirosawa et al, bioRxiv, 2024), which was submitted to *Nature Communications*, has been officially accepted and is now in press. The editor's e-mail is provided below.

SUZUKI Kenichi

差出人: robert.stephenson@nature.com
送信日時: 2025年2月21日金曜日 16:33
宛先: SUZUKI Kenichi
件名: Acceptance of NCOMMS-24-23064C

Dear Professor Suzuki,

We are delighted to accept your manuscript entitled "Uptake of small extracellular vesicles by recipient cells is facilitated by paracrine adhesion signaling" for publication in *Nature Communications*. Thank you for choosing to publish your interesting work with us.

Licence to Publish and Article-processing Charge

- In approximately 7-10 business days you will receive an email with a link to choose the grant of rights necessary for publishing your paper and – if applicable – to provide payment information for your article-processing charge (APC), either via credit card or by requesting an invoice.

If needed, our Author Services team will be in touch regarding any additional information that may be required.

In order to avoid any delays, please ensure that you have emails from Springer Nature whitelisted in your mail system. Please note that production will not continue until the Licence to Publish and Article-Processing Charge steps are completed and your proof corrections are submitted.

Original major issue 3: Although the data on EV binding to purified matrix components is convincing, the data on EV binding to live cell-associated matrix proteins is less so. The number

of EVs binding per cell appears relatively low.

The authors clarify that the low number of cell-associated EVs in the original study is due to the use of TIRFM, which can only detect EVs associated with the basal plasma membrane of the target cells. Using a different method in which cells were fixed and then analyzed via confocal microscopy, the authors provide new evidence that a larger number of EVs can bind per cell when both apical and basal EVs are imaged, and that integrin beta1 knockdown can reduce the number of EVs that bind to matrix rich MRC-5 cells, while matrix poor HS-5 cells still bind relatively few EVs. This is important because a low number of EVs associated with the basal plasma membrane could at least in part be explained by cell membrane protrusion over EVs that had already bound the cell-free substrate, rather than active EV binding to pericellular matrix. The data in Figure 4 micrographs show significant numbers of EVs on the substrate outside of cell boundaries. If possible, the authors should comment on any evidence or experimental detail that helps to show that basally located EVs result from active binding to pericellular matrix rather than being simply overspread by the cellular plasma membrane.

Thank you for your valuable comment. In these experiments, not all cells necessarily expressed fluorescent proteins such as mCherry or GFP, as these proteins were transiently expressed. Cells were present on the coverslip in the region where fluorescence was not detected, and sEVs were likely bound to cells that do not express fluorescent proteins. This situation was described in the legend of Figure 4I of the previous version as follows.

“While all the cells do not necessarily express GFP and many fluorescent spots of sEV-CD63Halo7-TMR are observed outside of GFP-expressing cells, we can quantify the number of sEV-CD63Halo7-TMR bound to both the apical and basal surface of the cell PM by counting the number of the fluorescent EV spots on cells expressing GFP.”

However, to provide a more precise explanation, we have revised these sentences as follows and incorporated them at the end of Fig. 4 legend.

“In Fig. 4, since not all cells necessarily express mCherry or GFP, many fluorescent spots of sEVs were observed either in regions containing non-expressing cells or potentially on the glass surface. Nevertheless, we can quantify the number of sEVs bound to both the apical and basal surface of the cell PM by counting the number of fluorescent EV spots on cells expressing mCherry or GFP.”

In addition, integrin β 1KO or α 6KO reduced the number of sEVs in regions containing GFP-expressing cells by approximately half (Figs. 4C, 4E, 4F, and 4H). These findings explicitly show that at least half of the sEVs in areas with GFP-expressing cells were bound to

the cell basal membrane through active binding to pericellular laminin.

To provide a more comprehensive explanation, we incorporated “(approximately twice)” in line 12 on page 19. Furthermore, we added the following sentence in the third-to-second lines from the bottom of p.19.

“These results show that at least half of the sEVs were bound to the basal membrane of MRC-5 cells due to active binding to pericellular laminin.”

Original major issue 4: The potential role of cholesterol in regulating EV binding to laminin is rather speculative, invoking a "molecule X."

The authors responded to this comment by changing the text to remove some of the speculative aspects. However, the results of the cholesterol depletion experiments in Figure 8 and S6 are still difficult to interpret. EVs treated with MBCD to deplete cholesterol from their membranes do appear to bind better to laminin-coated glass and MRC-5 cells. CD151-depleted EVs bind less well to laminin-coated glass, whether they were MBCD-treated or not. The authors interpret this to suggest that "an unidentified molecule interacts with CD151 in a cholesterol-dependent manner to inhibit the binding of EVs to laminin." It is not clear why such an unidentified molecule would necessarily need to physically interact with CD151 to have such an effect. In addition, MBCD treatment can cause the shedding of a subset of integrin-tetraspanin complexes (and GM1) from lipid membranes (Claas et al., *JBC* (276) 7974-7984 2001). The effect of MBCD treatment on the integrin-tetraspanin content of the EVs should therefore be evaluated. Potentially, integrins that are not associated with cholesterol-dependent raft-like domains might more efficiently bind ligands, once raft-associated complexes are depleted.

Thank you for your thoughtful and constructive suggestions. In accordance with the reviewer's recommendation, we performed a co-immunoprecipitation experiment using an anti-integrin $\beta 1$ antibody to examine whether the interaction between CD151 and integrin $\beta 1$ in sEVs is affected by MBCD treatment. The results showed that cholesterol depletion did not alter the interaction (Fig. 8F). Furthermore, we found that cholesterol depletion did not change the interaction of integrin $\beta 1$ with integrin $\alpha 6$ or $\alpha 3$ (Fig. 8F). Accordingly, we incorporated the following sentence in the last sentence on page 28.

“Meanwhile, a co-immunoprecipitation experiment showed that cholesterol depletion did not affect the interaction of integrin $\beta 1$ with CD151, integrin $\alpha 6$, or $\alpha 3$ in sEVs (Fig. 8F).”

Furthermore, in accordance with the reviewer's suggestion, we removed the description of an unidentified molecule that interacts with CD151 in a cholesterol-dependent manner to inhibit

the binding of EVs to laminin on page 29. Instead, as recommended by the reviewer, we incorporated the following sentence in the first paragraph on page 29.

“These results suggest that cholesterol in sEVs suppresses sEV binding to laminin through a CD151-dependent mechanism, while it does not alter integrin heterodimer formation or the integrin-CD151 interaction. Therefore, integrin heterodimers that are not associated with cholesterol-dependent raft-like domains in sEVs, might bind to laminin more effectively once raft-associated complexes are depleted.”

Upon revisiting these data, it also occurs to this reviewer that testing CD151-depleted EVs for binding to collagen might reveal a specificity of CD151 contribution to laminin-binding integrins. The prediction based on the model in Figure 10 is that CD151 depletion would not alter EV binding to collagen-coated glass. Conversely, it would be interesting to determine whether talin depletion did affect EV binding to collagen, despite its apparent lack of effect of EV binding to laminin.

Thank you for your thoughtful suggestion. As the reviewer suggested, we conducted additional experiments to assess sEVs binding to collagen I. As expected, the binding capacity of PC3-derived sEVs to collagen type I was not altered by CD151KD (Fig. 8E). This finding further supports the notion that CD151 selectively facilitates the binding of integrins to laminin. We have incorporated these results in the fifth-to-third lines from the bottom on page 28.

In addition, neither talin-1 KD nor talin-1 overexpression changed the binding affinity of sEVs to collagen type I (Fig. 7K, 7L and 7P). Since talin-1 in sEVs is hardly phosphorylated for activation (Fig. 7Q), it does not regulate the functions of either laminin-binding or collagen-binding integrins. We incorporated these results in the first paragraph of page 25 as follows.

“The number of sEVs attached to collagen type I was not changed by talin-1 KD or talin-1 OE (Fig. 7K and 7L). Moreover, the binding affinity of sEVs to both laminin and collagen type I, derived from talin-1 KD BxPC3 cells, in which the talin-1 level was reduced to 11% of that in sEVs from wild-type cells (Fig. 7M), was not significantly different from that of sEVs derived from wild-type cells (Fig. 7, N-P).”

The mild/moderate issues raised in the preliminary review have largely been addressed in the revision.

Reviewer #2 (Comments to the Authors (Required)):

In this revised manuscript, the authors extend their arguments regarding the criticality of integrin binding to laminin for extracellular vesicle activity. New data are added addressing the conformation of the integrin, molecular dependencies of binding, and the functional consequence of perturbations on endothelial cells. The work is significantly improved and could be published following some clarifications.

Thank you very much for your thoughtful comments. In response, we have revised the manuscript in accordance with the reviewer's suggestions.

1. Fig 5F - stats should be provided for WT vs ITGA6 KO

Thank you for your comment. We have incorporated the result of Welch's t-test for WT vs. ITGA6KO in the graph of Fig. 5F.

2. Why do there appear to be different levels of EV binding to the surrounding coverslip in the laminin depleted HUVEC image?

Thank you for your thoughtful comment. The reason for this result aligns with point 3 of Reviewer #1's feedback. Specifically, in this experiment (Fig. 9B), not all HUVEC expressed GFP, as GFP was transiently expressed in the cells. Consequently, some cells were located on the coverslip in regions where fluorescence was not detected. In the legend of Fig. 9B, we described the situation as follows.

“Since not all cells necessarily express GFP, many fluorescent spots of sEVs were observed either in regions containing non-expressing cells or potentially on the glass surface. Nevertheless, the number of sEVs bound to both the apical and basal surface of the cell PM can be quantified, as shown in Fig. 4.”

February 28, 2025

RE: JCB Manuscript #202404064RR

Kenichi Suzuki
Gifu University

Dear Prof. Suzuki,

Thank you for resubmitting your manuscript entitled "Extracellular vesicles adhere to cells primarily by interactions of integrins and GM1 with laminin." Your conscientious responses were deemed sufficient to resolve directly and effectively the remaining concerns of the expert reviewers. Nevertheless, in examining the statistical analysis now provided for Fig. 5F, the use of Welch's t-test did not seem appropriate for cross-comparing more than a single pair of conditions; that is, its correction for differing variances is helpful, but an uncorrected t-test should be used only for an isolated pair of conditions. Instead, either a t-test with Bonferroni correction or use of ANOVA with an appropriate post hoc test would be needed. The same issue is seen for Fig. 8C. Applying a more-appropriate test will not change the conclusions, but will satisfy statisticians. We would be happy to publish your paper in JCB pending final revisions to address this issue as well as to meet our formatting guidelines (see details below).

A. MANUSCRIPT ORGANIZATION AND FORMATTING:

1) Text limits: Character count for Articles is < 40,000, not including spaces. Count includes title page, abstract, introduction, results, discussion, and acknowledgments. Count does not include materials and methods, figure legends, references, tables, or supplemental legends.

2) Figure formatting: Articles may have up to 10 main text figures. Scale bars must be present on all microscopy images, including inset magnifications. Molecular weight or nucleic acid size markers must be included on all gel electrophoresis. Also, please avoid pairing red and green for images and graphs to ensure legibility for color-blind readers. If red and green are paired for images, please ensure that the particular red and green hues used in micrographs are distinctive with any of the colorblind types. If not, please modify colors accordingly or provide separate images of the individual channels.

3) Statistical analysis: Error bars on graphic representations of numerical data must be clearly described in the figure legend. The number of independent data points (n) represented in a graph must be indicated in the legend. Please, indicate whether 'n' refers to technical or biological replicates (i.e. number of analyzed cells, samples or animals, number of independent experiments). If independent experiments with multiple biological replicates have been performed, we recommend using distribution-reproducibility SuperPlots (please see Lord et al., JCB 2020) to better display the distribution of the entire dataset, and report statistics (such as means, error bars, and P values) that address the reproducibility of the findings.

Statistical methods should be explained in full in the materials and methods. For figures presenting pooled data the statistical measure should be defined in the figure legends. Please also be sure to indicate the statistical tests used in each of your experiments (both in the figure legend itself and in a separate methods section) as well as the parameters of the test (for example, if you ran a t-test, please indicate if it was one- or two-sided, etc.). Also, if you used parametric tests, please indicate if the data distribution was tested for normality (and if so, how). If not, you must state something to the effect that "Data distribution was assumed to be normal but this was not formally tested."

4) Materials and methods: Should be comprehensive and not simply reference a previous publication for details on how an experiment was performed. Please provide full descriptions (at least in brief) in the text for readers who may not have access to referenced manuscripts. The text should not refer to methods "...as previously described." Please also indicate the acquisition and quantification methods for immunoblotting/western blots.

5) For all cell lines, vectors, constructs/cDNAs, etc. - all genetic material: please include database / vendor ID (e.g. Addgene, ATCC, etc.) or if unavailable, please briefly describe their basic genetic features, even if described in other published work or gifted to you by other investigators (and provide references where appropriate). Please be sure to provide the sequences for all of your oligos: primers, si/shRNA, RNAi, gRNAs, etc. in the materials and methods. You must also indicate in the methods the source, species, and catalog numbers/vendor identifiers (where appropriate) for all of your antibodies, including secondary. If antibodies are not commercial, please add a reference citation if possible.

- 6) Microscope image acquisition: The following information must be provided about the acquisition and processing of images:
- Make and model of microscope
 - Type, magnification, and numerical aperture of the objective lenses
 - Temperature
 - Imaging medium
 - Fluorochromes
 - Camera make and model
 - Acquisition software
 - Any software used for image processing subsequent to data acquisition. Please include details and types of operations involved (e.g., type of deconvolution, 3D reconstitutions, surface or volume rendering, gamma adjustments, etc.).
- 7) References: There is no limit to the number of references cited in a manuscript. References should be cited parenthetically in the text by author and year of publication. Abbreviate the names of journals according to PubMed.
- 8) Supplemental materials: Articles may have up to 5 supplemental figures and 10 videos. You currently exceed this limit but, in this case, we will be able to give you the extra space. Please also note that tables, like figures, should be provided as individual, editable files. A summary of all supplemental material should appear at the end of the Materials and methods section. Please include one brief sentence per item.
- 9) Video legends: Each video should have a separate legend describing what is being shown, the cell type or tissue being viewed (including relevant cell treatments, concentration and duration, or transfection), the imaging method (e.g., time-lapse epifluorescence microscopy), what each color represents, how often frames were collected, the frames/second display rate, and the number of any figure that has related video stills or images.
- 10) eTOC summary: A ~40-50 word summary that describes the context and significance of the findings for a general readership should be included on the title page. The statement should be written in the present tense and refer to the work in the third person. It should begin with "First author name(s) et al..." to match our preferred style.
- 11) Conflict of interest statement: JCB requires inclusion of a statement in the acknowledgements regarding competing financial interests. If no competing financial interests exist, please include the following statement: "The authors declare no competing financial interests." If competing interests are declared, please follow your statement of these competing interests with the following statement: "The authors declare no further competing financial interests."
- 12) A separate author contribution section is required following the Acknowledgments in all research manuscripts. All authors should be mentioned and designated by their first and middle initials and full surnames. We encourage use of the CRediT nomenclature (<https://casrai.org/credit/>).
- 13) ORCID IDs: ORCID IDs are unique identifiers allowing researchers to create a record of their various scholarly contributions in a single place. Please note that ORCID IDs are required for all authors. At resubmission of your final files, please be sure to provide your ORCID ID and those of all co-authors.
- 14) JCB requires authors to submit Source Data used to generate figures containing gels and Western blots with all revised manuscripts. This Source Data consists of fully uncropped and unprocessed images for each gel/blot displayed in the main and supplemental figures. For assays performed using capillary electrophoresis and/or immunoassay-based detection, authors should instead provide the electropherogram graph(s) for each experiment, plotting fluorescence/chemiluminescence intensity vs. molecular weight/size. Since your paper includes cropped gel and/or blot images, please be sure to provide one Source Data file for each figure gels, blots, and/or capillary electrophoresis assays along with your revised manuscript files. File names for Source Data figures should be alphanumeric without any spaces or special characters (i.e., SourceDataF#, where F# refers to the associated main figure number or SourceDataFS# for those associated with Supplementary figures). For traditional gels and blots, the lanes of the gels/blots should be labeled as they are in the associated figure, the place where cropping was applied should be marked (with a box), and molecular weight/size standards should be labeled wherever possible. For capillary electrophoresis assays, each trace in the graph should be color-coded and labeled to indicate which protein, gene, or sample is being measured (please try to avoid red/green combinations to accommodate our color-blind readers).
- Source Data files will be directly linked to specific figures in the published article. Source Data Figures should be provided as individual PDF files (one file per figure). Authors should endeavor to retain a minimum resolution of 300 dpi or pixels per inch. Please review our instructions for export from Photoshop, Illustrator, and PowerPoint here: <https://rupress.org/jcb/pages/submission-guidelines#revised>
- 15) Journal of Cell Biology now requires a data availability statement for all research article submissions. These statements will be published in the article directly above the Acknowledgments. The statement should address all data underlying the research presented in the manuscript. Please visit the JCB instructions for authors for guidelines and examples of statements at (<https://rupress.org/jcb/pages/editorial-policies#data-availability-statement>).

B. FINAL FILES:

****It is JCB policy that if requested, original data images must be made available to the editors. Failure to provide original images upon request will result in unavoidable delays in publication. Please ensure that you have access to all original data images prior to final submission.****

****The license to publish form must be signed before your manuscript can be sent to production. A link to the electronic license to publish form will be sent to the corresponding author only. Please take a moment to check your funder requirements before choosing the appropriate license.****

Thank you for your attention to these final processing requirements. Please revise and format the manuscript and upload materials within 7 days. If you need an extension for whatever reason, please let us know and we can work with you to determine a suitable revision period. Please contact the journal office with any questions at cellbio@rockefeller.edu.

Thank you for this interesting contribution. We hope that you will agree that the rigorous reviewing at JCB has resulted in a highly rigorous paper of which you and your colleagues can be proud. We look forward to receiving the final version of your manuscript and publishing your paper in Journal of Cell Biology.

Sincerely,

Kenneth Yamada, MD, PhD
Editor
Journal of Cell Biology

Dan Simon, PhD
Scientific Editor
Journal of Cell Biology